# A transparent hybrid metal halide glassy scintillation screen for high-resolution fast neutron radiography

Zi'an Zhou [1,2], Jinxiao Zheng [1] ✉, Shihao Ruan[3], Guichu Yue[4], Tiao Feng[1,2], Yini An[1,2], Meimei Wu[3], Nü Wang[5], Shuyun Zhou[1], Linfeng He [3] ✉ & Chenghua Sun [1] ✉

Fast neutron radiography offers exceptional penetration for high-density and bulky objects, yet its resolution is hindered by light scattering in conventional scintillators and screen fabrication techniques. To address this, here, we develop a transparent glassy Mn-based hybrid metal halide scintillation screen, $(BTPP)_{1.8}(HTPP)_{0.2}MnBr_4$ (BTPP$^+$ = butyltriphenylphosphonium, HTPP$^+$ = heptyltriphenylphosphonium), leveraging temperature-dependent ordered-disordered transitions. The large-area screen boasts >70% visible light transmittance (500–800 nm), a high photoluminescence quantum yield (~85.54%), and threefold higher light output than commercial ZnS (Ag): PP screens. With a spatial resolution of 5 lp mm$^{-1}$, it surpasses existing scintillators. This hybrid material enables imaging of heavy objects with clear hierarchical details, providing accurate data for non-destructive detection while offering an alternative approach to scintillator design, advancing the potential of fast neutron radiography.

Neutrons, renowned for their exceptional penetration ability and immunity to external electrostatic fields[1–3], act as valuable probes for detecting the internal structure of matter. Depending on neutron energies, neutron radiography technology mainly includes cold neutron (<0.005 eV)[4], thermal neutron (0.005 - 0.5 eV)[5], and fast neutron (>0.5 MeV) radiography[6]. Thus, fast neutron radiography (FNR) possesses higher energy and unparalleled penetration capabilities, making it irreplaceable for detecting internal structures and low-density defects within large-scale and high-density equipments[7–9]. Moreover, fast neutrons binding detectors and related analytical methods offer distinct advantages in isotope differentiation and radioactive material detection[10,11], making them indispensable across diverse domains including aerospace, nuclear industry, energy, explosion-proof detection, archaeology, etc[10,12–14].

High resolution FNR is the prerequisite to realize its inherent advantages and further meet rising demand. Commonly used scintillation screens for FNR include organic crystal scintillators (such as anthracene[15] and stilbene[16]) and ZnS (Ag): polypropylene (PP) or ZnS (Cu): PP composite screens[17], due to their high hydrogen density and favorable luminescent performance. However, organic scintillators are difficult to fabricate, expensive, fragility, and sensitivity to storage conditions[18]. Meanwhile, ZnS (Ag): PP or ZnS (Cu): PP composite screens suffer from inadequate detection efficiency and spatial resolution owing to the energy transfer loss and light scattering arising from the non-uniform distribution of two components[9]. This implies that single-component scintillator composed of hydrogen-rich unit and luminous unit could be more suitable for high resolution FNR.

Hybrid perovskites contain an organic unit which should be a good candidate for neutron absorption. And the distinctive electronic structure of the inorganic moieties gives it excellent fluorescence emission and a tunable band gap in the visible light range[19,20]. In recent years, hybrid perovskites have been shown as promising candidates

[1]Key Laboratory of Photochemical Conversion and Optoelectronic Materials, Technical Institute of Physics and Chemistry, Chinese Academy of Sciences, Beijing, China. [2]University of Chinese Academy of Sciences, Beijing, China. [3]China Institute of Atomic Energy, Beijing, China. [4]College of Chemical Engineering, Inner Mongolia University of Technology, Hohhot, China. [5]School of Chemistry, Beihang University, Beijing, China. ✉e-mail: zhengjinxiao@mail.ipc.ac.cn; hlf1212@sina.com; sunchenghua@mail.ipc.ac.cn

for scintillators due to their ability to emit visible light when excited by high-energy rays (X-ray, β-ray, γ-ray, protons, neutrons)[21–25]. The Mn-$(C_{18}H_{37}NH_3)_2PbBr_4$ (Mn-STA$_2$PbBr$_4$) perovskite scintillation screen, prepared through the compression method, has achieved fast neutron imaging[26]. However, current perovskite fast neutron scintillation screens are still made of powders, leading to significant light scattering and poor imaging resolution, thereby hindering the development of FNR[13,27]. On this basis, it is imperative to design a transparent hybrid metal halide with a perovskite-derived structure to serve as a neutron scintillator material to realize full potential of FNR.

Herein, we develop a large-area glassy state hybrid metal halide high-transparent scintillation screen $(BTPP)_{1.8}(HTPP)_{0.2}MnBr_4$ (BTPP$^+$ = butyltriphenylphosphonium, HTPP$^+$ = heptyltriphenylphosphonium) achieving high-resolution FNR. The transparent screen is a single-component fast neutron scintillator composed of triphenylphosphonium hydrogen-rich unit and $[MnBr_4]^{2-}$ luminous unit. The high-steric-hindrance of triphenylphosphonium unit ensures that the molecules remain in an amorphous state during the quenching process, thus obtaining a highly transparent glassy state. Notably, the scintillation screen has high transmittance over 70% (500–800 nm), high photoluminescence quantum yield (PLQY) of 85.54%, and a light output three times greater than that of commercial scintillation screens like ZnS (Ag): PP. Furthermore, it achieves a remarkable resolution of 5 line pair mm$^{-1}$ (lp mm$^{-1}$), which is higher than currently reported in the field of FNR. This strategy offers insights for designing high performance hybrid metal halide transparent glass materials.

## Results and Discussion

### Design of transparent fast neutron scintillation screens

Figure 1 illustrates the advantages of a transparent fast neutron scintillation screen over conventional scintillation screens. During the FNR, the neutron ray firstly penetrates target tested object, then is decayed due to a series of reactions such as absorption, elastic scattering, and inelastic scattering with nuclei. The penetrated neutron ray thereby carries the spatial structural information of the object. Then the scintillation screen converts the invisible neutron ray information from the sample into visible light signal through luminescence excited by recoil protons energy deposition, which is subsequently captured by charge-coupled device (CCD). The clarity of the obtained image is closely associated with the quality of the scintillation screen. Conventional fast neutron scintillation screens, typically composed of powder materials, result in emitted light scattering in various directions, leading to poor spatial resolution and consequently blurry images. So, a transparent scintillation screen effectively mitigates the light scattering issue, enabling clear imaging.

The design of a transparent scintillation screen for FNR must satisfy four fundamental criteria[28]: (1) a high hydrogen density to ensure effective recoil proton generation, (2) a high PLQY to guarantee sufficient radiative emission, (3) enhanced optical transparency to minimize light scattering, thereby improving light extraction efficiency and (4) adequate mechanical integrity to support practical usability in FNR. Guided by these criteria, we develop a transparent scintillation screen $(BTPP)_{1.8}(HTPP)_{0.2}MnBr_4$ using a one-step solvent-free melt quenching method, which can meet the aforementioned

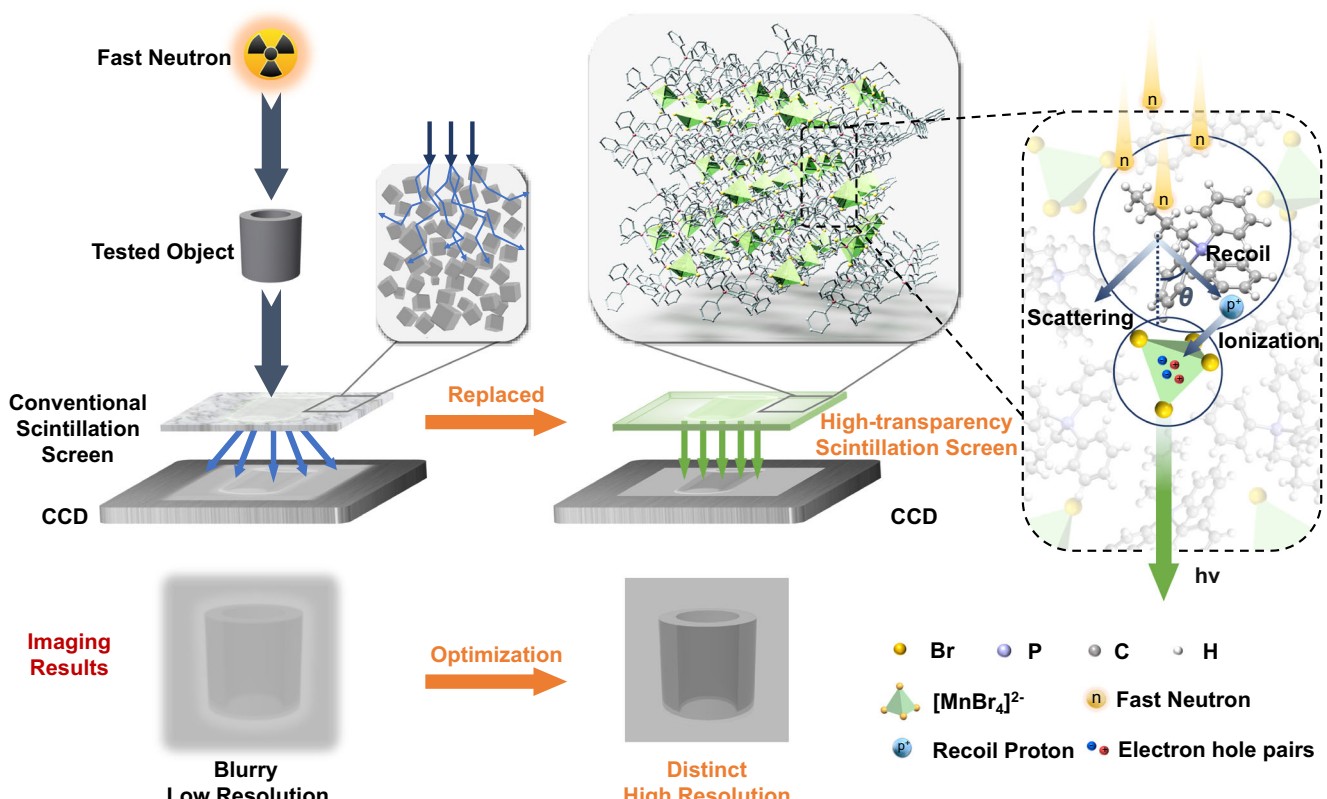

**Fig. 1 | Design of an alternative FNR strategy.** In the FNR approach based on conventional scintillation screen, blurry and poor resolution imaging is obtained due to the light scattering of grain boundary or bi-phase interface. The alternative FNR strategy is based on single-component high-transparent scintillation screen $(BTPP)_{1.8}(HTPP)_{0.2}MnBr_4$, which is composed of randomly arranged high-steric-hindrance cations (BTPP$^+$, HTPP$^+$) and luminous unit $[MnBr_4]^{2-}$. The fast neutrons carrying the internal information of the tested object, interact with H atom in the organic cation BTPP$^+$, HTPP$^+$ to generate recoil protons. The recoil protons deposit energy on the luminous unit $[MnBr_4]^{2-}$ and induce the generation of electron hole pairs, stimulating visible fluorescence, and then the fluorescence information is received by charge-coupled device (CCD). Distinct and high-resolution imaging is obtained, thanks to the transparency of the scintillation screen reducing light scattering.

requirements. To achieve the target material, we utilize $BTPP^+$ as the main monovalent $A^+$ cation due to its high steric hindrance, which facilitates the formation of a transparent medium during quenching while also providing interaction sites for fast neutrons[29]. We also dope the other triphenylphosphonium organic cations with different branched chain lengths ($HTPP^+$, $DTPP^+$ = dodecyltriphenylphosphonium, $CTPP^+$ = cetyltriphenylphosphonium) to increase the flexibility of the transparent medium and its hydrogen density. Fast neutrons undergo elastic scattering with hydrogen nuclei from the $A^+$ cations, resulting in the production of recoil protons. The recoil protons deposit energy on the luminescent center ($[MnBr_4]^{2-}$), thereby inducing visible light emission. The high transparency of the $(BTPP)_{1.8}(HTPP)_{0.2}MnBr_4$ can effectively resolve the issues of energy transfer inefficiency and light scattering which plaguing conventional fast neutron scintillation screens.

## Synthesis and performance optimization

To meet the high hydrogen density requirements of fast neutron scintillation screen, we calculate the hydrogen density, elastic scattering cross section, and fast neutron stopping power of the transparent scintillation screens with various doping components, based on their composition and density. ZnS (Ag): PP scintillators and other commonly used scintillators are used as references (Supplementary Table 1). The results show that the hydrogen density, elastic scattering cross section and fast neutron stopping power of the transparent scintillator with different components ranged from 68.92 to $87.09\ kg\ m^{-3}$, 0.2698 to $0.3052\ cm^{-1}$, and 2.662% to 3.006%, respectively. These values are comparable to those of stilbene ($67.33\ kg\ m^{-3}$, $0.2690\ cm^{-1}$ and 2.654%) and anthracene ($71.91\ kg\ m^{-3}$, $0.3100\ cm^{-1}$ and 3.052%), but are lower than ZnS (Ag): PP ($136.9\ kg\ m^{-3}$, $0.4556\ cm^{-1}$ and 4.454%). For a series of transparent scintillation screens, we observe that increasing the length of the doped branched alkyl chain results in higher hydrogen density, which subsequently enhances the scattering cross section and improves the fast neutron stopping ability.

The transparent scintillation screens are synthesized using a modified one-step solvent-free melt quenching method (Fig. 2a)[30]. Specifically, $MnBr_2$ and triphenylphosphonium bromide (TPPBr) with different branched alkyl chain lengths (BTPPBr, HTPPBr, DTPPBr, CTPPBr) are mixed in a mortar with predetermined molar ratios (The selection of the optimal cations and its ratio are shown in Supplementary Figs. 1–3). Taking $(BTPP)_{1.8}(HTPP)_{0.2}MnBr_4$ as an example, the mixture is then ground thoroughly and transferred to a glass culture dish, where it is stirred and heated until it completely melts into a transparent yellow-green liquid. During this process, a reaction occurs between BTPPBr, HTPPBr and $MnBr_2$. The above molten liquid is poured into a preheated silicone mold and quenched rapidly to room temperature (RT), resulting in transparent media with controllable shape and thickness (refer to the Methods section for detailed procedures). Silicone molds are chosen for their softness, high temperature resistance, and ease of demolding. These molds allow for the efficient and quick production of transparent scintillation screens of arbitrary shapes and sizes which emit intense green light when exposed to a 365 nm ultraviolet lamp. The transparent media provide the potential for meeting the diverse size and shape requirements of different applications in the optical field. In addition to quenching operation, it is discovered that by manipulating the holding time and annealing temperature at around 110 °C, amorphous $(BTPP)_{1.8}(HTPP)_{0.2}MnBr_4$ can be transformed into polycrystalline state (Supplementary Fig. 4). To gain further insights into the conversion process between glassy transparent media and polycrystals, we also prepare $(BTPP)_2MnBr_4$ single crystal using a slow solvent evaporation method (Supplementary Fig. 5)[31]. The detailed crystallographic data of $(BTPP)_2MnBr_4$ are listed in Supplementary Table 2. Next, we conduct a detailed analysis of as-prepared several materials. Firstly, we compare the powder X-ray diffraction

(PXRD) spectra of $(BTPP)_2MnBr_4$/ $(BTPP)_{1.8}(HTPP)_{0.2}MnBr_4$/ $(BTPP)_{1.8}(DTPP)_{0.2}MnBr_4$/ $(BTPP)_{1.8}(CTPP)_{0.2}MnBr_4$ transparent media, $(BTPP)_2MnBr_4$ in polycrystalline experiments and the single crystal X-ray diffraction (SCXRD) simulation of $(BTPP)_2MnBr_4$ single crystal, respectively (Fig. 2b and Supplementary Fig. 6). These transparent media exhibit a bulging structure of amorphous phases, which is related to the loss of long-range ordering caused by random molecular arrangement during the formation process of transparent media[30,32]. To investigate the effect of doping organic cations with different branched alkyl chain lengths on the mechanical strength of the prepared transparent scintillation screen, we conduct three-point bending test and elastic modulus test (Fig. 2c and Supplementary Figs. 7, 8)[33]. The displacement curve gradually lengthens and the elastic modulus decreases while the bending strength gradually increases with the longer doped branched alkyl chain, indicating better toughness of the scintillation screen (Fig. 2c and Supplementary Fig. 8)[34–36]. The main reason for this is the softer lattice caused by the larger size of the cation[37], which helps improve the mechanical toughness of the scintillation screen, preventing fracture during operation and increasing practicality.

While long branched alkyl chains can enhance the hydrogen density and the toughness of scintillation screens, it is also necessary to consider the impact on luminescent performance. The normalized photoluminescence (PL) emission peak of transparent media doped with different branched alkyl chain lengths remains at 525 nm, with a full width at half maximum (FWHM) of 59 nm (Fig. 2d). In addition, the high overlap of the normalized photoluminescence excitation (PLE) spectra indicates that the emission of the transparent media originates from the same excited state relaxation, regardless of the cations size. Therefore, it can be concluded that the green luminescence of transparent medium originates from the characteristic emission of $Mn^{2+}$ $^4T_1(G) \rightarrow {}^6A_1$[38,39]. The $[MnBr_4]^{2-}$ ions are a typical tetrahedral coordination structure, similar to that of other zero-dimensional (0D) Mn-based metal halides, but different from the octahedral coordination structure of one-dimensional (1D) Mn-based perovskites[40]. Moreover, the PLQY is basically not impacted by small amount of doping with $HTPP^+$, but it decreases after doping $DTPP^+$ and $CTPP^+$ due to their overly long chain lengths leading to excessive spatial separation and self-absorption (Fig. 2e, Supplementary Figs. 9–12)[41–43]. It indicates that the trade-off between the luminescence performance and screen toughness needs to be synthetically considered. Therefore, we select the $(BTPP)_{1.8}(HTPP)_{0.2}MnBr_4$ sample for further application exploration. After melting and cooling 10 cycles repeatedly, the PLQY remains basically unchanged (Fig. 2e), which means that the optical performance of this transparent medium is stable and it can be recycled and reused[44]. We also obtain the PLQY of the $(BTPP)_2MnBr_4$ single crystal and polycrystal, which are 88.54% and 89.24%, respectively (Supplementary Fig. 13). Furthermore, the average decay times of $(BTPP)_2MnBr_4$ single crystal and polycrystal are 295 μs and 303 μs, respectively (Supplementary Fig. 14). However, the decay times of transparent media including $(BTPP)_2MnBr_4$/ $(BTPP)_{1.8}(HTPP)_{0.2}MnBr_4$/ $(BTPP)_{1.8}(DTPP)_{0.2}MnBr_4$/ $(BTPP)_{1.8}(CTPP)_{0.2}MnBr_4$, are shortened to 166 μs, 152 μs, 159 μs, 137 μs, respectively (Supplementary Fig. 15), which is beneficial for rapid response to fast neutron. Remarkably, the transmittance of the four transparent scintillation screens exceeds 70% in the visible light range of 500−800 nm, while the transmittance of the polycrystalline counterpart is only about 50% (Fig. 2f). This indicates that compared to the opaque polycrystalline, the transparent scintillation screen greatly reduces light scattering and shows great potential in high-resolution optical imaging applications.

## Controllable glass-crystal transformation

To gain a comprehensive insight of the luminescence behavior of transparent scintillation screen, we also study the temperature-dependent PL spectra of $(BTPP)_{1.8}(HTPP)_{0.2}MnBr_4$ within the

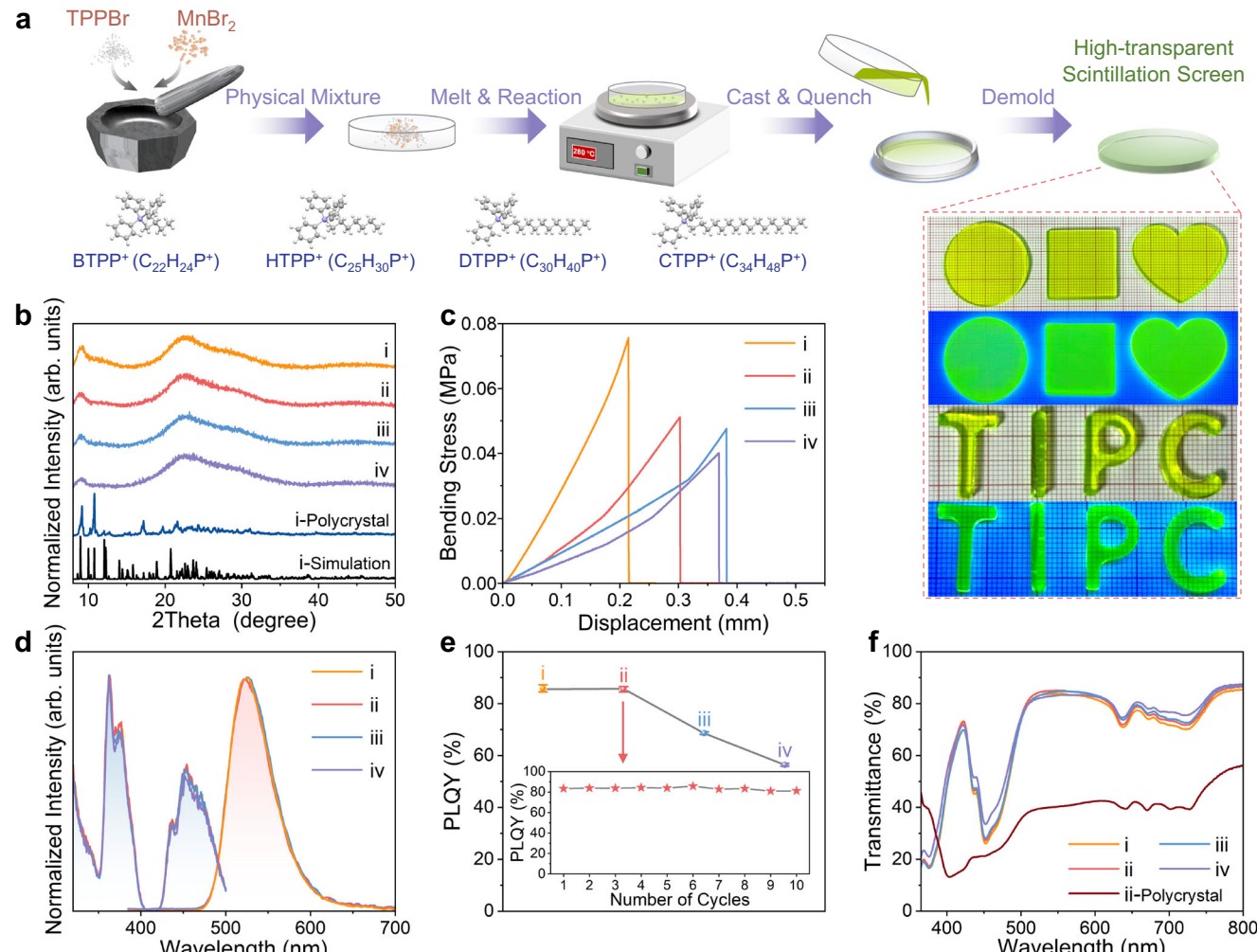

**Fig. 2 | Preparation and characterization of hybrid metal halide transparent scintillation screen. a** Preparation scheme of transparent scintillation screen. MnBr$_2$ and TPPBr with different branched alkyl chain lengths are physically mixed, then heated and melted, and the resulting melt is quenched at a preheated silicone mold to form a transparent media with controllable shape and thickness. **b** The PXRD of the (i)-(iv) quenching transparent scintillation screens show amorphous transparent glassy state, while PXRD of (i)-polycrystal by slow annealing method showing polycrystalline state, basically corresponds with the SCXRD simulated pattern. **c** The bending stress-displacement curves indicate that the longer doping chain, the stronger mechanical strength of the scintillation screen. **d** The normalized PL, PLE and **e** PLQY of the (i)-(iv) transparent scintillation screens (Error bars are presented as mean ± standard deviation (SD), $n = 3$ presents three independent experiments). The inset shows the PLQY of the (ii) transparent media which is melted and cooled repeatedly for 10 cycles. **f** The transmittance of the (i)-(iv) glassy and (ii)-polycrystalline scintillation screen. ((i): (BTPP)$_2$MnBr$_4$, (ii): (BTPP)$_{1.8}$(HTPP)$_{0.2}$MnBr$_4$, (iii): (BTPP)$_{1.8}$(DTPP)$_{0.2}$MnBr$_4$, (iv): (BTPP)$_{1.8}$(CTPP)$_{0.2}$MnBr$_4$).

temperature range of 30−150 °C, and then quenching to 30 °C (Fig. 3a). It can be seen that the luminescence intensity of (BTPP)$_{1.8}$(HTPP)$_{0.2}$MnBr$_4$ gradually decreases and disappears as the temperature increases from 30 °C to 90 °C. However, at around 100 °C, the luminescence peak reappears, and reaches its maximum at 110 °C. Meanwhile, the emission peak position undergoes a blue shift because of the lattice thermal expansion effect (Supplementary Fig. 16)[39]. This indicates that (BTPP)$_{1.8}$(HTPP)$_{0.2}$MnBr$_4$ exhibits another state within the temperature range of 100−110 °C. Subsequently, the intensity gradually attenuates as the temperature continues to rise, eventually disappearing above 150 °C, which can be attributed to the sample melting into a liquid state. Upon quenching back to 30 °C, the fluorescence fully recovers.

Next, we conduct a series of tests at different temperatures to further investigate the transformation between the glassy state and polycrystalline state. Differential scanning calorimetry (DSC) is used to study temperature-induced changes in molecular structure states (Fig. 3b and Supplementary Figs. 17, 18). The exothermic peak nears 109.85 °C and an endothermic peak appears at 136.85 °C, which represents the crystallization temperature ($T_c$) and the melting

temperature ($T_m$), respectively[45] (Fig. 3b). This temperature point, at which the degree of molecular order changes, coincides with that of the temperature-dependent PL spectra in Fig. 3a. Then, we conduct temperature-dependent XRD testing to further investigate the structural transition (Fig. 3c). The transparent media at RT exhibits an amorphous dispersion peak. As the temperature increases above 70 °C, diffraction peaks gradually appear and sharpen, accompanied preferred orientation along specific crystallographic planes. The diffraction peak at 110 °C is approximately consistent with the crystalline state, indicating that the degree of molecular order in the (BTPP)$_{1.8}$(HTPP)$_{0.2}$MnBr$_4$ transparent media undergoes a rearrangement at $T_c$ and becomes more orderly. As the temperature further raises above 130 °C, the diffraction peaks disappear again, implying that the (BTPP)$_{1.8}$(HTPP)$_{0.2}$MnBr$_4$ melts. After quenching to RT, the dispersion peak reappears, indicating that the sample returns to a glassy state. If the sample is annealed directly to RT at 110 °C ($T_c$), the diffraction peaks of the polycrystals are preserved (Fig. 3d).

The temperature-dependent polarized optical microscopy (POM) test is also performed, which clearly shows the changes between the

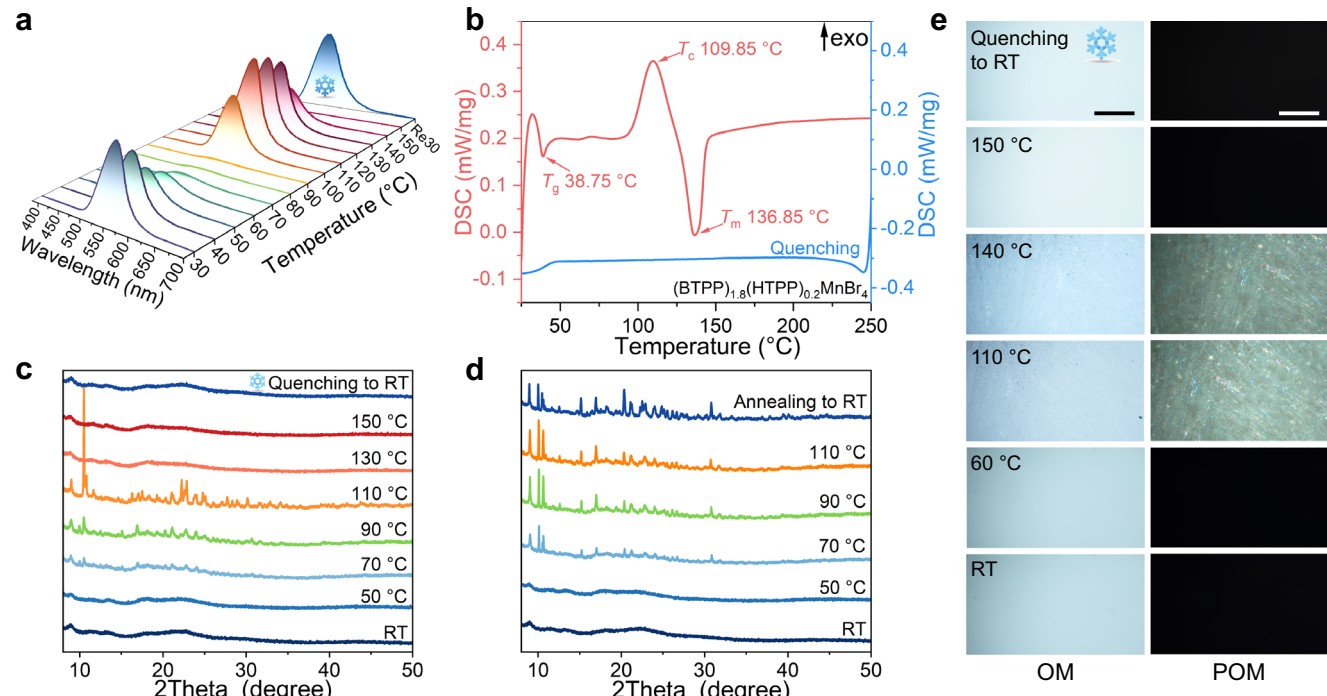

**Fig. 3 | Characterizations of (BTPP)$_{1.8}$(HTPP)$_{0.2}$MnBr$_4$ transparent scintillation screen and crystalline transformation. a** Temperature-dependent PL spectra from 30 °C to 150 °C, then quenching to 30 °C, indicating that as the temperature increases, the material state changes, leading to changes in the fluorescence performance (The snowflake represents quenching to RT). **b** The DSC of the (BTPP)$_{1.8}$(HTPP)$_{0.2}$MnBr$_4$ transparent medium crystallizes at around 110 °C, then melts at 150 °C (Red) and is quenched to RT (Blue). **c** Formation of transparent glassy state: temperature-dependent XRD from RT to 150 °C, then quenching to RT. **d** Formation of poly-crystalline state: temperature-dependent XRD from RT to 110 °C, then slowly annealing to RT. The diffraction peak of the crystal state remains unchanged after annealing at 110 °C (T$_c$) to RT. **e** Pictures of (BTPP)$_{1.8}$(HTPP)$_{0.2}$MnBr$_4$ transparent medium under OM and POM that heating up from RT to 150 °C and then quenching to RT, indicating that crystallization begins at 110 °C and disappears in the field of view at 150 °C. Scale bars: 200 μm.

glassy state, polycrystalline state, and molten state (Fig. 3e and Supplementary Figs. 19, 20). Glass and melt are disordered states with many similarities, exhibiting single refraction and a completely dark field of view under the POM[46]. Due to the anisotropy of polycrystalline media, birefringence occurs, resulting in a bright field of view under the POM, which can clearly distinguish the crystallization situation[47]. From Fig. 3e and Supplementary Fig. 19, we can clearly see the reversible transition between the glassy and crystalline states. However, these distinct phenomena are difficult to obtain under ordinary optical microscopes (OM). Furthermore, after crystallization occurs at the temperature of 110 °C, the opaque polycrystalline sample can be obtained by directly annealing to RT (Supplementary Fig. 20). The conversion process is consistent with above temperature change tests.

**Glassy structure and its transparent imaging mechanism**
To further understand the formation mechanism of glassy and crystalline states, we conduct ab initio molecular dynamics (AIMD) simulations using the CP2K package to study the evolution of molecules and atoms during the processes of melting, quenching, and crystallizing at elevated temperature, lowered temperature and T$_c$[48]. In the simulations, we use the Nose-Hoover thermostat (canonical ensemble, NVT) for melting and crystallizing, and microcanonical ensemble (NVE) for quenching process[49,50]. Due to the computational time cost of AIMD, the time scale of the AIMD simulations (typically within the picosecond range) is significantly shorter than that of the actual experimental melting process. Therefore, simulations with a total run time of up to 10 ps are performed at the density functional theory (DFT) level, using a time step of 1.0 fs. Meanwhile, we choose 1600 K which is much higher than the actual melting temperature to simulate the molten state of transparent media. The molecular structures are shown in Fig. 4a, representing as the initial single crystal cell at 300 K,

the molten state at 1600 K, the polycrystalline state after annealing at 360 K, and the glassy state after quenching from 1600 K to 300 K, respectively[44,51]. In order to determine the distance between specific atomic more intuitively, the radial distribution function (RDF) is used to characterize the likelihood of finding a particle at a given distance $r$ from another particle. Consequently, RDF determined for specific atomic pairs Mn-P and P-P delineates particular bond lengths and their long-range ordering. As shown in Fig. 4b and 4c, with the molecular state changes from single crystal state to molten state, the distance distribution of Mn-P and P-P significantly widens, indicating an increase in thermal vibration of particles and the loss of long-range order. The generalized Lindemann ratio is a widely accepted criterion for assessing the occurrence of melting, conventionally defined as a threshold of 15%. Based on the calculations using $G_{\text{Mn-P}}(r)$ and $G_{\text{P-P}}(r)$, it can be observed that the Lindemann ratio surpasses 15% at 1600 K, signifying the attainment of a molten state within the system (Supplementary Fig. 21). When quenching from molten state to glassy state, the RDF does not revert to the initial single crystal cell pattern, and although the disorder decreases, it still exists, indicating the formation of glassy state.

Additionally, AIMD simulations are performed to determine the mean square displacement (MSD) at various temperature. As the temperature increases, the diffusion coefficient of the sample gradually increases, which illustrates liquid-like diffusion behavior within the molten state (1600 K)[52] (Supplementary Fig. 22). In order to gain a deeper understanding of the mechanism of glassy state formation, we conduct the MSD (all elements, element P and Mn) at 10 ps time scale simulating the quenching process (Fig. 4d and Supplementary Fig. 23). The MSD of the system diminishes as the simulation time progresses indicating that during the quenching process, the disordered state of the melt can gradually converge, ultimately forming a glassy state[46].

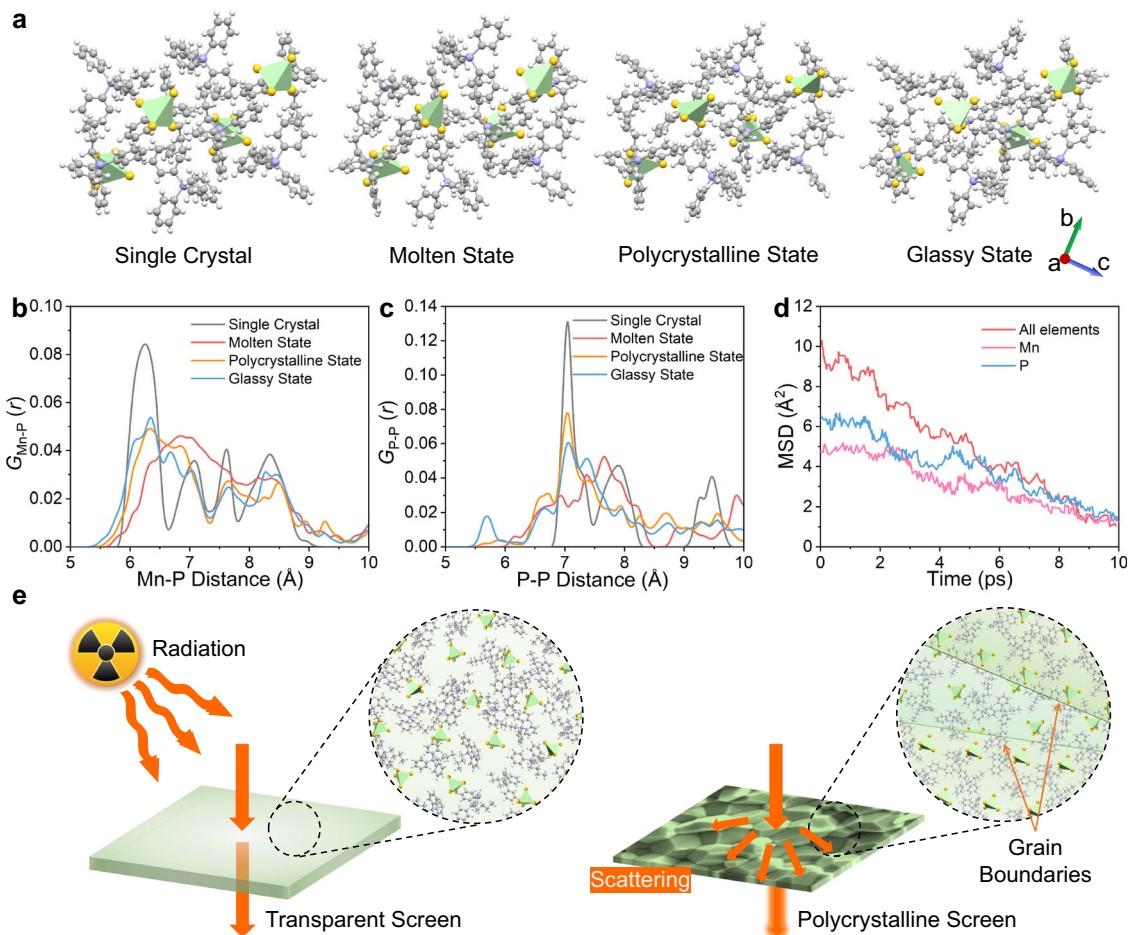

**Fig. 4 | AIMD and light transmission mechanism diagram for glassy and poly-crystalline state scintillation screens. a** Molecular structures of single crystal, molten state, polycrystalline state, and glassy state. Due to the time scale limitation of AIMD simulation, a much higher melting temperature of 1600 K is chosen to simulate the molten state of transparent media. **b** The RDF of $G_{Mn-P}(r)$ for Mn-P distance. **c** $G_{P-P}(r)$ for P-P distance. The degree of dispersion is related to the molecular state. **d** The MSD (all elements, element P and Mn) *vs.* simulation time during the quenching process. **e** Schematic diagram of light transmission for transparent glassy and opaque polycrystalline scintillation screens (The enlarged part is a schematic diagram of molecular arrangement. The dashed color blocks in the right figure represent different crystal planes and grain boundaries.). In the transparent scintillation screen: the homogeneous disordered glassy state of the transparent medium allows emitted light to propagate straightly without scattering, resulting in the sharp image. In the case of polycrystalline scintillation screen: the light undergoes numerous refractions because it meets a great amount of crystal planes, resulting in the blurred luminescence that seriously impacts the image sharpness.

Based on above results, we obtain the molecular morphology of glassy state and crystalline state, which can explain why transparent scintillation screen produces clear imaging. In the transparent glassy scintillation screen, there is a homogeneous medium consisting of disordered molecules. So, light propagates through the glass straightly without scattering, thus ensuring full light transmission and the formation of clear image. Conversely, in polycrystalline scintillation screens, numerous microscopic grain boundaries cause light to scatter in random directions, ultimately resulting in image blurring (Fig. 4e). Therefore, a high transparent glassy scintillation screen presents great superiority for achieving high-resolution fast neutron imaging in real applications.

## High-resolution fast neutron imaging

The high transparency, excellent luminescence performance as well as good mechanical strength of $(BTPP)_{1.8}(HTPP)_{0.2}MnBr_4$ screen provide essential conditions for high quality FNR imaging. To further investigate its potential application, we conduct fast neutron detection experiments on a large-area $(BTPP)_{1.8}(HTPP)_{0.2}MnBr_4$ transparent scintillation screen (Fig. 5a). The tested object is placed between the fast neutron source and the $(BTPP)_{1.8}(HTPP)_{0.2}MnBr_4$ transparent scintillation screen. Incident neutrons are converted into visible light

emission by the scintillation screen. This optical signal is subsequently digitized using a CCD camera, facilitating fast neutron detection and imaging. The light output of scintillation screen is examined under fast neutron irradiation by analyzing the gray values of the images. The neutron flux in the experiment is about $10^7$ n cm$^{-2}$ s$^{-1}$ with an average energy of 2.69 MeV. A boron$^{10}$ ($B^{10}$) filter is utilized to remove thermal and epithermal neutrons with energy below 0.01 MeV (Supplementary Fig. 24). Four transparent scintillation screens show much higher light output up to three times than commercial ZnS (Ag): PP, which serves as a benchmark, and they also greatly outperform the opaque two-dimensional (2D) perovskite Mn-STA$_2$PbBr$_4$, which light output is only 79.05% of ZnS (Ag): PP[26] (Fig. 5b). The enhanced performance of $(BTPP)_{1.8}(HTPP)_{0.2}MnBr_4$ is primarily attributed to its amorphous structure induced high transparency. While ZnS (Ag): PP (an opaque mixture of polymer and powder) exhibits severe light scattering, resulting in a limited optical absorption length of 0.142 mm. As a result, only photons generated within the superficial 0.142 mm layer are effectively captured by the detector. In contrast, high transparency $(BTPP)_{1.8}(HTPP)_{0.2}MnBr_4$ has an optical absorption length of approximately 15.9 mm, which allows the majority of emitted photons to reach the detector[53] (Supplementary Note1 and Supplementary Fig. 25).

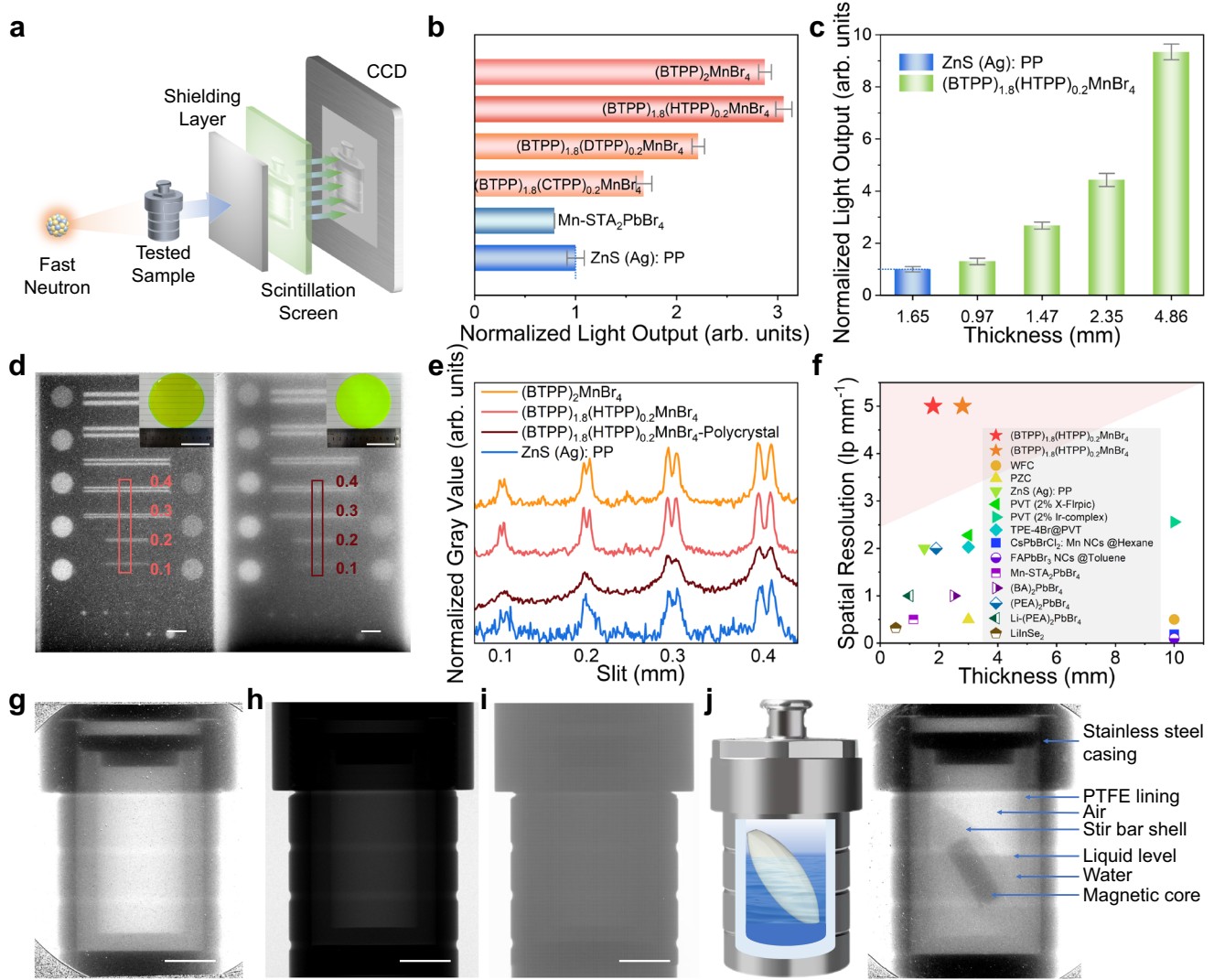

**Fig. 5 | Practical FNR imaging characterization of (BTPP)$_{1.8}$(HTPP)$_{0.2}$MnBr$_4$ transparent glassy screen. a** Schematic diagram of the experimental setup for FNR. Fast neutrons pass through the tested sample and the shielding layer, and are irradiated onto the scintillation screen to excite it luminescence, then receiving the image information by CCD. **b** The light output of different transparent scintillation screens, 2D Mn-STA$_2$PbBr$_4$ perovskite plate and ZnS (Ag): PP commercial fast neutron scintillation screen (ZnS (Ag): PP as a benchmark value of 1) (Error bars are presented as mean ± SD, n = 3 presents three independent experiments). **c** The light output of (BTPP)$_{1.8}$(HTPP)$_{0.2}$MnBr$_4$ transparent scintillation screens with different thicknesses (mean ± SD, n = 3). **d** Resolution test with standard sample (steel plate with different slits width). (BTPP)$_{1.8}$(HTPP)$_{0.2}$MnBr$_4$ transparent scintillation screen (left), and (BTPP)$_{1.8}$(HTPP)$_{0.2}$MnBr$_4$ opaque polycrystalline screen (right). Scale bars: 5 mm. The insets show the photos of each scintillation screen. Scale bars: 4 cm. **e** Curves of the relative gray value distribution of (BTPP)$_2$MnBr$_4$ and (BTPP)$_{1.8}$(HTPP)$_{0.2}$MnBr$_4$ transparent scintillation screen, (BTPP)$_{1.8}$(HTPP)$_{0.2}$MnBr$_4$

opaque polycrystalline screen and ZnS (Ag): PP commercial screen (extract from Fig. 5d and Supplementary Fig. 29), showing the much finer distinguishable slit of transparent screens (0.1 mm) over commercial ZnS (Ag): PP screen (0.3 mm). **f** Comparison of the thickness and spatial resolution of (BTPP)$_{1.8}$(HTPP)$_{0.2}$MnBr$_4$ transparent scintillation screen and several reported fast neutron scintillation screens[7,17,24,26,28,54–62] (The shadow indicates a decrease in resolution from high to low). **g** The hydrothermal reactor images by fast neutron imaging with (BTPP)$_{1.8}$(HTPP)$_{0.2}$MnBr$_4$ transparent screen, (**h**), thermal neutron imaging with ⁶Li ZnS: Cu of PSI and (**i**), high-energy X-ray industrial CT imaging at 280 kV. Scale bars: 2 cm. **j** Fast neutron imaging of a very complicated scenario that a thick stainless steel hydrothermal reactor with polymer lining containing liquid and a stir bar. Schematic diagram of the imaging object (left) and the fast neutron imaging (right) of the hydrothermal reactor showing the distinct hierarchical structures of every parts.

Furthermore, we utilize Geant4 simulations to model the physical interactions between neutrons and the scintillation screen, providing detailed energy data at each stage of the interaction process. Through analysis of the entire scintillation and energy transfer process, we identify the reasons behind the high light output of (BTPP)$_{1.8}$(HTPP)$_{0.2}$MnBr$_4$ (Supplementary Note 2 and Supplementary Table 3). The results reveal that the neutron energy deposition efficiency of (BTPP)$_{1.8}$(HTPP)$_{0.2}$MnBr$_4$ (1.116%) is lower than that of ZnS (Ag): PP (2.137%), primarily due to difference in hydrogen density. Moreover, a large number of recoil protons and a small amount of secondary γ-rays are observed in the simulated reaction channel

(Supplementary Figs. 26, 27 and Supplementary Discussion 1), but the production of β-rays is not detected. The recoil proton energy deposition efficiency (calculated relative to neutron deposition energy) is 77.35% for (BTPP)$_{1.8}$(HTPP)$_{0.2}$MnBr$_4$ and 84.05% for ZnS (Ag): PP, respectively. Consequently, in the neutron response process, the primary difference between (BTPP)$_{1.8}$(HTPP)$_{0.2}$MnBr$_4$ and ZnS (Ag): PP lies in neutron deposition which has a greater impact on the overall performance, whereas the effect of recoil protons is relatively small. It is noted that both (BTPP)$_{1.8}$(HTPP)$_{0.2}$MnBr$_4$ and ZnS (Ag): PP scintillators are modeled as homogeneous single component structures due to theoretical model limitations, representing idealized values. This

modeling does not account for the advantages of a single component over physical mixing. Notably, the PLQY of the $(BTPP)_{1.8}(HTPP)_{0.2}MnBr_4$ itself is 85.54%, approximately twice that of ZnS (Ag): PP (31.38%) (Supplementary Table 4). From a theoretical perspective, this higher PLQY offsets the impact of neutron deposition on light output. This suggests that the light output of $(BTPP)_{1.8}(HTPP)_{0.2}MnBr_4$ is determined by its favorable photon statistics, stemming from its higher transparency and more efficient photon transmission (Supplementary Table 4). Consequently, the light output of $(BTPP)_{1.8}(HTPP)_{0.2}MnBr_4$ surpasses that of ZnS (Ag): PP.

The increase in light output from $(BTPP)_{1.8}(HTPP)_{0.2}MnBr_4$ closely mirrors the increase in screen thickness when subjected to high-energy fast neutron irradiation (Fig. 5c). However, according to the imaging experience of traditional scintillation screens, an excessively thick screen can affect the imaging resolution[17,54]. In the following steps, we further investigate the actual performance of $(BTPP)_{1.8}(HTPP)_{0.2}MnBr_4$ in fast neutron imaging, using a standard steel slit plate (Supplementary Figs. 28–30) and modulation transfer function (MTF) (Supplementary Note 3 and Supplementary Figs. 31, 32) to test the resolution of the scintillation screen. The $(BTPP)_2MnBr_4$ and $(BTPP)_{1.8}(HTPP)_{0.2}MnBr_4$ transparent scintillation screen with same thickness of 1.8 mm can clearly reproduce the slits of testing standard resolution sample, while the images displayed on the $(BTPP)_{1.8}(HTPP)_{0.2}MnBr_4$ opaque polycrystalline screen and ZnS (Ag): PP commercial screen are very blurry and difficult to distinguish (Fig. 5d and Supplementary Fig. 29). The distribution curves of relative gray value of these scintillation screens (Fig. 5e) are extracted from Fig. 5d and Supplementary Fig. 29. It should be noted that in the double slit resolution test, the peak of the curve can only be considered distinguishable if it is divided into two parts. According to the smallest distinguishable width of 0.1 mm, the spatial resolution of the $(BTPP)_2MnBr_4$ and $(BTPP)_{1.8}(HTPP)_{0.2}MnBr_4$ transparent scintillation screen is 5 lp mm$^{-1}$, and the doping of HTPP$^+$ with longer branches does not affect its imaging performance. Compared to transparent scintillation screens, $(BTPP)_{1.8}(HTPP)_{0.2}MnBr_4$ opaque polycrystalline screen and ZnS (Ag): PP commercial screen have relatively poor resolution of only 0.3 mm slits visible, which is equivalent to 1.67 lp mm$^{-1}$ because of the serious light scattering problem (Fig. 5e). Even if the thickness of the transparent glass scintillator is increased to 2.8 mm, the resolution remains the same, along with the corresponding extracted relative gray values (5 lp mm$^{-1}$) (Supplementary Fig. 30). Then the resolution decreases with the increase of the thickness (from 2.8 mm to 3.8 mm). At a thickness of 3.8 mm, the MTF value drops to 2.1 lp mm$^{-1}$, corresponding to a spatial resolution of 2.8 lp mm$^{-1}$, but is still higher than that of ZnS (Ag): PP (Supplementary Fig. 32). A prominent advantage of transparent screen over opaque screen is that its spatial resolution is not significantly impacted by increasing screen thickness within 2.8 mm range, which means it can achieve high resolution image with brighter light output. The comparison of the thickness and resolution of fast neutron scintillation screens in recent years is also been provided (Fig. 5f and Supplementary Table 5). Satisfyingly, the transparent scintillation screen maintains high resolution significantly surpassing that of the commercial ZnS (Ag): PP as well as other reported fast neutron scintillation screen[7,17,24,26,28,54–62]. This further confirms that the transparent scintillation screen has considerable application potential in FNR.

Furthermore, we conduct practical heavy target experiments to evaluate the practical imaging capabilities of the fast neutron transparent scintillation screen. We choose a complex structural object with high-density materials and low-density materials inside, such as a hydrothermal reactor with a polytetrafluoroethylene (PTFE) lining, to demonstrate the imaging advantages of our transparent fast neutron scintillation screen (Fig. 5g and Supplementary Fig. 33). It is seen the $(BTPP)_{1.8}(HTPP)_{0.2}MnBr_4$ screen clearly reveals the distinct details of the reactor and internal PTFE lining (Fig. 5g). However, when thermal neutron radiography is performed with a commercial screen ($^6$Li ZnS: Cu of PSI) only faint cavity can be seen while the details are barely distinguishable (Fig. 5h). This limitation arises from the weak penetration of thermal neutrons due to the thick wall of the reactor. The imaging result of high-energy X-ray industrial computerized tomography (CT) at 280 kV is also unsatisfactory (Fig. 5i), making it difficult to distinguish internal details. Consequently, determining the internal structure of the hydrothermal reactor becomes unfeasible. These results demonstrate the distinctive advantages of FNR over other high-energy radiation methods. Furthermore, a more complicated scenario is arranged that a thick stainless steel hydrothermal reactor with PTFE lining loading liquid and a stir bar (the left side of Fig. 5j and Supplementary Fig. 34). This object contains liquid and different kinds of solids with various shapes and densities. The imaging result clearly reveals the hierarchical structures of every component part (the right side of Fig. 5j), compared to the imaging results of high-energy X-ray industrial CT at different energies (Supplementary Fig. 35). Even with more sophisticated CT three-dimensional (3D) reconstruction techniques, it is impossible to see the liquid and PTFE lining inside the reactor (Supplementary Fig. 36). This extraordinary penetrating power and spatial resolution capability open up the possibility for in-situ and non-destructive observation of internal phenomena within large-scale equipment.

Given its excellent neutron imaging performance, we further investigate the stability of the transparent scintillation screen. Based on the actual neutron test environment, we monitor the stability of the storage conditions (nitrogen at 25 °C) and fast neutron imaging conditions (air at 25 °C and 25% humidity) of $(BTPP)_{1.8}(HTPP)_{0.2}MnBr_4$, respectively (Supplementary Figs. 37, 38). The results show that the transparent scintillation screen retains 99.88% of its initial quantum efficiency after 90 days of storage in a nitrogen. However, in actual experiments, the scintillation screen is used for brief periods, allowing it to be stored in nitrogen and exposed only during use. Meanwhile, we also discuss the impact of humidity on transparency (Supplementary Figs. 39–42). $(BTPP)_{1.8}(HTPP)_{0.2}MnBr_4$ maintains its transparency for nearly 30 days at 75% humidity, but $(BTPP)_2MnBr_4$ gradually crystallizes after 10 days at 75% humidity. Furthermore, $(BTPP)_{1.8}(HTPP)_{0.2}MnBr_4$ also exhibits good fast neutron irradiation stability (Supplementary Fig. 43), after 10 days of continuous irradiation (fast neutron flux: $10^7$ n cm$^{-2}$ s$^{-1}$), the light output of $(BTPP)_{1.8}(HTPP)_{0.2}MnBr_4$ remains at 84.58% of the initial level, better than 80.24% of pure $(BTPP)_2MnBr_4$, indicating that the scintillation screen exhibits excellent irradiation stability, and further enhanced by HTPP$^+$ doping. Notably, even after becoming opaque or its performance declines, it can be restored to its original PLQY by reheating and melting, making it practical for long-term use (Supplementary Fig. 44). Overall, considering the operating and storage conditions, the stability of the transparent $(BTPP)_{1.8}(HTPP)_{0.2}MnBr_4$ scintillation screen can meet the standard requirements of fast neutron imaging application.

In summary, we demonstrate an alternative transparent fast neutron scintillation screen $(BTPP)_{1.8}(HTPP)_{0.2}MnBr_4$ through a one-step solvent-free melt quenching method, which can be processed to large-area screen with controllable shape and good toughness. Temperature-dependent XRD, PL, and AIMD calculations reveal the reversible conversion mechanism between opaque polycrystalline state and transparent glassy state. The formation mechanism of glassy state is attributed to the cooperation of high-steric-hindrance of triphenylphosphonium as well as fast quenching treatment that freezes of the random arrangement of molecules. The transparent scintillation screen achieves an impressive high spatial resolution of 5 lp mm$^{-1}$ and high light output (three times as much as commercial ZnS (Ag): PP). It shows effective distinguishability to heavy equipment with very complicated inner structures and compositions. This work has significantly enhanced the applicability of fast neutron imaging and

further advanced the development of non-destructive detection technology for large-scale equipment.

## Methods

### Materials

Manganese (II) bromide ($MnBr_2$ Aladdin, 98%), Butyltriphenylphosphonium bromide (BTPPBr, Macklin, 99%), Heptyltriphenylphosphonium bromide (HTPPBr, Aladdin, 98%), Dodecyltriphenylphosphonium bromide (DTPPBr, Bidepharm, 99%), Cetyltriphenylphosphonium bromide (CTPPBr, Alfa, 98%+), Hydrobromic acid (HBr, Macklin, 48%), ZnS (Ag): PP commercial fast neutron scintillator screen (1.65 mm, 1.8 mm thickness, RC TRITEC, Swiss). All materials were used without any purification.

### Synthesis of $(BTPP)_2MnBr_4$ transparent scintillation screen

The hybrid metal halide $(BTPP)_2MnBr_4$ transparent scintillation screen was synthesized using a simple and solvent-free low-temperature melt quenching method. All preparation operations were carried out in a nitrogen-filled glove box. Mix and grind $MnBr_2$ and BTPPBr in a molar ratio of 1: 2 and place them in a glass culture dish. Then, heat them at a rate of $10\,°C\,min^{-1}$ to $200\,°C$ on a heating table and hold for 10 min. Continue heating and stirring at a rate of $5\,°C\,min^{-1}$ to $280\,°C$ until all are melted. The melt in the culture dish appears yellow-green and transparent. It is poured into a silicon mold pre-heated to the same temperature as the melt. The silicone mold filled with melt is then removed from the heating stage and place on the tabletop for natural cooling to RT. This cooling process typically took 1–2 min. After demolding, obtain a transparent scintillation screen sample.

### Synthesis of other transparent scintillation screens

Considering the stability and optical properties of the materials, a certain ratio of long-chain organic cation $TPP^+$ ($HTPP^+/DTPP^+/CTPP^+$) was doped to control the toughness of the transparent scintillation screen ($(BTPP)_{1.8}(HTPP)_{0.2}MnBr_4/$ $(BTPP)_{1.8}(DTPP)_{0.2}MnBr_4/$ $(BTPP)_{1.8}(CTPP)_{0.2}MnBr_4$). All preparation operations were carried out in a nitrogen-filled glove box. The molar ratio of $MnBr_2$, BTPPBr, and TPPBr (HTPPBr/DTPPBr/CTPPBr) is 1: 1.8: 0.2. And then, mix and grind them and place them in a glass culture dish. Next, heat them at a rate of $10\,°C\,min^{-1}$ to $200\,°C$ on a heating table and hold them for 10 min. Continue heating and stirring at a rate of $5\,°C\,min^{-1}$ to $280\,°C$ until all are melted. The melt in the culture dish appears yellow-green and transparent. It is poured into a silicon mold pre-heated to the same temperature as the melt. The silicone mold filled with melt is then removed from the heating stage and placed on the tabletop for natural cooling to RT. This cooling process typically took 1–2 min. After demolding, obtain a transparent scintillation screen sample.

### Growth of $(BTPP)_2MnBr_4$ single crystal

$(BTPP)_2MnBr_4$ single crystal obtained by solvent evaporation method[31]. Specifically, $MnBr_2$ (0.2147 g, 1 mmol) and BTPPBr (0.7986 g, 2 mmol) were dissolved in 2 mL HBr solution, stirred and heated at $110\,°C$ for 90 min, then cooled at RT, and the $(BTPP)_2MnBr_4$ single crystal was obtained in transparent solution by slowly evaporating the solvent.

### Characterizations

PXRD patterns of samples were collected on a Bruker X-ray diffractometer (D8 focus, Cu Kα, λ = 0.15178 nm) operated. SCXRD was obtained by single crystal X-ray diffractometer (XtaLAB PRO 007HF(Mo)) (Rigaku, Japan). Use Instron 5966 for three-point bending test and obtain elastic modulus data. The test span is 20 mm, the speed is 1 mm $min^{-1}$, and there are three indenters totally (R = 2 mm) that two on bottom and one on top, respectively. The substrate material of the indenter is 3CR13, and the pressure roller is made of tungsten steel. The PL, PLE, temperature-dependent PL and the TRPL spectra were

obtained by a FLS1000 system (Edinburgh Instruments, UK). Absolute PLQYs were obtained using an integrating sphere connected by a FLS1000 system (Edinburgh Instruments, UK). UV-vis transmittance and absorption spectra of transparent and polycrystalline screens were obtained on a Cary 7000 spectrophotometer (Agilent, USA) equipped with an integrating sphere accessory. Raman spectra were obtained by the inVia-Qontor Raman microscope (Renishaw, England) with a laser (λ = 785 nm). Thermogravimetric analysis test (TG-DTA) and DSC was performed by NETZSCH STA 449 F5/F3 Jupiter (Netzsch, Germany) and NETZSCH DSC 3500 Sirius (Netzsch, Germany) in an $N_2$ atmosphere with a heating rate of $10\,°C\,min^{-1}$. Temperature-dependent XRD patterns of $(BTPP)_{1.8}(HTPP)_{0.2}MnBr_4$ was collected on a variable temperature XRD (Rigaku Smart Lab 9 kW) operated at a scan rate of $0.1\,°\,s^{-1}$. Use an Ocean Optics spectrometer and OM, POM to calibrate using a standard reflective aluminum mirror (Shanghai Idea optics Corp., Ltd.) as a reference. The structural transformation of the $(BTPP)_{1.8}(HTPP)_{0.2}MnBr_4$ were observed through variable temperature POM microscope (Olympus BX53, Japan). High-energy X-ray industrial CT imaging was obtained from CD-350BX/μCT microfocal industrial CT with a scanning resolution of 3 μm. The humidity stability experiment was conducted in the environmental testing chamber (HK WEWON TECHNOLOGY LIMITED).

### Fast neutron image detection and process

The fast neutron related tests were conducted at the China Institute of Atomic Energy (Beijing, China, Fission Neutron). The average energy of the fast neutron spectra is 2.69 MeV, produced by thermal neutron induced $U^{235}$ fission. The neutron flux of the experiment is about $10^7\,n\,cm^{-2}\,s^{-1}$, and a $B^{10}$ filter is used to remove thermal and epithermal neutrons with energy below 0.01 MeV. The collimation ratio L/D is set to 160. For all neutron imaging experiments in this study, a CCD camera (ANDOR iKon-936L) with pixels of 2048 * 2048 was used, and the camera was cooled to $-60\,°C$. In practical applications of fast neutron imaging, the operating temperature is maintained between 23–28 $°C$, the relative humidity is kept between 20% and 30% to prevent influence caused by excessive moisture and heat. We introduced a lead plate between the sample and the scintillation screen, to prevent interference from high-energy rays carried by neutrons. Set the exposure time to 40 s during shooting, and then take the value of 5 photos. And the image data is further processed using ImageJ software.

### Computational methods

In this study, we employed AIMD simulations with the CP2K package (open source) to explore the molecules and atoms evolution of $(BTPP)_2MnBr_4$ $((C_{22}H_{24}P)_2MnBr_4)$, during the processes of melting, quenching, and crystallizing under varying temperature conditions (elevated, lowered and fixed temperature)[48]. The generalized-gradient approximation in the parameterization of Perdew, Burke, and Ernzerhof (PBE)[63,64] was used to compute the exchange-correlation energy. Due to the dispersion interaction mainly dominate the melting, quenching, and crystallizing of molecular crystal, the long-range interaction is non-negligible, thereby, dispersion correction was considered in molecular dynamics simulation by using the empirical parameterized Grimme (D3) method[65]. The anti-ferromagnetic spin-spin coupling of Mn-3d electrons were applied in all calculation and such spin-spin coupling remain the same during AIMD simulation. The hybrid Gaussian and plane-wave (GPW) basis sets were employed to describe the valence-shell electrons of all elements, assigned as follows: H ($1s^1$), C ($2s^22p^2$), P ($3s^23p^3$), Br ($4s^24p^5$), and Mn ($3s^23p^63d^54s^2$). To reduce basis set superposition errors, specially optimized triple-ζ valence plus polarization (TZVP) basis sets were employed[66]. Auxiliary plane-wave basis sets were used with the cutoff energy of 600 Rydberg. Core electrons were treated using scalar relativistic norm-conserving pseudopotentials[67]. Brillouin zone integration was carried

out using a reciprocal space mesh consisting of only the gamma point. The simulations used the NVT for both melting and crystallizing processes, while the NVE for quenching process[49,50]. A time step of 1.0 fs was adopted, and each simulation was run for a total duration of up to 10 ps at the DFT level. The statistical sampling was conducted at different temperature of 300, 360 and 1600 K, respectively. The self-diffusion coefficients were computed from the MSDs using the Einstein relation by Eq. (1)[46]:

$$D = \frac{1}{6nt} \lim_{t \to \infty} \left\langle \sum_{i=1}^{n} [ri(t) - ri(0)]^2 \right\rangle \qquad (1)$$

The Lindemann ratio $\Delta$ is computed from the width of the first peaks in the different partial radial distribution functions as a measure of the fluctuation of atomic positions and interparticular distances by Eq. (2)[52]:

$$\Delta = \frac{\text{FWHM}}{d_0} \qquad (2)$$

### Calculation of fast neutron elastic scattering cross-section

The fast neutron reaction cross-sections were calculated and compared by combining the TENDL-2023 database with the energy spectra of the fast neutrons used in the experiment (average energy: 2.69 MeV).

### Fast neutron theory calculation methods

In order to investigate the physical processes of interaction, Geant4 software was used to perform theoretical calculations and simulations of the entire interaction process between fast neutrons and the sample. The physical model used for simulation is FTFP_SERT_HP. During these calculations, the number of neutrons emitted was $10^7$, and the average energy of incident neutrons was 2.69 MeV.

## Data availability

All data supporting the key findings of this study are available within the article and the Supplementary Information file, or from the corresponding author upon request. The X-ray crystallographic coordinates for structures reported in this study have been deposited at the Cambridge Crystallographic Data Centre (CCDC), under deposition numbers 2456170. These data can be obtained free of charge from The Cambridge Crystallographic Data Centre via www.ccdc.cam.ac.uk/data_request/cif. Crystallographic data are also provided as Supplementary Data.

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

## Acknowledgements

The authors acknowledge the National Natural Science Foundation of China (NSFC) (grant nos. 21975007 to N.W.), Joint Funds of the National Natural Science Foundation of China (U2241282 to L.F.H.), National Key Research and Development Program of China (2023YFA1609203 to L.F.H.) and Research Initiation Fund of Inner Mongolia University of Technology (No. BS2025005 to G.C.Y). We thank Dr. Yong Li (University of Bremen) and Professor Lei Kang and Dr. Yunfei Li (Technical Institute of Physics and Chemistry, CAS) for the helpful support and analysis of AIMD calculations. We thank Dr. Hongchao Yang and Professor Qibiao Wang (Sichuan University of Science & Engineering, SUSE) for the helpful support and analysis of simulation calculation of fast neutron physics processes.

## Author contributions

C.H.S., J.X.Z. and Z.A.Z. conceived the study and designed the experiments. Z.A.Z. and J.X.Z. performed most of the experiments and wrote the manuscript. Z.A.Z. and J.X.Z. draw the schematic diagram and mechanism diagram of the manuscript. Z.A.Z., T.F. and Y.N.A. performed a series of temperature-dependent tests. G.C.Y. and N.W. performed three-point bending test. Z.A.Z., J.X.Z., and G.C.Y. for the helpful AIMD calculations. S.H.R., M.M.W. and L.F.H. provided fast neutron sources and imaging systems and provided recommendations for fast neutron image detection and process. J.X.Z., S.Y.Z., L.F.H. and C.H.S. revised the paper. All authors discussed the results and commented on the manuscript.

## Competing interests

The authors declare no competing interests.

## Additional information

**Peer review information** *Nature Communications* thanks Cuong Dang, Hongwei Liang, Kyle McCall and the other anonymous reviewer(s) for

their contribution to the peer review of this work. A peer review file is available.

