## [Transparent Peer Review file · Nature Communications]

A transparent hybrid metal halide glassy scintillation screen for high-resolution fast neutron radiography

Corresponding Author: Professor Chenghua Sun

Version 0:

Reviewer comments:

Reviewer #1

(Remarks to the Author)

The manuscript "A transparent hybrid perovskite glassy scintillation screen for high-resolution fast neutron radiography" reports a transparent manganese halide system scintillator for fast neutron imaging. The most appealing part of this manuscript is the high-resolution fast neutron imaging achieved by utilizing the high transparency of manganese halides to reduce light scattering. However, transparent manganese halides are not novel and have been widely reported. Most of the content in the manuscript comprises some routine characterization data, which have also been reported in many previous papers. The fast neutron imaging part is relatively shallow and lacks sufficient investigation of the physical mechanisms. Therefore, I cannot recommend this manuscript for publication in a prestigious journal such as Nature Communications. Please refer to the following comments for your consideration:

1. (BTPP)1.8(HTPP)0.2MnBr₄ should not be called a "perovskite" because it does not possess the typical octahedral structure. It should be described as a metal halide with a perovskite-derived structure. I suggest the authors check the full text and change the designation "perovskite" accordingly.
2. The authors have devoted a great deal of space to investigating the properties of mixed A-site cations (Figures 2 to 4), such as structural, optical, thermal and mechanical properties, as well as transitions between crystalline and amorphous states. However, these properties have been reported extensively and in even greater detail in previous publications (10.1002/adom.202300216; 10.1002/adom.202102793; 10.1002/anie.202216504; 10.1002/adom.202302434). The authors just replaced a cation within this system, which does not seem particularly attractive for these properties.
3. The manuscript lacks a detailed explanation of how these mixed or single-cation species affect the scintillation performance of fast neutrons. For example, why does (BTPP)1.8(HTPP)0.2MnBr₄ exhibits a higher fast neutron light output response than other cations?
4. For fast neutron imaging, this manuscript appears to be a shallow, routine experiment, and does not provide an in-depth study of the physical process. For example, why does the fast neutron light output of this manganese halide system much higher than that of ZnS (Ag): PP? One of the reasons for ZnS (Ag): PP as a conventional fast neutron scintillation screen is that it exhibits high sensitivity to charged particles, and the polymer can provide a high hydrogen density. This manganese halide does not seem to provide a higher reaction cross section or hydrogen density than ZnS (Ag): PP, so does it show a higher response to charged particles?
5. The authors need to provide the energy spectra of the fast neutrons used in the experiment.
6. I suggest that the authors add a comparison of fast neutron reaction cross sections for manganese halide scintillators, ZnS (Ag): PP scintillators, or other commonly used scintillators. I suggest calculating or simulating the conversion efficiency of manganese halides for fast neutrons by combining the energy spectra of the fast neutrons used in the experiment.
7. The scintillator-fast neutron interaction is a complex multi-step process. I believe that the generated recoil protons' channel is the main contribution to the scintillation process. In addition to this, the interaction may be accompanied by the generation of gamma or beta rays, which would likewise contribute to the scintillation process. However, the manuscript also lacks the investigation of the energy of the resulting charged recoil protons and the process or depth of their interaction with manganese halide scintillators. The authors neglected to elaborate and analyze many important physical processes.
8. The question turns to stability. In Fig. 3a, it looks like the luminescent intensity of manganese halides decreases significantly at temperatures of a few tens of degrees. Does this mean they cannot work properly at slightly higher temperatures? Additionally, manganese halides are also highly susceptible to transitioning from the transparent amorphous state to the opaque crystalline state in the presence of moisture, making it difficult to maintain a consistently high resolution in fast neutron imaging. Irradiation stability data, especially under fast neutron irradiation, are likewise not included.

Reviewer #2

(Remarks to the Author)

Review for "A transparent hybrid perovskite glassy scintillation screen for high-resolution fast neutron radiography"

The work submitted by Zhou et al., "A transparent hybrid perovskite glassy scintillation screen for high-resolution fast neutron radiography" presents the development of a glassy scintillating screen based on Mn-based hybrid perovskite, for fast neutron imaging. This screen is fabricated using a solvent-free melt quenching method, resulting in a high-transparency, glassy material. The glassy scintillator exhibits up to 80% transmittance in the visible spectrum and 84% photoluminescence quantum yield. The transparency of the glassy screen leads to a significantly improved spatial resolution compared to the polycrystalline screen, with a spatial resolution of 5 lp/mm. The work is well presented, and we suggest publication with minor revisions, however, we would like the authors to answer the following queries below.

1- What was the rationale for the choice of triphenyl phosphonium (TPP) derivatives for cations? Is a very bulky cation necessary to obtain such a glassy state or would a smaller cation (e.g. phenylethylammonium or hexylammonium) also work? Would this be possible with other metallic cations (e.g. Pb^{2+} or Sn^{2+})?

2- The authors claim a spatial resolution of 5 lp/mm, however no modulation transfer function (MTF) is provided, which provides information regarding the contrast at a certain spatial frequency. In particular, Figure 5e shows that for 5 lp/mm (slit width of 0.1 mm) the contrast is quite low. The authors should measure and report the MTF vs. spatial frequency, as it's very important to understand the actual resolution capabilities of the screen quantitatively.

2- Regarding the fabrication of the glassy screen, was the operation performed in an oxygen and water-free atmosphere to avoid possible oxidation reactions between oxygen and the organic cation?

3- Is the glassy screen resistant to humidity? What is the material environmental stability? In addition, how stable is the material under irradiation? Have tests been performed to gauge the radiation hardness of the glassy TPP₂MnBr₄ screens?

3- There needs to be some further clarification on the experimental fabrication of the glassy screens. How was the quick quenching from melt temperature to 30 C done? The authors say the silicon moulds were pre-heated. Were the moulds pre-heated to the temperature of the melt? Was the quenching done by putting the silicon mould with the melt into a cooling liquid? How fast was the cooling? Please, elaborate further on these points.

4- The decay times of the scintillation of the TPP₂MnBr₄ material is quite long, in the order of hundreds of microseconds. Does this pose a risk for significant afterglow and thus possible after images that might affect the imaging process?

5- In the XRD data provided in Supplementary Fig. 4, the authors claim "polycrystal showed almost identical diffraction peaks compared to the simulation of (BTPP)₂MnBr₄ single crystal". While understandably a polycrystalline film peak intensity will differ from the theoretical prediction due to preferred orientation, I would suggest adding either the single crystal XRD diffractogram that was used in Supplementary Table 1 as well, and possibly the powder XRD diffractogram as well for comparison in Supplementary Fig. 4.

6- Would TPP₂MnBr₄ also perform well for X-ray imaging?

Reviewer #3

(Remarks to the Author)

This work by Zhou et al. describes the development of a candidate fast neutron scintillation screen based on alkyltriphenylphosphonium cations and MnBr₄ tetrahedral units, which can form glassy moldable screens with bright green emission. The mechanical, optical, and scintillation performance are reported, with good performance shown by this scintillator relative to established materials. The glassy nature of the screens is a significant advantage due to their transparency and moldability, and the use of mechanical property testing is a nice touch that isn't typically included in the other candidate FNR detector materials I've seen. My primary concerns are on the data reporting and manuscript clarity, especially regarding the spatial resolution. Once addressed, this manuscript would be a suitable advance for a Nature Communications publication.

Comments/questions for the authors:

1) Error bars and error values are missing from most metrics reported in this manuscript (e.g. PLQY, spatial resolution, PL lifetime, bond distances, light yield) – these must be included.

2) The spatial resolution must be calculated using the modular transfer function, as it is typically done in the radiography community. Fitting a Gaussian to one of the larger lines in the image of Figure 5d would be sufficient to enable a quantitative measure of spatial resolution. The data in Figure 5e do not clearly resolve the 0.1 slit as described by the authors, as the valley separating the two peaks is similar to the noise level of the image, and the peaks are not fully distinct. More likely, an

appropriate resolution value is the 0.3 mm slit, which would lead to a spatial resolution of 1.7 lp/mm. This is still an excellent and competitive value! Using the MTF will be able to provide more precision and reduce the error on this estimate.

3) This methodology should also be used to calculate the spatial resolution for all thicknesses of the detector – even transparent detectors offer reduced resolution as thickness increases, and this should be quantified.

4) All materials presented in this work are 0D hybrid halides, not perovskites. The perovskite structure requires octahedra with corner-shared connectivity – here all units are tetrahedral. It is appropriate to reference other perovskite structures such as the Mn-STA2PbBr₄ described in the introduction, but these Mn-based compounds are not perovskites.

5) The abstract and conclusion would benefit from more specific details (e.g. light output values vs. reference).

6) Larger Mn-Mn distances have been shown to improve PLQY (e.g. DOI: 10.1021/jacs.0c06039) in manganese halides, not decrease it as suggested – thus, the larger cation size must exert a different influence into the glassy state than the suggestion of the authors. This may also be tied to the decreased luminescence lifetime in the glassy state, and this connection deserves further consideration and investigation.

7) Are the AIMD-derived bond distances in Figure 4a meaningful and reproducible if the simulations are re-run? Given the breadth of the bond distributions in the following figures, it's not obvious to me that these values are truly reliable and comparable for the molten and glassy states.

8) Can the authors determine the hydrogen density of these screens based on the composition and density? It is important to calculate this parameter and compare with other candidate materials – it will enable the authors to also test if their detector has the needed stopping power for these high energy neutrons.

9) How was the gamma-ray sensitivity of these detectors evaluated? While the authors do use lead shielding in the beam path, it's not specified how efficiently this screens the gamma-ray background – the details of the beam characteristics with and without the shielding should be discussed. If the gamma-ray sensitivity is appreciable, this may contribute to the imaging performance observed here, especially for the spatial resolution testing.

10) The transmission data in Fig 2f is only above 80% from 500-600nm and 750-800nm, not from 400-800 as stated – the description should clarify this. A more accurate statement could be that the transmission is above 70% from 500-800nm.

11) The following phrase is unclear to me: “fast neutrons possess a unique advantage in isotope discrimination and radioactive material detection.....explosion-proof” – how are fast neutrons utilized in radioactive material detection? How do neutrons make or test explosion-proof materials? Do the authors mean fast neutron scintillation detectors or the neutrons themselves?

12) This statement is not correct and should be reworded for clarity: “The [MnBr₄]²⁻ serves as the luminescent center that efficiently transfers the recoil protons and excites them to emit visible light.”

13) The manuscript should be very carefully edited. A small selection of minor issues are listed for the author's convenience but there are myriad similar errors in the present manuscript:

a. Line 88 – presented should be replaced by captured or detected

b. Line 90 – omit the word seriously

c. Line 94 – The designing proposal of a should be replaced by “A prospective”

Version 1:

Reviewer comments:

Reviewer #1

(Remarks to the Author)

I am satisfied with the author's answer and recommend that this article be accepted.

Reviewer #4

(Remarks to the Author)

This study demonstrates an innovative application of hybrid halide glass scintillators in fast neutron radiography by suppressing light scattering through amorphous glass-state design. While the suppression of crystal boundary scattering using glassy materials has been well-established in X-ray imaging (as acknowledged in the response letter), the current work needs to better express its scientific breakthrough beyond technology transfer. So I don't think this work meet the criteria of Nature Communication. Also, the following issues require clarification:

1. The high imaging resolution stems from the intrinsic advantage of glassy-state materials. So are there other glassy materials used for neutron imaging? A direct comparison with previously reported amorphous scintillators would better position the innovation.

2. I don't see much improvement of the doping of HTPP+ while using for neutron imaging, as neither light output (Fig.5b) nor transparency (Fig.2f) shows measurable enhancement compared to undoped (BTPP)₂MnBr₄. The claimed benefits of HTPP+ doping require stronger experimental verification including stability tests of pure (BTPP)₂MnBr₄ under neutron irradiation and humidity.

Version 2:

Reviewer comments:

Reviewer #4

(Remarks to the Author)

The authors addressed most of my original comments, however, there are some other problems about the effect of the organic cation chain.

1. The output of fast neutron imaging performance of (BTPP)_{1.8}(HTPP)_{0.2}MnBr₄ glass depends on light output and photon transmission effect. With the increase of the length of the doped organic cation chain, the PLQY of DTPP⁺ and CTPP⁺ decreases very fast, while HTPP⁺ does not change, please explain the reason.

2. In Figure 2b, further demonstrate that the XRD of the slowly annealed polycrystalline sample i[(BTPP)₂MnBr₄] is consistent with the simulated SCXRD pattern.

3. Please explain in detail the reasons for the excellent stability of (BTPP)_{1.8}(HTPP)_{0.2}MnBr₄ glass in the high humidity environment. The author explains that it is attributed to the large steric hindrance of organic cations, so does the doping of DTPP⁺ and CTPP⁺ have the same or better effect?

Version 3:

Reviewer comments:

Reviewer #4

(Remarks to the Author)

The points raised in the review have been addressed by the authors. I would therefore recommend its publication in Nature Communications.

Thank you for the reviewers' insightful comments concerning our manuscript entitled "A transparent hybrid perovskite glassy scintillation screen for high-resolution fast neutron radiography" (NCOMMS-24-30110-T). The professional reviewers' comments are invaluable and very helpful for improving our paper, as well as the important guiding significance to our research. We have evaluated the comments carefully and made revisions according to these suggestions. Revised portions are highlighted in yellow in the paper. The main revisions in the paper and the responses to the reviewers' comments are as follows:

Reviewer #1:

The manuscript "A transparent hybrid perovskite glassy scintillation screen for high-resolution fast neutron radiography" reports a transparent manganese halide system scintillator for fast neutron imaging. The most appealing part of this manuscript is the high-resolution fast neutron imaging achieved by utilizing the high transparency of manganese halides to reduce light scattering. However, transparent manganese halides are not novel and have been widely reported. Most of the content in the manuscript comprises some routine characterization data, which have also been reported in many previous papers. The fast neutron imaging part is relatively shallow and lacks sufficient investigation of the physical mechanisms. Therefore, I cannot recommend this manuscript for publication in a prestigious journal such as Nature Communications. Please refer to the following comments for your consideration:

Response:

We greatly appreciate the reviewer's valuable feedback and have carefully considered each comment.

Regarding the novelty of transparent manganese halides and the routine nature of the characterization data: While manganese halides have been previously reported, our work is the first to explore their application in fast neutron radiography (FNR). The design of a transparent fast neutron scintillation screen must meet four criteria: (1) **high hydrogen density** to interact with fast neutrons and produce recoil protons, (2) **high photoluminescence quantum yield (PLQY)** to ensure bright luminescence, (3) **high transparency** to reduce light scattering, thereby enhancing light extracting efficiency and (4) **mechanical robustness** to support practical usability in FNR. Face to these criteria, our work reported a transparent scintillation screen (BTTP)_{1.8}(HTPP)_{0.2}MnBr₄

(BTPP⁺ = butyltriphenylphosphonium, HTPP⁺ = heptyltriphenylphosphonium) by using a one-step solvent-free melt quenching method. To achieve the target material, BTPP⁺ was selected as the main monovalent A⁺ cation due to its high steric hindrance, which facilitates the formation of a transparent medium while also provides interaction sites for fast neutrons. The [MnBr₄]²⁻ acts as the luminescent center, efficiently emitting visible light. Additionally, long-branched alkyl chain cations were incorporated to improve the mechanical strength of scintillation screen, enhance its hydrogen density, and enable reusability. Therefore, **Figure 2-4** in the manuscript mainly discuss the design, preparation, optical properties, transparency, mechanical strength, and formation mechanism of this fast neutron glassy scintillator. Based on these characterizations, the transparent hybrid metal halide (BTPP)_{1.8}(HTPP)_{0.2}MnBr₄ could be employed as a promising candidate for FNR.

As for physical mechanisms, a more comprehensive discussion of the fundamental physical processes was added, including the elastic scattering cross-section between fast neutrons and scintillation screen, neutron deposition efficiency within the scintillation screen, recoil proton deposition efficiency and its contribution to luminescence, etc. These revisions provide a more comprehensive understanding of the material's performance and potential in fast neutron imaging.

In summary, our work designed a transparent hybrid metal halide scintillator tailored for FNR applications. In addition, a detailed analysis of the physical mechanisms underlying the interaction between fast neutrons and scintillation screens has also been supplemented in the manuscript. These revisions provide a deeper understanding of the fast neutron scintillation process.

We have made significant revisions to the manuscript based on your insightful suggestions. Detailed responses to each comment can be found in the following sections of this document and in the revised manuscript.

Comment 1:

(BTPP)_{1.8}(HTPP)_{0.2}MnBr₄ should not be called a “perovskite” because it does not possess the typical octahedral structure. It should be described as a metal halide with a perovskite-derived structure. I suggest the authors check the full text and change the designation “perovskite” accordingly.

Response 1:

Thank you for the reviewer's valuable feedback regarding the classification of (BTPP)_{1.8}(HTPP)_{0.2}MnBr₄ as a “perovskite”. It is true that the material does not possess the

characteristic octahedral structure associated with conventional perovskites. As the structures in this work are tetrahedral, they are more accurately to be described as “perovskite-derived hybrid metal halides”. Thus, following reviewer’s suggestion, the manuscript has been revised to avoid any misleading use of the term “perovskite” and has consistently referred to the materials as “hybrid metal halides”.

In the revised manuscript, these changes have been implemented throughout the text.

Comment 2:

The authors have devoted a great deal of space to investigating the properties of mixed A-site cations (Figures 2 to 4), such as structural, optical, thermal and mechanical properties, as well as transitions between crystalline and amorphous states. However, these properties have been reported extensively and in even greater detail in previous publications (10.1002/adom.202300216; 10.1002/adom.202102793; 10.1002/anie.202216504; 10.1002/adom.202302434). The authors just replaced a cation within this system, which does not seem particularly attractive for these properties.

Response 2:

Thanks for the reviewer's comments. Regarding the publications mentioned by the reviewer: In work 1 (10.1002/adom.202300216), cations of varying sizes were selected to optimize the melt-quenching temperature, with Heptyl(triphenyl)phosphonium (HTP⁺) ultimately being chosen. This HTP⁺ cation enables the preparation of transparent glass integrated with TFT arrays, which is used to X-ray imaging applications. The work 2 (10.1002/adom.202102793) introduced (ETP)₂MnBr₄ (ethyltriphenylphosphonium (ETP⁺)), which has demonstrated a light yield of $\approx 35000 \pm 2000$ photon per MeV and a spatial resolution of 13.4 lp mm⁻¹ for X-ray imaging. The work 3 (10.1002/anie.202216504) explored the process of glass-crystal interconversion of (HTPP)₂MnBr₄ (hexyltriphenylphosphonium (HTPP⁺)) glassy metal halides, and emphasized the potential in X-ray imaging, which has exhibited a spatial resolution of 10 lp mm⁻¹. The work 4 (10.1002/adom.202302434) introduces [CH₃Ph₃P]₂MnX₄ glass can be used to produce curved screens for X-ray imaging. It is seen all these works are focused on X-ray imaging applications differing only in aspects such as the selection of cation types or variations in device structures, like integration with TFTs or curved surfaces. They are entirely unrelated to fast neutron radiography.

Our work focuses on designing a large-area transparent scintillation screen for FNR. The fast neutron response is closely related to hydrogen density, which can be modulated by adjusting the branched lengths of alkyl chains in the A-site cations. This introduces a trade-off between hydrogen density and PLQY allowing optimization for FNR applications. Our work leverages the inherent high transparency of metal halides in the glassy state to reduce light scattering and achieve high-resolution, high light output FNR. Moreover, the practical application requirements of FNR were addressed by introducing long-branched alkyl chain cations, which enhanced the mechanical toughness and hydrogen density of the scintillation screen.

Based on the characterizations (**Figure 2 to 4**) of structural, optical, thermal, and mechanical properties, the transparent metal halide $(\text{BTPP})_{1.8}(\text{HTPP})_{0.2}\text{MnBr}_4$ has been selected. The subtle effects of cationic modification on hydrogen density, light output, resolution, mechanical properties, and phase transitions, remain largely unexplored in the above-mentioned and other existing literatures.

Comment 3:

The manuscript lacks a detailed explanation of how these mixed or single-cation species affect the scintillation performance of fast neutrons. For example, why does $(\text{BTPP})_{1.8}(\text{HTPP})_{0.2}\text{MnBr}_4$ exhibits a higher fast neutron light output response than other cations?

Response 3:

We appreciate the reviewer's insightful comment. The scintillation performance for fast neutrons is influenced by several key factors: (1) hydrogen density, (2) interaction efficiency between fast neutrons and scintillation screen (neutron deposition efficiency, recoil proton deposition efficiency), (3) PLQY of material, (4) number of photon output from the scintillation screen reaching the detector (ACS Nano 2020, 14, 14686–14697).

To address the high hydrogen density requirement of fast neutron scintillation screens, the hydrogen density and elastic scattering cross-section of these mixed or single-cation metal halide scintillators were initially calculated, based on their composition and density (**Table R1**). The results demonstrated that an increase in the length of the branched alkyl chain leads to a higher hydrogen density, thereby enhancing the scattering cross-section and improving fast neutron absorption.

Furthermore, the PLQY of these mixed or single-cation metal halide scintillators was systematically tested (**Table R1**, **Figure R1**). As the length of the branched alkyl chain increased, a trade-off emerged that longer branched alkyl chains enhance hydrogen density but compromise PLQY (**Table R1**). Balancing these competing demands is crucial for improving the light output of fast neutron scintillation screens.

To achieve this goal, (BTTP)_{1.8}(HTTP)_{0.2}MnBr₄ was ultimately selected due to its balanced hydrogen density and PLQY.

We have incorporated the above discussion, elaborating on the role of cation species in influencing fluorescence performance (Page 5 and Page 7 in the revised manuscript, as well as Pages 2–3 in the revised Supplementary Information). And **Table R1** has been added to **Supplementary Table 1**, and **Figure R1** has been included as **Supplementary Fig. 1**.

Table R1 Density, hydrogen density, PLQY and fast neutron reaction cross-section of hybrid metal halide scintillators.

Compound*	Density (g cm ⁻³)	Hydrogen density (kg m ⁻³)	Average elastic scattering cross-section under fast neutron energy spectrum (cm ⁻¹)	PLQY
(BTTP) ₂ MnBr ₄	1.454	68.92	0.2698	85.56%
(HTTP) ₂ MnBr ₄	1.373	75.10	0.2817	64.08%
(DTTP) ₂ MnBr ₄	1.277	82.61	0.2963	3.830%
(CTTP) ₂ MnBr ₄	1.223	87.09	0.3052	2.720%
(BTTP)_{1.8}(HTTP)_{0.2}MnBr₄	1.440	69.40	0.2703	85.54%
(BTTP) _{1.8} (DTTP) _{0.2} MnBr ₄	1.423	70.39	0.2719	58.20%
(BTTP) _{1.8} (CTTP) _{0.2} MnBr ₄	1.407	70.99	0.2726	51.47%

* BTTP⁺ = C₂₂H₂₄P⁺; HTTP⁺ = C₂₅H₃₀P⁺; DTTP⁺ = C₃₀H₄₀P⁺; CTTP⁺ = C₃₄H₄₈P⁺.

Figure R1 The PLQY of the single-cation metal halide transparent scintillation screens.

Comment 4:

For fast neutron imaging, this manuscript appears to be a shallow, routine experiment, and does not provide an in-depth study of the physical process. For example, why dose the fast neutron light output of this manganese halide system much higher than that of ZnS (Ag): PP? One of the reasons for ZnS (Ag): PP as a conventional fast neutron scintillation screen is that it exhibits high sensitivity to charged particles, and the polymer can provide a high hydrogen density. This manganese halide does not seem to provide a higher reaction cross section or hydrogen density than ZnS (Ag): PP, so does it show a higher response to charged particles?

Response 4:

Thank you for the reviewer's helpful comment. As pointed out by the reviewer, the light output performance of the material is related to its hydrogen density and its interaction with charged particles. In addition, it is also closely related to the intrinsic fluorescence performance of the material itself and the number of photons ultimately emitted by the scintillation screen that reach the detector.

In order to further investigate the physical processes involved in these interactions, Geant4 software was used to perform theoretical calculations and simulations of the entire interaction process

between fast neutrons and the sample. The simulations provided detailed energy data of each stage of the interaction process (**Table R2**). During these calculations, the number of neutrons emitted was 10^7 , and the average energy of incident neutrons was 2.69 MeV. The physical model used for simulation is FTFP_BERT_HP. It can be seen that the hydrogen density of transparent scintillation screens (BTTP)_{1.8}(HTTP)_{0.2}MnBr₄ (69.40 kg m^{-3}) is lower than that of ZnS (Ag): PP (136.9 kg m^{-3}). This difference in hydrogen density contributes to the smaller elastic scattering cross-section of (BTTP)_{1.8}(HTTP)_{0.2}MnBr₄ (0.2703 cm^{-1}) compared to ZnS (Ag): PP (0.4556 cm^{-1}). Consequently, the neutron deposition energy of ZnS (Ag): PP ($5.748 \times 10^5 \text{ MeV}$) is approximately twice that of (BTTP)_{1.8}(HTTP)_{0.2}MnBr₄ ($3.003 \times 10^5 \text{ MeV}$). This is because hydrogen density affects the elastic scattering cross-section, which in turn affects the deposition of fast neutrons in the scintillation screen. According to **Table R2** and **Equation (1)**, the neutron energy deposition efficiency of (BTTP)_{1.8}(HTTP)_{0.2}MnBr₄ (1.116%) is lower than that of ZnS (Ag): PP (2.137%), primarily due to difference in hydrogen density.

Furthermore, the recoil proton deposition energy was calculated to be $2.323 \times 10^5 \text{ MeV}$ for (BTTP)_{1.8}(HTTP)_{0.2}MnBr₄ and $4.831 \times 10^5 \text{ MeV}$ for ZnS (Ag): PP, respectively. According to **Table R2** and **Equation (2)**, the recoil proton energy deposition efficiency for (BTTP)_{1.8}(HTTP)_{0.2}MnBr₄ and ZnS (Ag): PP (calculated relative to neutron deposition energy) is 77.35% and 84.05%, respectively, which shows that the responses of the two scintillation screens to charged particles are not significantly different. (Due to current theoretical model limitations, both (BTTP)_{1.8}(HTTP)_{0.2}MnBr₄ and ZnS (Ag): PP scintillators are modeled as homogeneous single component structures representing idealized values. This modeling does not account for the advantages of a single component over physical mixing.) Therefore, in the neutron response process, the primary difference between (BTTP)_{1.8}(HTTP)_{0.2}MnBr₄ and ZnS (Ag): PP lies in neutron deposition which has a greater impact on the overall performance, while the effect of recoil protons is relatively small.

Notably, the PLQY of the (BTTP)_{1.8}(HTTP)_{0.2}MnBr₄ is 85.54%, approximately twice that of ZnS (Ag): PP (31.38%) (**Table R2**). From a theoretical perspective, this difference in PLQY offsets the impact of neutron deposition on light output.

Moreover, the grayscale values were extracted from the detection results (**Table R3**). Under the experimental condition of a neutron flux of $10^7 \text{ n cm}^{-2} \text{ s}^{-1}$, an imaging size of $50 \text{ } \mu\text{m pix}^{-1}$ and an

exposure time of 40 s, the number of neutrons reaching the sample was calculated as 10000 n pix^{-1} . Using theoretical calculations of the number of neutrons generating recoil protons (**Table R3**), the conversion probabilities of recoil protons were calculated using **Equation (3)** to be 1.278% for $(\text{BTPP})_{1.8}(\text{HTPP})_{0.2}\text{MnBr}_4$ and 2.505% for ZnS (Ag): PP, respectively. Therefore, under the same conditions, the number of neutrons that generate recoil protons is 127.8 n pix^{-1} in $(\text{BTPP})_{1.8}(\text{HTPP})_{0.2}\text{MnBr}_4$ and 250.5 n pix^{-1} in ZnS (Ag): PP. Despite the lower neutron interaction and recoil proton generation probabilities for $(\text{BTPP})_{1.8}(\text{HTPP})_{0.2}\text{MnBr}_4$, its detected grayscale value is approximately 208, which is nearly three times that of ZnS (Ag): PP (grayscale value: 70). This indicates that the light output from $(\text{BTPP})_{1.8}(\text{HTPP})_{0.2}\text{MnBr}_4$ is determined by its superior photon statistics, stemming from its higher transparency and efficient photon transmission.

The superior performance of $(\text{BTPP})_{1.8}(\text{HTPP})_{0.2}\text{MnBr}_4$ is primarily attributed to its high transparency. ZnS (Ag): PP is an opaque mixture of polymer and powder components, which exhibits severe light scattering, resulting in a limited optical absorption length of 0.142 mm (**Figure R2**). As a result, only photons generated within the superficial 0.142 mm layer are effectively captured by the detector. In contrast, high transparency $(\text{BTPP})_{1.8}(\text{HTPP})_{0.2}\text{MnBr}_4$ has an optical absorption length of approximately 15.9 mm, which allows the majority of emitted photons to reach the detector.

Therefore, despite $(\text{BTPP})_{1.8}(\text{HTPP})_{0.2}\text{MnBr}_4$ has low hydrogen density and neutron reaction probability compared to ZnS (Ag): PP, its higher PLQY and transparency significantly reduce scattering loss, enabling more photons reach the detector. Consequently, the fast neutron light output of $(\text{BTPP})_{1.8}(\text{HTPP})_{0.2}\text{MnBr}_4$ surpasses that of ZnS (Ag): PP.

We have added the **Table R2** into **Supplementary Table 3**, and **Table R3** and **Figure R2** as **Supplementary Table 4** and **Supplementary Fig. 21**, respectively. And, we have included the explanation regarding for the excellent light output of $(\text{BTPP})_{1.8}(\text{HTPP})_{0.2}\text{MnBr}_4$ in the revised manuscript (Page 17-18).

Table R2 Key parameters and the summary of the physical process of the interaction between fast neutrons and (BTTP)_{1.8}(HTTP)_{0.2}MnBr₄, ZnS (Ag): PP.

Parameters	(BTTP) _{1.8} (HTTP) _{0.2} MnBr ₄	ZnS (Ag): PP
PLQY	85.54%	31.38%
Density (g cm ⁻³)	1.440	1.900
Hydrogen density (kg m ⁻³)	69.40	136.9
Average elastic scattering cross-section under fast neutron energy spectrum (cm ⁻¹)	0.2703	0.4556
Total energy of fast neutron incidence (E_0 , MeV)	2.69×10^7	2.69×10^7
Neutron deposition energy (E_n , MeV)	3.003×10^5	5.748×10^5
Neutron energy deposition efficiency (δ)	1.116%	2.137%
Generated recoil proton energy (E_{r1} , MeV)	2.688×10^5	5.368×10^5
Recoil proton deposition energy (E_{r2} , MeV)	2.323×10^5	4.831×10^5
Recoil proton energy deposition efficiency (σ)	77.35%	84.05%

Table R3 The conversion probability of recoil protons and the grayscale value related data of (BTTP)_{1.8}(HTTP)_{0.2}MnBr₄ and ZnS (Ag): PP.

Parameters	(BTTP) _{1.8} (HTTP) _{0.2} MnBr ₄	ZnS (Ag): PP
Number of incident fast neutrons (N_0)	1×10^7	1×10^7
Number of neutrons that generate recoil protons (N)	1.278×10^5	2.505×10^5
The conversion probability of recoil protons (ϵ)	1.278%	2.505%
Number of neutrons reaching the sample* ¹	10000	10000
Number of neutrons that generate recoil protons* ²	127.8	250.5
Measured grayscale value	208 ± 5	70 ± 3
PLQY	85.54%	31.38%

*¹ Under the detection condition of a neutron flux of 10^7 n cm⁻² s⁻¹, an imaging size of 50 μ m pix⁻¹ and an exposure time of 40 s, the number of neutrons reaching the sample was calculated to be 10000 n pix⁻¹: 10^7 n cm⁻² s⁻¹ \times (50×10^{-4} cm pix⁻¹)² \times 40 s = 10000 n pix⁻¹.

*² Number of neutrons reaching the sample is 10000 n pix⁻¹. The conversion probability of recoil protons (ϵ) of (BTTP)_{1.8}(HTTP)_{0.2}MnBr₄ and ZnS (Ag): PP is 1.278% and 2.505%, respectively. The number of neutrons that generate recoil protons was calculated to be 127.8 and 250.5, respectively: $10000 \times 1.278\% = 127.8$; $10000 \times 2.505\%$

= 250.5.

The neutron energy deposition efficiency is defined by **Equation (1)**,

$$\delta = \frac{E_n}{E_0} \quad (1)$$

where E_0 is the total energy of fast neutron incidence and E_n is the neutron deposition energy.

The recoil proton energy deposition efficiency is defined by **Equation (2)**,

$$\sigma = \frac{E_{r2}}{E_n} \quad (2)$$

where E_n is the neutron deposition energy and E_{r2} is the recoil proton deposition energy.

The conversion probability of recoil protons is defined by **Equation (3)**,

$$\varepsilon = \frac{N_0}{N} \quad (3)$$

where N_0 is the number of incident fast neutrons and N is the number of neutrons that generate recoil protons (there are cases where one neutron generates multiple recoil protons).

The absorption length was determined from the attenuation of absorption, following the standard Beer–Lambert, while accounting for the influence of reflection on the absorption process (Nucl. Instrum. Methods Phys. Res., A 2019, 940, 393–404), as **Equation (4)**:

$$\lambda_L = \frac{L / \ln 10}{A + 2 \log(1 - R(\lambda))} \quad (4)$$

where: λ_L : absorption length; L : screen thickness; A : attenuation; $R(\lambda)$: reflectivity, calculated by Fresnel equation.

The attenuation, A , is defined as **Equation (5)**:

$$A = -\log\left(\frac{I}{I_0}\right) \quad (5)$$

where: I : the intensity of light passing through the sample; I_0 : the intensity of the reference beam.

The transmittance of scintillation screens with varying thicknesses was measured using a Cary 7000 spectrophotometer to calculate the attenuation corresponding to each thickness.

The Fresnel equation calculates reflectivity as **Equation (6)**:

$$R(\lambda) = \left(\frac{n_1(\lambda) - n_2(\lambda)}{n_1(\lambda) + n_2(\lambda)}\right)^2 \quad (6)$$

where: $n_1(\lambda)$ is the refractive index of air, the refractive index of air is approximately 1. $n_2(\lambda)$ is the refractive index of the screen. $n_2(\lambda)$ is 1.660 for (BTPP)_{1.8}(HTPP)_{0.2}MnBr₄ and 1.548 for ZnS (Ag): PP.)

Therefore, a linear equation related to the length of light absorption is obtained as **Equation (7)**:

$$L = \ln 10 \lambda_L A + 2\lambda_L \ln(1 - R(\lambda)) \quad (7)$$

And linear fitting using attenuation A and sample thickness as the x- and y- coordinates were performed, respectively. The slope of the linear equation is obtained that is 36.85 for $(\text{BTTP})_{1.8}(\text{HTPP})_{0.2}\text{MnBr}_4$ and 0.3274 for ZnS (Ag): PP . By calculating the slope of the linear equation, the fitted light absorption length value can be obtained. The light absorption length is $\lambda_L \approx 15.9$ mm for $(\text{BTTP})_{1.8}(\text{HTPP})_{0.2}\text{MnBr}_4$ and $\lambda_L \approx 0.142$ mm for ZnS (Ag): PP .

Figure R2 Linear fitting diagrams for attenuation and sample thickness: (a) $(\text{BTTP})_{1.8}(\text{HTPP})_{0.2}\text{MnBr}_4$ and ZnS (Ag): PP ; (b) Magnified view of ZnS (Ag): PP .

Comment 5:

The authors need to provide the energy spectra of the fast neutrons used in the experiment.

Response 5:

Thanks for referee's kind advice. The energy spectra of the fast neutron used in the experiments are extracted from the reactor (**Figure R3**). The average energy of the fast neutron spectra is 2.69 MeV, produced by thermal neutron induced U^{235} fission. To ensure the purity of the fast neutron beam, a boron¹⁰ (B^{10}) filter was employed to effectively remove thermal and epithermal neutrons with energy below 0.01 MeV.

In the revised manuscript, we have included **Figure R3** as **Supplementary Fig. 20**, along with its corresponding description (Page 13 of the Supplementary Information and Page 27 of the revised manuscript).

Figure R3 The energy spectra of the fast neutrons.

Comment 6:

I suggest that the authors add a comparison of fast neutron reaction cross sections for manganese halide scintillators, ZnS (Ag): PP scintillators, or other commonly used scintillators. I suggest calculating or simulating the conversion efficiency of manganese halides for fast neutrons by combining the energy spectra of the fast neutrons used in the experiment.

Response 6:

Thank you for the reviewer's careful suggestions. This point has been addressed by combining the TENDL-2023 database with the energy spectra of the fast neutrons used in the experiment (average energy: 2.69 MeV) to calculate and compare the fast neutron reaction cross-sections of manganese halide scintillators, ZnS (Ag): PP scintillators, and other commonly used scintillators (**Table R4**). The results show that the elastic scattering cross-section of (BTPP)_{1.8}(HTPP)_{0.2}MnBr₄ is 0.2703 cm⁻¹, which is slightly lower than that of ZnS (Ag): PP (0.4556 cm⁻¹), while the cross-section of stilbene (0.2690 cm⁻¹) and anthracene (0.3100 cm⁻¹) are comparable to that of (BTPP)_{1.8}(HTPP)_{0.2}MnBr₄. In the series of transparent hybrid metal halide scintillators, it is observed that as the length of the branched alkyl chains of cations increases, the scattering cross-section gradually increases. Meanwhile, using the energy spectra of the fast neutrons (**Figure R3**), the neutron deposition energy

of (BTTP)_{1.8}(HTTP)_{0.2}MnBr₄ and ZnS (Ag): PP were simulated and calculated using Geant4, yielding values of 5.748×10^5 MeV and 3.003×10^5 MeV, respectively. According to **Table R2** and **Equation (1)**, the energy conversion efficiency (neutron energy deposition efficiency) for fast neutrons were determined to be 1.116% for (BTTP)_{1.8}(HTTP)_{0.2}MnBr₄ and 2.137% for ZnS (Ag): PP, respectively (**Table R2**).

Although the fast neutron reaction cross-section and energy conversion efficiency of (BTTP)_{1.8}(HTTP)_{0.2}MnBr₄ are lower than those of ZnS (Ag): PP, the light output of fast neutron scintillator depends on more than just these two factors. Crucially, the PLQY of the material and the number of photons that the scintillation screen can ultimately emit to reach the detector also play significant roles (ACS Nano 2020, 14, 14686-14697). In this regard, (1) the PLQY of the (BTTP)_{1.8}(HTTP)_{0.2}MnBr₄ is 85.54%, approximately twice that of ZnS (Ag): PP (31.38%) (**Table R4**), effectively compensating for the lower fast neutron deposition efficiency in terms of light output; (2) under identical conditions, the number of neutrons generating recoil protons is 150.1 n pix^{-1} for (BTTP)_{1.8}(HTTP)_{0.2}MnBr₄ and 294.3 n pix^{-1} for ZnS (Ag): PP (**Table R3**). However, the grayscale value detected by (BTTP)_{1.8}(HTTP)_{0.2}MnBr₄ is approximately 208, which is nearly three times higher than that of ZnS (Ag): PP (grayscale value: 70). These results indicate that the high transparency of (BTTP)_{1.8}(HTTP)_{0.2}MnBr₄ reduces the light scattering losses, allowing more photons reach the detector, resulting in a higher light output under the same thickness.

Therefore, although the fast neutron reaction cross-section and energy conversion efficiency of (BTTP)_{1.8}(HTTP)_{0.2}MnBr₄ are lower than those of ZnS (Ag): PP, the comprehensive performance comparison demonstrates that (BTTP)_{1.8}(HTTP)_{0.2}MnBr₄ achieves higher light output under fast neutron irradiation due to its superior PLQY and optical transparency.

In the revised manuscript, the discussions regarding the hydrogen density, elastic scattering cross-section, and fast neutron stopping power of the transparent scintillation screens, as well as other commonly used scintillators, have been added (Page 5).

Table R4 Density, hydrogen density, PLQY and fast neutron reaction cross-section of hybrid metal halide scintillators, ZnS (Ag): PP scintillators, and other commonly used scintillators.

Compound*	Density (g cm ⁻³)	Hydrogen density (kg m ⁻³)	Average elastic scattering cross-section under fast neutron energy spectrum (cm ⁻¹)	PLQY
(BTPP) ₂ MnBr ₄	1.454	68.92	0.2698	85.56%
(HTPP) ₂ MnBr ₄	1.373	75.10	0.2817	64.08%
(DTPP) ₂ MnBr ₄	1.277	82.61	0.2963	3.830%
(CTPP) ₂ MnBr ₄	1.223	87.09	0.3052	2.720%
(BTPP) _{1.8} (HTPP) _{0.2} MnBr ₄	1.440	69.40	0.2703	85.54%
(BTPP) _{1.8} (DTPP) _{0.2} MnBr ₄	1.423	70.39	0.2719	58.20%
(BTPP) _{1.8} (CTPP) _{0.2} MnBr ₄	1.407	70.99	0.2726	51.47%
ZnS (Ag): PP	1.900	136.9	0.4556	31.38%
Stilbene	1.010	67.33	0.2690	65.00%
Anthracene	1.280	71.91	0.3100	64.00%

* BTPP⁺ = C₂₂H₂₄P⁺; HTPP⁺ = C₂₅H₃₀P⁺; DTPP⁺ = C₃₀H₄₀P⁺; CTPP⁺ = C₃₄H₄₈P⁺; PP = (C₅H₆)_n; stilbene = C₁₄H₁₂; anthracene = C₁₄H₁₀.

Comment 7:

The scintillator-fast neutron interaction is a complex multi-step process. I believe that the generated recoil protons' channel is the main contribution to the scintillation process. In addition to this, the interaction may be accompanied by the generation of gamma or beta rays, which would likewise contribute to the scintillation process. However, the manuscript also lacks the investigation of the energy of the resulting charged recoil protons and the process or depth of their interaction with manganese halide scintillators. The authors neglected to elaborate and analyze many important physical processes.

Response 7:

Thank you for your insightful suggestions. Geant4 software was used to simulate the interaction process between fast neutrons and scintillators, collecting detailed information on secondary particles generated by these interactions. During the simulation process, it confirms that the energy

deposited by fast neutrons on the scintillation screens is primarily due to recoil protons. The recoil proton energy spectra of different scintillation screens are shown in **Figure R4**. Meanwhile, through simulation, it was found that in addition to recoil protons, secondary γ -ray was also generated through neutron inelastic scattering and neutron capture without secondary electron generation, as shown in the energy spectrum (**Figure R5**). In addition, the generation of β -rays was not observed during the simulation of the reaction channel process.

Furthermore, the energy and conversion efficiency of charged particles generated during the interaction process were calculated (**Table R5**). The parameters used in these calculations include: the number of neutrons emitted was 10^7 , and the average energy of incident neutrons was 2.69 MeV. The recoil proton deposition energy is calculated to be 2.323×10^5 MeV for (BTTPP)_{1.8}(HTPP)_{0.2}MnBr₄ and 4.831×10^5 MeV for ZnS (Ag): PP, respectively. According to **Table R5** and **Equation (2)**, the recoil proton energy deposition efficiency is 77.35% and 84.05%, respectively. Then, according to **Table R5** and **Equation (4)**, the secondary γ -ray energy deposition efficiency is 0.07640% and 0.05167%, respectively. (Due to current theoretical model limitations, both (BTTPP)_{1.8}(HTPP)_{0.2}MnBr₄ and ZnS (Ag): PP scintillators are modeled as homogeneous single component structures representing idealized values. This modeling does not account for the advantages of a single component over physical mixing.)

In short, the generated recoil protons' channel is the main contribution to the scintillation process. The energy deposited by secondary γ -ray is significantly lower than that of recoil protons, and has a relatively small impact on the scintillation process. Additionally, no secondary β -rays was observed, further highlighting the dominant role of recoil protons.

In the revised manuscript, details regarding the physical interactions between neutrons and the scintillation screen have been updated (Page 16 of the revised manuscript and Pages 15–17 of the revised Supplementary Information). Additionally, in the revised Supplementary Information, **Figure R4–5** and the complete **Table R5** have been incorporated as **Supplementary Fig. 22–23** and **Supplementary Table 3** (Pages 15–17).

Figure R4 Recoil proton energy spectrum of $(\text{BTTP})_{1.8}(\text{HTTP})_{0.2}\text{MnBr}_4$ and $\text{ZnS}(\text{Ag})\text{:PP}$.

Figure R5 Secondary γ -ray spectrum of $(\text{BTTP})_{1.8}(\text{HTTP})_{0.2}\text{MnBr}_4$ and $\text{ZnS}(\text{Ag})\text{:PP}$.

Table R5 Summary of the physical process of the interaction between fast neutrons and $(\text{BTTP})_{1.8}(\text{HTTP})_{0.2}\text{MnBr}_4$, $\text{ZnS}(\text{Ag})\text{:PP}$.

Parameters	$(\text{BTTP})_{1.8}(\text{HTTP})_{0.2}\text{MnBr}_4$	$\text{ZnS}(\text{Ag})\text{:PP}$
Total energy of fast neutron incidence (E_0 , MeV)	2.69×10^7	2.69×10^7
Neutron deposition energy (E_n , MeV)	3.003×10^5	5.748×10^5
Generated recoil proton energy (E_{r1} , MeV)	2.688×10^5	5.368×10^5
Recoil proton deposition energy (E_{r2} , MeV)	2.323×10^5	4.831×10^5
Recoil proton energy deposition efficiency (σ)	77.35%	84.05%
Generated secondary γ -ray energy (E_{g1} , MeV)	1.279×10^5	1.688×10^5
Secondary γ -ray deposition energy (E_{g2} , MeV)	229.4	297.0
Secondary γ -ray energy deposition efficiency (ρ)	0.07640%	0.05167%

The secondary γ -ray energy deposition efficiency is defined by **Equation (8)**:

$$\rho = \frac{E_{g2}}{E_n} \quad (8)$$

where E_n is the neutron deposition energy and E_{g2} is the secondary γ -ray deposition energy.

Comment 8:

The question turns to stability. In Fig. 3a, it looks like the luminescent intensity of manganese halides decreases significantly at temperatures of a few tens of degrees. Does this mean they cannot work properly at slightly higher temperatures? Additionally, manganese halides are also highly susceptible to transitioning from the transparent amorphous state to the opaque crystalline state in the presence of moisture, making it difficult to maintain a consistently high resolution in fast neutron imaging. Irradiation stability data, especially under fast neutron irradiation, are likewise not included.

Response 8:

Thanks to the careful reviewer. In practical applications of fast neutron imaging the operating temperature is maintained between 23-28°C, while the relative humidity is kept between 20% and 30% to prevent influence caused by excessive moisture and heat.

To address the reviewer's concerns, the environmental storage stability of the (BTTP)_{1.8}(HTTP)_{0.2}MnBr₄ scintillation screen was monitored under both storage conditions (at 25°C nitrogen) and fast neutron imaging conditions (at 25°C and 25% humidity air) (**Figure R6**). After 90 days of storage in nitrogen, the PLQY of (BTTP)_{1.8}(HTTP)_{0.2}MnBr₄ remained at approximately 99.88% of its initial value (**Figure R6a**). In addition, the PLQY decreased to 57.17% of the initial value after 40 days under fast neutron imaging conditions (at 25°C and 25% humidity air) (**Figure R6b**). However, in actual experiments, the scintillation screen is commonly used in short time (up to two hours), allowing it to be stored in nitrogen and exposed in air only during use. Meanwhile, due to the limited operating temperature of the FNR (23-28°C), the effect of elevated temperature on storage stability was evaluated. When stored in nitrogen at 40°C, the PLQY of (BTTP)_{1.8}(HTTP)_{0.2}MnBr₄ decreased to 96.31% of the initial value after 90 days (**Figure R7**), indicating minimal impact of higher temperatures during storage.

To assess the effect of humidity, the transparency retention was tested at 25%, 50% and 75% humidity at 25°C. As shown in **Figure R8**, the $(\text{BTPP})_{1.8}(\text{HTPP})_{0.2}\text{MnBr}_4$ remains transparent at 25% and 50% humidity after 30 days, but becomes opaque at 75% humidity. The powder X-ray diffraction (PXRD) also revealed crystallization peaks under high humidity, indicating a transition from a transparent amorphous state to an opaque crystalline state (**Figure R9**). Nevertheless, it can be restored to its original PLQY by reheating and melting, making it practically reusable (**Figure R10**).

Furthermore, the stability of the $(\text{BTPP})_{1.8}(\text{HTPP})_{0.2}\text{MnBr}_4$ scintillation screen under continuous fast neutron irradiation was assessed. As can be seen from **Figure R11**, the light output intensity remains at around 98.3% of its initial value after 120 minutes of continuous irradiation, indicating good irradiation stability.

Overall, considering the operating and storage conditions, the stability of the transparent $(\text{BTPP})_{1.8}(\text{HTPP})_{0.2}\text{MnBr}_4$ scintillation screen can meet the standard test condition of fast neutron imaging application.

In response to these opinions, we have included the revised Supplementary Information (**Supplementary Fig. 33-38**, Page 25-27) and relevant descriptions in the revised manuscript (Page 19-20).

Figure R6 The stability of the PLQY of $(\text{BTPP})_{1.8}(\text{HTPP})_{0.2}\text{MnBr}_4$ scintillation screen under both storage conditions (at 25°C nitrogen) (a) and fast neutron imaging conditions (at 25°C and 25% humidity air) (b).

Figure R7 The stability of the PLQY of $(\text{BTPP})_{1.8}(\text{HTPP})_{0.2}\text{MnBr}_4$ stored at slightly higher temperature (at 40°C nitrogen).

Figure R8 Photos of changes in $(\text{BTPP})_{1.8}(\text{HTPP})_{0.2}\text{MnBr}_4$ under different humidity environments for 30 days (up: visible light; bottom: 365 nm UV light).

Figure R9 The initial PXR and the PXR after being stored at 25%, 50% and 75% humidity levels at the same temperature for 30 days of $(\text{BTPP})_{1.8}(\text{HTPP})_{0.2}\text{MnBr}_4$ transparent scintillation screen.

Figure R10 Photos of remolded $(\text{BTTP})_{1.8}(\text{HTTP})_{0.2}\text{MnBr}_4$ restored to its original PLQY by reheating and melting (up: visible light; bottom: 365 nm UV light).

Figure R11 The stability of the $(\text{BTTP})_{1.8}(\text{HTTP})_{0.2}\text{MnBr}_4$ scintillation screen under continuous fast neutron irradiation.

Reviewer #2:

Review for "A transparent hybrid perovskite glassy scintillation screen for high-resolution fast neutron radiography"

The work submitted by Zhou et al., "A transparent hybrid perovskite glassy scintillation screen for high-resolution fast neutron radiography" presents the development of a glassy scintillating screen based on Mn-based hybrid perovskite, for fast neutron imaging. This screen is fabricated using a solvent-free melt quenching method, resulting in a high-transparency, glassy material. The glassy scintillator exhibits up to 80% transmittance in the visible spectrum and 84% photoluminescence quantum yield. The transparency of the glassy screen leads to a significantly improved spatial resolution compared to the polycrystalline screen, with a spatial resolution of 5 lp/mm. The work is

well presented, and we suggest publication with minor revisions, however, we would like the authors to answer the following queries below.

Response:

We appreciate reviewer's very encouraging comment.

Comment 1:

What was the rationale for the choice of triphenyl phosphonium (TPP) derivatives for cations? Is a very bulky cation necessary to obtain such a glassy state or would a smaller cation (e.g. phenylethylammonium or hexylammonium) also work? Would this be possible with other metallic cations (e.g. Pb^{2+} or Sn^{2+})?

Response 1:

Thanks for the referee's comment. Triphenyl phosphonium (TPP) derivatives were selected as the A-site cations primarily due to their high steric hindrance. The bulky TPP cations effectively suppress the formation of ordered crystalline phase, promoting the formation of amorphous structures. This structural characteristic minimizes light scattering and enhances material transparency, which is critical for achieving highly efficient transparent scintillators (J. Am. Chem. Soc. 2020, 142, 17878-17883; Nat. Photonics 2021, 15, 644-650; Angew. Chem. Int. Ed. 2023, 62, e202302406; Angew. Chem. Int. Ed. 2023, 62, e202216504; Chem. Sci., 2023, 14, 12238).

Although smaller cations such as phenylethylammonium or hexylammonium can also contribute to forming amorphous structures, their smaller size has less steric hindrance compared to TPP. This limitation could result in reduced transparency and lower amorphous stability, which make them easier to form opaque polycrystals (J. Mater. Chem. C, 2021, 9, 9952-9961; Angew. Chem. Int. Ed. 2023, 62; e202216504; Chem. Eng. J. 2024, 483; 149239; Adv. Optical Mater. **2023**, 2300216).

Furthermore, other metallic cations such as Pb^{2+} , Sb^{3+} , when combined with suitable organic cations, the transparent glassy state can also be formed. Based on the suggestions of the reviewers, $(\text{ETPP})_2\text{PbBr}_4$ transparent scintillator was successfully synthesized using the same synthesis method, but its PLQY is low (**Figure R12**). In addition, in another reported work of ours, $(\text{ETPP})_2\text{SbCl}_5$ demonstrates high transmittance exceeding 80% in the 450–800 nm range (ACS Nano 2024, 18, 26, 16715-16725). In this work, Mn-based halide was specifically chosen due to its lower toxicity, favorable scintillation properties, and good compatibility with the detection efficiency of CCDs,

making them highly suitable for fast neutron imaging. We will further investigate other potential metal cations to expand their applications in fast neutron imaging in future studies.

Figure R12 The photographs of the (ETPP)₂PbBr₄ scintillator (left: visible light; right: 365 nm UV light).

Comment 2:

The authors claim a spatial resolution of 5 lp/mm, however no modulation transfer function (MTF) is provided, which provides information regarding the contrast at a certain spatial frequency. In particular, Figure 5e shows that for 5 lp/mm (slit width of 0.1 mm) the contrast is quite low. The authors should measure and report the MTF vs. spatial frequency, as it's very important to understand the actual resolution capabilities of the screen quantitatively.

Response 2:

We appreciate the reviewer's insightful comments. In fast neutron imaging, Spatial resolution is typically evaluated using resolution templates and gray-scale extraction, rather than directly employing the modulation transfer function (MTF). This is because the non-parallel nature of fast neutron beams requires MTF test templates to have sharp edges and be as thin as possible. However, due to the strong penetration ability of fast neutrons, thin materials cannot fully block neutrons, resulting in reduced image contrast. To establish a quantitative relationship between MTF and spatial resolution, the contrast must be enhanced by increasing the thickness of the mask. Unfortunately, this introduces geometric unsharpness into the measurements, as described by the **Equation (9)** (Nucl. Tech. 2023, 46, 30-35):

$$G = \frac{d}{L/D} \quad (9)$$

where L is the distance from the neutron source to the image detector; D is the diameter at the exit of the neutron source, and d is the distance from the sample center to Scintillation screen.

Therefore, the intrinsic resolution of the scintillation screen can thus be expressed as **Equation**

(10):

$$Resolution_s \approx \sqrt{Resolution_{MTF}^2 - Resolution_G^2} \quad (10)$$

Where, $Resolution_s$ is the intrinsic resolution of the scintillation screen, $Resolution_{MTF}$ is the MTF value, and $Resolution_G$ refers to the resolution of the geometric unsharpness.

In the experiment, a 3 cm-thick stainless steel block was used as a mask to perform MTF testing on $(BTPP)_{1.8}(HTPP)_{0.2}MnBr_4$ transparent scintillation screen. The results that the spatial resolution is approximately 3.5 lp mm^{-1} at $MTF=0.1$, corresponding to a slit width of 0.145 mm (**Figure R13**).

Additionally, the L/D ratio of the neutron imaging system used in this work is 160. The geometric unsharpness G is calculated to be 0.094 mm when $d=1.5$ cm. Combining these results, the $Resolution_s$ of the scintillation screen is determined to be approximately 0.11 mm, consistent with the resolution obtained using the resolution template.

Moreover, when using double-slit resolution templates and gray-scale extraction methods, emphasis is placed on whether the double slits can be clearly distinguished (Nucl. Instrum. Methods. Phys. Res. A 2023, 1050 168179; IEEE Trans Nucl Sci. 2022, 69:11, 2245-2251; Phys. Procedia 2013, 43 205–215). We have re-extracted and plotted the resolution gray-scale values of the $(BTPP)_{1.8}(HTPP)_{0.2}MnBr_4$ transparent scintillation screen. (**Figure R14**). The contrast and separation of 5 lp mm^{-1} (at the 0.1 mm slit) were calculated to assess resolvability.

The Contrast was calculated as follows **Equation (11)**:

$$Contrast = \frac{I_{max} - I_{min}}{I_{max} + I_{min}} \quad (11)$$

Where I_{max} is the peak intensity; I_{min} is the valley intensity.

The contrast of the $(BTPP)_{1.8}(HTPP)_{0.2}MnBr_4$ transparent scintillation screen was determined to be approximately 19.31%–23.63%, exceeding the threshold of 10%–20%, indicating that the 0.1 mm slits are distinguishable.

Additionally, Separation and full width at half maximum (FWHM) values were used to evaluate resolvability based on the Rayleigh criterion by **Equation (12)**:

$$Separability = \frac{Separation}{FWHM} > 1.22 \quad (12)$$

Separation is the distance between two peaks by **Equation (13)**:

$$Separation = Peak_2 - Peak_1 \quad (13)$$

Where $Peak_1$ and $Peak_2$ are the positions of double peaks respectively.

The Separation between peaks was calculated to be 0.235 mm, while the FWHM values for the two peaks were 0.120 mm and 0.115 mm, respectively. The separability values of 1.96 and 2.03 meet the Rayleigh criterion, confirming that the peaks at 0.1 mm slit are distinctly resolvable (Figure R14).

In summary, the resolution results obtained from MTF testing align well with those derived from the gray-scale extraction method using resolution templates.

In response to these opinions, we have added Figure R13 and explanation in the revised Supplementary Information (Supplementary Fig. 27, Page 20-21) and relevant descriptions in the revised manuscript (Page 17).

Figure R13 MTF of (BTPP)_{1.8}(HTPP)_{0.2}MnBr₄ transparent scintillation screen.

Figure R14 Curves of the relative gray value distribution of (BTPP)_{1.8}(HTPP)_{0.2}MnBr₄ transparent scintillation screen.

Comment 3:

Regarding the fabrication of the glassy screen, was the operation performed in an oxygen and water-free atmosphere to avoid possible oxidation reactions between oxygen and the organic cation?

Response 3:

Thanks to the thoughtful reviewer, to minimize potential oxidation reactions between oxygen and the organic cations during the high-temperature heating process, the preparation of the glassy $(\text{BTPP})_{1.8}(\text{HTPP})_{0.2}\text{MnBr}_4$ scintillation screens was carried out in the nitrogen-filled glove box. The oxygen- and water-free environment was selected to ensure optimal material properties and performance. For comparison, $(\text{BTPP})_{1.8}(\text{HTPP})_{0.2}\text{MnBr}_4$ scintillation screens were also prepared in an air environment (**Figure R15a**). The results showed an obviously reduction in PLQY, which was measured at 53.53%, compared to 85.54% for those synthesized under oxygen- and water-free nitrogen conditions (**Figure R15b, Fig. 2e**). This substantial drop in PLQY underscores the critical importance of controlling the preparation atmosphere.

In the Methods section of the revised manuscript, we have added the “All preparation operations were carried out in a nitrogen-filled glove box.” (page 25).

Figure R15 a, The photographs of the $(\text{BTPP})_{1.8}(\text{HTPP})_{0.2}\text{MnBr}_4$ scintillation screen prepared in the air (up: visible light; bottom: 365 nm UV light). **b**, the PLQY of the $(\text{BTPP})_{1.8}(\text{HTPP})_{0.2}\text{MnBr}_4$ scintillation screen

Comment 4:

Is the glassy screen resistant to humidity? What is the material environmental stability? In addition, how stable is the material under irradiation? Have tests been performed to gauge the radiation

hardness of the glassy $\text{TPP}_2\text{MnBr}_4$ screens?

Response 4:

Thanks for the referee's comment. Humidity has an impact on glassy screen. To evaluate this, the environmental stability was tested with humidity of 25%, 50% and 75% under 25°C. As shown in **Figure R16**, after 30 days, $(\text{BTPP})_{1.8}(\text{HTPP})_{0.2}\text{MnBr}_4$ remains transparent in environments with 25% and 50% humidity, but becomes opaque under 75% humidity. PXRD also revealed crystallization peaks in the 75% humidity condition, indicating a transition from a transparent amorphous state to an opaque crystalline state (**Figure R17**). Nevertheless, it can be restored to its original PLQY by reheating and melting, making it practically for long-term use (**Figure R18**).

In addition, based on the actual neutron test environment, the stability of the storage conditions (at 25°C nitrogen) and fast neutron imaging conditions (at 25°C and 25% humidity air) of $(\text{BTPP})_{1.8}(\text{HTPP})_{0.2}\text{MnBr}_4$ were monitored, respectively (**Figure R19**). The results show that the transparent scintillation screen retains 99.88% of its initial quantum efficiency after 90 days of storage in a nitrogen (**Figure R19a**). Even in fast neutron imaging conditions, it maintains 57.17% of its original efficiency after 40 days (**Figure R19b**). However, in actual experiments, the scintillation screen is commonly used in short time (up to two hours), allowing it to be stored in nitrogen and exposed in air only during use.

Furthermore, $(\text{BTPP})_{1.8}(\text{HTPP})_{0.2}\text{MnBr}_4$ also exhibits good fast neutron irradiation stability, the light output intensity remains at around 98.3% of its initial value after 120 minutes of continuous irradiation (**Figure R20**), which can meet the normal requirements of FNR.

Therefore, the $(\text{BTPP})_{1.8}(\text{HTPP})_{0.2}\text{MnBr}_4$ scintillation screen has good environmental stability and radiation stability, which can meet the normal requirements of fast neutron imaging.

In response to these opinions, we have included the revised Supplementary Information (**Supplementary Fig. 33-38**, Page 25-27) and relevant descriptions in the revised manuscript (Page 19-20).

Figure R16 Photos of changes in $(\text{BTTP})_{1.8}(\text{HTPP})_{0.2}\text{MnBr}_4$ under different humidity environments for 30 days (up: visible light; bottom: 365 nm UV light).

Figure R17 The initial PXRD and the PXRD after being stored at 25%, 50% and 75% humidity levels at the same temperature for 30 days of $(\text{BTTP})_{1.8}(\text{HTPP})_{0.2}\text{MnBr}_4$ transparent scintillation screen.

Figure R18 Photos of remolded $(\text{BTTP})_{1.8}(\text{HTPP})_{0.2}\text{MnBr}_4$ restored to its original PLQY by reheating and melting. (up: visible light; bottom: 365 nm UV light).

Figure R19 The stability of the PLQY of $(\text{BTPP})_{1.8}(\text{HTPP})_{0.2}\text{MnBr}_4$ scintillation screen under both storage conditions (at 25°C nitrogen) (a) and fast neutron imaging conditions (at 25°C and 25% humidity air) (b).

Figure R20 The stability of the $(\text{BTPP})_{1.8}(\text{HTPP})_{0.2}\text{MnBr}_4$ scintillation screen under continuous fast neutron irradiation.

Comment 5:

There needs to be some further clarification on the experimental fabrication of the glassy screens. How was the quick quenching from melt temperature to 30°C done? The authors say the silicon moulds were pre-heated. Were the moulds pre-heated to the temperature of the melt? Was the quenching done by putting the silicon mould with the melt into a cooling liquid? How fast was the cooling? Please, elaborate further on these points.

Response 5:

We appreciate this very careful suggestion. Detailed quenching process for the fabrication of the

glassy screens is described as follows: After the melt was formed, it was quickly transferred into a silicone mold. It is noted that the silicone mold was pre-heated to the same temperature as the melt to prevent bubbles formation during the transfer and to maintain uniformity in the cooling process. The quick quenching from the melt temperature to 30°C was achieved by transfer the filled mold from the heating stage and allowing it to cool naturally on the tabletop at room temperature. This cooling process typically took 1-2 minutes.

In the Methods section of the revised manuscript, the relevant description has been updated (page 25).

Comment 6:

The decay times of the scintillation of the $\text{TPP}_2\text{MnBr}_4$ material is quite long, in the order of hundreds of microseconds. Does this pose a risk for significant afterglow and thus possible after images that might affect the imaging process?

Response 6:

Thanks for the referee's insightful comment. The lifetime of the scintillator $(\text{BTPP})_{1.8}(\text{HTPP})_{0.2}\text{MnBr}_4$ is 152 μs (Supplementary Fig. 9), but it wouldn't impact the imaging process in fast neutron radiography. This is primarily because fast neutron radiography typically requires long exposure time (tens of seconds) due to the strong penetration capability of fast neutrons. For example, in imaging test (Figure 5d, 5g, 5h), the single exposure time is 40 seconds, which is substantially longer than the scintillator's decay time. Under such conditions, the afterglow effect is negligible and does not introduce noticeable impact on image quality.

Comment 7:

In the XRD data provided in Supplementary Fig. 4, the authors claim "polycrystal showed almost identical diffraction peaks compared to the simulation of $(\text{BTPP})_2\text{MnBr}_4$ single crystal". While understandably a polycrystalline film peak intensity will differ from the theoretical prediction due to preferred orientation, I would suggest adding either the single crystal XRD diffractogram that was used in Supplementary Table 1 as well, and possibly the powder XRD diffractogram as well for comparison in Supplementary Fig. 4.

Response 7:

Thanks for the reviewer's helpful suggestion. PXRD of $(\text{BTPP})_2\text{MnBr}_4$ single crystal were conducted (**Figure R21**) that exhibit good alignment with the peaks simulated from single crystal X-ray diffraction (SCXRD). As expected, the polycrystalline sample displays variations in peak intensity due to preferred orientation. However, the main diffraction peaks of the $(\text{BTPP})_{1.8}(\text{HTPP})_{0.2}\text{MnBr}_4$ polycrystal and the $(\text{BTPP})_2\text{MnBr}_4$ polycrystal remain almost identical when compared to the PXRD of $(\text{BTPP})_2\text{MnBr}_4$ single crystal.

In the revised Supplementary Information, we have added **Figure R21** as **Supplementary Fig. 6** and supplemented its related description (Page 7).

Figure R21 PXRD of $(\text{BTPP})_{1.8}(\text{HTPP})_{0.2}\text{MnBr}_4/(\text{BTPP})_2\text{MnBr}_4$ polycrystal, the PXRD and SCXRD simulation of $(\text{BTPP})_2\text{MnBr}_4$ single crystal.

Comment 8:

Would $\text{TPP}_2\text{MnBr}_4$ also perform well for X-ray imaging?

Response 8:

Thanks for the reviewer's comment. $(\text{BTPP})_{1.8}(\text{HTPP})_{0.2}\text{MnBr}_4$ also performs well for X-ray imaging. It has a competitive spatial resolution of 30 lp mm^{-1} for X-ray imaging (**Figure R22**). Despite these good results in X-ray imaging, this work mainly focuses on fast neutron imaging.

Figure R22 (a) $(\text{BTPP})_{1.8}(\text{HTPP})_{0.2}\text{MnBr}_4$ resolution image and (b) grayscale value distribution of the marked area.

Reviewer #3:

This work by Zhou et al. describes the development of a candidate fast neutron scintillation screen based on alkyltriphenylphosphonium cations and MnBr_4 tetrahedral units, which can form glassy moldable screens with bright green emission. The mechanical, optical, and scintillation performance are reported, with good performance shown by this scintillator relative to established materials. The glassy nature of the screens are a significant advantage due to their transparency and moldability, and the use of mechanical property testing is a nice touch that isn't typically included in the other candidate FNR detector materials I've seen. My primary concerns are on the data reporting and manuscript clarity, especially regarding the spatial resolution. Once addressed, this manuscript would be a suitable advance for a Nature Communications publication.

Response:

We appreciate reviewer's very encouraging comment.

Comment 1:

Error bars and error values are missing from most metrics reported in this manuscript (e.g. PLQY, spatial resolution, PL lifetime, bond distances, light yield) – these must be included.

Response 1:

Thanks to the careful reviewer, following the suggestion, additional experiments were conducted to include error bars for PLQY and light output (Figure R23-25), reflecting the repeatability and reliability of experimental results.

For spatial resolution and PL lifetime, it is worth noting that these metrics are not typically reported with error values in the previously published literature (Nat. Commun. 2023, 14, 2808; Nature 2018, 561, 88–93; J. Am. Chem. Soc. 2020, 142, 13582–13589).

Regarding bond distances, the annotation has been removed from **Figure 4a** to avoid any ambiguity. Instead, **Figure 4b-c** can more accurately depict the statistical distribution of bond distance variations under different conditions, providing a clearer and more comprehensive representation.

In the revised manuscript we replaced **Figure 2e** and **Figures 5b-c** with **Figure R23** and **Figures 25a-b** which include error bars, and modified the corresponding figure description (page 6; page 14).

In the revised Supplementary Information, the **Supplementary Fig. 3** was replaced with **Figure R24** containing error bars, and the corresponding figure descriptions was modified (page 4).

Figure R23 The PLQY of the (i)-(iv) transparent scintillation screens. (i): (BTTP)₂MnBr₄, (ii): (BTTP)_{1.8}(HTTP)_{0.2}MnBr₄, (iii): (BTTP)_{1.8}(DTTP)_{0.2}MnBr₄, (iv): (BTTP)_{1.8}(CTTP)_{0.2}MnBr₄.

Figure R24 The PLQY of [(BTTP)_{1-x}(HTTP)_x]₂MnBr₄ (1-x: x = 1: 0; 0.9: 0.1; 0.8: 0.2; 0.7: 0.3).

Figure R25 a, The light output of different transparent scintillation screens, 2D Mn-STA₂PbBr₄ perovskite plate and ZnS (Ag): PP commercial fast neutron scintillation screen (ZnS (Ag): PP as a benchmark value of 1). **b**, The light output of (BTTP)_{1.8}(HTTP)_{0.2}MnBr₄ transparent scintillation screens with different thicknesses.

Comment 2:

The spatial resolution must be calculated using the modular transfer function, as it is typically done in the radiography community. Fitting a Gaussian to one of the larger lines in the image of Figure 5d would be sufficient to enable a quantitative measure of spatial resolution. The data in Figure 5e do not clearly resolve the 0.1 slit as described by the authors, as the valley separating the two peaks is similar to the noise level of the image, and the peaks are not fully distinct. More likely, an appropriate resolution value is the 0.3 mm slit, which would lead to a spatial resolution of 1.7 lp/mm. This is still an excellent and competitive value! Using the MTF will be able to provide more precision and reduce the error on this estimate.

Response 2:

We appreciate the reviewer's insightful comments. In fast neutron imaging, Spatial resolution is typically evaluated using resolution templates and gray-scale extraction, rather than directly employing the modulation transfer function (MTF). This is because the non-parallel nature of fast neutron beams requires MTF test templates to have sharp edges and be as thin as possible. However, due to the strong penetration ability of fast neutrons, thin materials cannot fully block neutrons, resulting in reduced image contrast. To establish a quantitative relationship between MTF and spatial resolution, the contrast must be enhanced by increasing the thickness of the mask.

Unfortunately, this introduces geometric unsharpness into the measurements, as described by the **Equation (14)** (Nucl. Tech, 2023, 46,30-35):

$$G = \frac{d}{L/D} \quad (14)$$

where L is the distance from the neutron source to the image detector; D is the diameter at the exit of the neutron source, and d is the distance from the sample center to Scintillation screen.

Therefore, the intrinsic resolution of the scintillation screen can thus be expressed as **Equation (15)**:

$$Resolution_s \approx \sqrt{Resolution_{MTF}^2 - Resolution_G^2} \quad (15)$$

Where, $Resolution_s$ is the intrinsic resolution of the scintillation screen, $Resolution_{MTF}$ is the MTF value, and $Resolution_G$ refers to the resolution of the geometric unsharpness.

In the experiment, a 3 cm-thick stainless steel block was used as a mask to perform MTF testing on (BTTPP)_{1.8}(HTPP)_{0.2}MnBr₄ transparent scintillation screen. The results that the spatial resolution is approximately 3.5 lp mm⁻¹ at MTF=0.1, corresponding to a slit width of 0.145 mm (**Figure R26**).

Additionally, the L/D ratio of the neutron imaging system used in this work is 160. The geometric unsharpness G is calculated to be 0.094 mm when d=1.5 cm. Combining these results, the $Resolution_s$ of the scintillation screen is determined to be approximately 0.11 mm, consistent with the resolution obtained using the resolution template.

Moreover, when using double-slit resolution templates and gray-scale extraction methods, emphasis is placed on whether the double slits can be clearly distinguished (Nucl. Instrum. Methods. Phys. Res. A 2023, 1050 168179; IEEE Trans Nucl Sci, 2022, 69:11, 2245-2251; Phys. Procedia, 2013, 43 205–215). We have re-extracted and plotted the resolution gray-scale values of the (BTTPP)_{1.8}(HTPP)_{0.2}MnBr₄ transparent scintillation screen. (**Figure R27**). The contrast and separation of 5 lp mm⁻¹ (at the 0.1 mm slit) were calculated to assess resolvability.

The Contrast was calculated as follows **Equation (16)**:

$$Contrast = \frac{I_{max} - I_{min}}{I_{max} + I_{min}} \quad (16)$$

Where I_{max} is the peak intensity; I_{min} is the valley intensity.

The contrast of the (BTTPP)_{1.8}(HTPP)_{0.2}MnBr₄ transparent scintillation screen was determined to be approximately 19.31%–23.63%, exceeding the threshold of 10%–20%, indicating that the 0.1 mm slits are distinguishable.

Additionally, Separation and full width at half maximum (FWHM) values were used to evaluate resolvability based on the Rayleigh criterion by **Equation (17)**:

$$\text{Separability} = \frac{\text{Separation}}{\text{FWHM}} > 1.22 \quad (17)$$

Separation is the distance between two peaks as **Equation (18)**:

$$\text{Separation} = \text{Peak}_2 - \text{Peak}_1 \quad (18)$$

Where Peak_1 and Peak_2 are the positions of double peaks respectively.

The Separation between peaks was calculated to be 0.235 mm, while the FWHM values for the two peaks were 0.120 mm and 0.115 mm, respectively. The separability values of 1.96 and 2.03 meet the Rayleigh criterion, confirming that the peaks at 0.1 mm slit are distinctly resolvable (**Figure R27**).

In summary, the resolution obtained from the MTF test is consistent with the resolution derived using the gray-scale extraction method with resolution templates, confirming that the resolution of the $(\text{BTPP})_{1.8}(\text{HTPP})_{0.2}\text{MnBr}_4$ transparent scintillation screen is 5 lp mm^{-1} (0.1 mm slit).

In response to these opinions, we have added **Figure R26** and explanation in the revised Supplementary Information (**Supplementary Fig. 27**, Page 20-21) and relevant descriptions in the revised manuscript (Page 17).

Figure R26 MTF of $(\text{BTPP})_{1.8}(\text{HTPP})_{0.2}\text{MnBr}_4$ transparent scintillation screen.

Figure R27 Curves of the relative gray value distribution of $(\text{BTTPP})_{1.8}(\text{HTPP})_{0.2}\text{MnBr}_4$ transparent scintillation screen.

Comment 3:

This methodology should also be used to calculate the spatial resolution for all thicknesses of the detector – even transparent detectors offer reduced resolution as thickness increases, and this should be quantified.

Response 3:

We appreciate the reviewer’s thoughtful comments. We assessed the resolution of scintillation screens with thicknesses of 1.8 mm and 2.8 mm using resolution templates (Fig. 5d and Supplementary Fig. 26) and observed no significant changes in resolution. To quantify the relationship between scintillation screen thickness and resolution, MTF measurements for $(\text{BTTPP})_{1.8}(\text{HTPP})_{0.2}\text{MnBr}_4$ transparent scintillation screen at various thicknesses, using ZnS (Ag): PP as a reference were conducted (**Figure R28**). The results show that: When the MTF is increased from 1.8 mm to 2.8 mm, the MTF is almost unchanged, both are 3.5 lp mm^{-1} , combined with geometric unsharpness, the corresponding spatial resolution is about 4.55 lp mm^{-1} , and then the resolution decreases with the increase of the thickness (from 2.8 mm to 3.8 mm). At a thickness of 3.8 mm, the MTF value dropped to 2.1 lp mm^{-1} , corresponding to a spatial resolution of 2.8 lp mm^{-1} .

¹, but is still higher than that of ZnS (Ag): PP (1.72 lp mm⁻¹ at MTF=0.1, the corresponding spatial resolution is about 1.82 lp mm⁻¹).

In response to these opinions, we have added **Figure R28** and explanation in the revised Supplementary Information (**Supplementary Fig. 28**, Page 20-21) and relevant descriptions in the revised manuscript (Page 17).

Figure R28 MTF of (BTPP)_{1.8}(HTPP)_{0.2}MnBr₄ transparent scintillation screen at various thicknesses, using ZnS (Ag): PP as a reference.

Comment 4:

All materials presented in this work are OD hybrid halides, not perovskites. The perovskite structure requires octahedra with corner-shared connectivity – here all units are tetrahedral. It is appropriate to reference other perovskite structures such as the Mn-STA₂PbBr₄ described in the introduction, but these Mn-based compounds are not perovskites.

Response 4:

Thank you for the reviewer’s valuable evaluation of all materials proposed in this work as part of the “perovskite” classification. The concern that the material does not possess the characteristic octahedral structure associated with conventional perovskites is comprehended. As the structures in this work are tetrahedral, they are more accurately described as “hybrid metal halides”.

Thus, following reviewer's suggestion and references, the manuscript has been revised to avoid any misleading use of the term "perovskite" and has consistently referred to the materials as "hybrid metal halides". These changes have been implemented throughout the text in the revised manuscript.

Comment 5:

The abstract and conclusion would benefit from more specific details (e.g. light output values vs. reference).

Response 5:

Thanks to the careful reviewer. In the revised manuscript, more specific details such as light output (BTTPP)_{1.8}(HTPP)_{0.2}MnBr₄ is three times higher than commercial scintillation screens ZnS (Ag): PP, have been added to the abstract and conclusion (page 1; page 20).

Comment 6:

Larger Mn-Mn distances have been shown to improve PLQY (e.g. DOI: 10.1021/jacs.0c06039) in manganese halides, not decrease it as suggested – thus, the larger cation size must exert a different influence into the glassy state than the suggestion of the authors. This may also be tied to the decreased luminescence lifetime in the glassy state, and this connection deserves further consideration and investigation.

Response 6:

Thanks for the referee's comment. The statement that "Larger Mn-Mn distances have been shown to improve PLQY (e.g. DOI: 10.1021/jacs.0c06039) in manganese halides, not decrease it as suggested" is somewhat one-sided. The cited work (DOI: 10.1021/jacs.0c06039) discusses the effect of Mn-Mn distance on PLQY only in relation to specific types and subjects of cations (e.g. trimethylphenylammonium (TMPEA⁺), tetraphenylphosphonium (PPh₄⁺)). However, it does not systematically examine this effect by varying the branched alkyl chain length on the same kind of cation. And the coordination effects between N-Mn and P-Mn in these systems also influence the results, making it difficult to attribute changes in PLQY solely to Mn-Mn distance.

On the contrary, another work (J. Mater. Chem. C, 2021, 9, 9952–9961) we cited, systematically investigated the relationship between Mn-Mn distance and PLQY by varying the alkyl chain length on a single type of cation, which aligns more closely with our system. Their findings demonstrate

that the cations with excessive steric hindrance such as those with longer branches (e.g. tetradecyltrimethylammonium (TTMA⁺), octadecyltrimethylammonium (OTMA⁺)) can cause the low concentration of [MnX₄]²⁻ units. This reduction in [MnX₄]²⁻ units diminishes the light absorption of inorganic components and ultimately reduces emission efficiencies (J. Mater. Chem. C, 2021, 9, 9952–9961, Chem. Sci., 2024, 15, 16338–16346). These results are consistent with the PLQY tests for pure cationic metal halides in this work (**Figure R29**), wherein a gradual decrease in PLQY is observed as the length of the branched alkyl chain increases within the system.

In summary, there are many factors that ultimately affect PLQY, and Mn-Mn distance is just one of them, so we do not attribute all changes in PLQY to Mn-Mn distance.

In the revised manuscript, we have deleted the relevant sections.

Figure R29 The PLQY of the single-cation metal halide transparent scintillation screens.

Comment 7:

Are the AIMD-derived bond distances in Figure 4a meaningful and reproducible if the simulations are re-run? Given the breadth of the bond distributions in the following figures, it's not obvious to me that these values are truly reliable and comparable for the molten and glassy states.

Response 7:

Thank you for your careful comment. The bond distances in **Figure 4 (Figure R30)** are meaningful and reproducible, as they are derived from statistical analyses of nearly 10,000 geometric structures obtained during the *ab initio* molecular dynamics (AIMD) simulations. This methodology aligns with established practices in previously reported studies (Nat. Commun., 2024, 15, 6292; Nat. Mater., 2017, 16, 1149–1154).

The radial distribution function (RDF), which is based on these statistics, provides a clearer understanding of structural behavior by capturing variations in bond distances. The broader bond distributions observed for the molten and glassy states reflect their inherent dynamic nature, which results in fluctuating bond lengths over time due to short-range disorder. This behavior is typical for amorphous systems and is well-documented in the literature (J. Phys. Chem. B, 2018, 122, 3401–3409; J. Mater. Chem. A, 2021, 9, 16530–16539).

Furthermore, these broader distributions are a characteristic feature of the molten and glassy states, distinguishing them from crystalline structures. The comparison between the molten and glassy states is particularly meaningful, as the broader distribution itself serves as an important indicator of the structural differences between these phases. (Phys. Rev. B, 2004, 70, 144201). Regarding the bond distances labeled in **Figure 4a**, we acknowledge that these represent their individual snapshots rather than statistically significant values. As such, they may not fully capture the structural distribution across the entire simulation. To avoid potential misinterpretation, it was decided to remove these labels from **Figure 4a** (J. Am. Chem. Soc. 2024, 146, 7373–7385; J. Chem. Phys. 2009, 130, 014703).

In the revised manuscript, we have made the revisions to **Figure 4** as the following figure, and we also modified the corresponding description (page 12-13).

Figure R30 (Figure 4 in the revised manuscript) AIMD and light transmission mechanism diagram

for glassy and polycrystalline state scintillation screens.

Comment 8:

Can the authors determine the hydrogen density of these screens based on the composition and density? It is important to calculate this parameter and compare with other candidate materials – it will enable the authors to also test if their detector has the needed stopping power for these high energy neutrons.

Response 8:

Thank you for the reviewer's suggestions. The hydrogen density of the transparent scintillation screens with various doping components can be determined based on their composition and density. ZnS (Ag): PP scintillators and other commonly used scintillators were used as references (**Table R6**). And the elastic scattering cross section, and fast neutron stopping power were calculated using TENDL-2023 database combined with the fast neutron spectra used in the experiment (average energy: 2.69 MeV) (**Figure R31** and **Table R6**). The results show that the hydrogen density, elastic scattering cross section and fast neutron stopping power of the transparent scintillator with different components ranged from 68.92 to 87.09 kg m⁻³, 0.2698 to 0.3052 cm⁻¹, and 2.662% to 3.006%, respectively. These values are comparable to those of stilbene (67.33 kg m⁻³, 0.2690 cm⁻¹ and 2.553 %) and anthracene (71.91 kg m⁻³, 0.3100 cm⁻¹ and 2.982%), but are lower than ZnS (Ag): PP (136.9 kg m⁻³, 0.4556 cm⁻¹ and 4.454%). For a series of transparent scintillation screens, we observe that increasing the length of the doped branched alkyl chain resulted in higher hydrogen density, which subsequently enhanced the scattering cross section and improved the fast neutron stopping ability.

Taking (BTPP)_{1.8}(HTPP)_{0.2}MnBr₄ and ZnS (Ag): PP as examples, although (BTPP)_{1.8}(HTPP)_{0.2}MnBr₄ has lower hydrogen density and fast neutron stopping power than ZnS (Ag): PP, it does not directly affect its high light output. Because the scintillation performance for fast neutrons is influenced by several key factors, including: (1) hydrogen density which directly affects the fast neutron response, (2) interaction efficiency between fast neutrons and scintillation screen (neutron deposition efficiency, recoil proton deposition efficiency), (3) PLQY of the material, (4) number of photon output from the scintillation screen reaching the detector (ACS Nano 2020, 14, 14686–14697).

Further, Geant4 software was used to simulate the interaction between fast neutrons and the sample, providing energy data for each stage (**Table R7**). The simulation involved 10^7 neutrons with an average energy of 2.69 MeV, using the FTFP_SERT_HP model. The recoil proton deposition energy for (BTTP)_{1.8}(HTTP)_{0.2}MnBr₄ and ZnS (Ag): PP is 2.323×10^5 and 4.831×10^5 MeV, respectively. The recoil proton energy deposition efficiency is 77.35% for (BTTP)_{1.8}(HTTP)_{0.2}MnBr₄ and 84.05% for ZnS (Ag): PP (**Table R7** and **Equation (21)**), showing that their responses to charged particles are not significantly different. (Due to current theoretical model limitations, both (BTTP)_{1.8}(HTTP)_{0.2}MnBr₄ and ZnS (Ag): PP scintillators are modeled as homogeneous single component structures representing idealized values. This modeling does not account for the advantages of a single component over physical mixing.)

Therefore, the primary difference between the two scintillators lies in neutron deposition, which has a greater impact on performance, while recoil protons have a smaller effect. Notably, the PLQY of (BTTP)_{1.8}(HTTP)_{0.2}MnBr₄ (85.54%) is twice that of ZnS (Ag): PP (31.38%), compensating for the impact of neutron deposition on light output.

Moreover, utilizing theoretical calculations of neutron-generated recoil protons (**Table R8**) and **Equation (22)**, the number of neutrons generating recoil protons was 127.8 n pix^{-1} for (BTTP)_{1.8}(HTTP)_{0.2}MnBr₄ and 250.5 n pix^{-1} for ZnS (Ag): PP. Despite lower neutron interaction and recoil proton generation probabilities for (BTTP)_{1.8}(HTTP)_{0.2}MnBr₄, its detected grayscale value (approximately 208) is nearly three times that of ZnS (Ag): PP (approximately 70), indicating superior photon statistics due to higher transparency and efficient photon transmission. The grayscale values were extracted from detection results (**Table R8**).

The enhanced performance of (BTTP)_{1.8}(HTTP)_{0.2}MnBr₄ is primarily attributed to its amorphous structure induced high transparency. In contrast, ZnS (Ag): PP is an opaque mixture of polymer and powder components, which exhibits severe light scattering, leading to a limited optical absorption length of 0.142 mm. Thus, only photons generated within the superficial 0.142 mm layer are effectively captured by the detector. However, high transparency (BTTP)_{1.8}(HTTP)_{0.2}MnBr₄ has an optical absorption length of approximately 15.9 mm, which allows the majority of emitted photons to reach the detector.

In summary, despite the hydrogen density and stopping power of (BTTP)_{1.8}(HTTP)_{0.2}MnBr₄ are lower than those of ZnS (Ag): PP, the light output of metal halide (BTTP)_{1.8}(HTTP)_{0.2}MnBr₄ under

fast neutrons is still higher than that of ZnS (Ag): PP after comprehensive performance comparison. In the revised manuscript, the discussions regarding the hydrogen density, elastic scattering cross-section, and fast neutron stopping power of the transparent scintillation screens, as well as other commonly used scintillators, have been added (Page 5).

Table R6 Density, hydrogen density, fast neutron reaction cross-section and fast neutron stopping power of hybrid metal halide scintillators, ZnS (Ag): PP scintillators, and other commonly used scintillators.

Compound*	Density (g/cm ³)	Hydrogen density (kg/m ³)	Elastic scattering cross-section (cm ⁻¹)	Fast neutron stopping power (Scattered fast neutrons in 1 mm of host) (φ , %)
(BTPP) ₂ MnBr ₄	1.454	68.92	0.2698	2.662
(HTPP) ₂ MnBr ₄	1.373	75.10	0.2817	2.777
(DTPP) ₂ MnBr ₄	1.277	82.61	0.2963	2.919
(CTPP) ₂ MnBr ₄	1.223	87.09	0.3052	3.006
(BTPP) _{1.8} (HTPP) _{0.2} MnBr ₄	1.440	69.40	0.2703	2.667
(BTPP) _{1.8} (DTPP) _{0.2} MnBr ₄	1.423	70.39	0.2719	2.682
(BTPP) _{1.8} (CTPP) _{0.2} MnBr ₄	1.407	70.99	0.2726	2.689
ZnS (Ag): PP	1.900	136.9	0.4556	4.454
Stilbene	1.010	67.33	0.2690	2.553
Anthracene	1.280	71.91	0.3100	2.982

* BTPP⁺ = C₂₂H₂₄P⁺; HTPP⁺ = C₂₅H₃₀P⁺; DTPP⁺ = C₃₀H₄₀P⁺; CTPP⁺ = C₃₄H₄₈P⁺; PP = (C₃H₆)_n; stilbene = C₁₄H₁₂; anthracene = C₁₄H₁₀.

Table R7 Key parameters and the summary of the physical process of the interaction between fast neutrons and (BTTP)_{1.8}(HTTP)_{0.2}MnBr₄, ZnS (Ag): PP.

Parameters	(BTTP) _{1.8} (HTTP) _{0.2} MnBr ₄	ZnS (Ag): PP
PLQY	85.54%	31.38%
Density (g cm ⁻³)	1.440	1.900
Hydrogen density (kg m ⁻³)	69.40	136.9
Average elastic scattering cross-section under fast neutron energy spectrum (cm ⁻¹)	0.2703	0.4556
Total energy of fast neutron incidence (E_0 , MeV)	2.69×10^7	2.69×10^7
Neutron deposition energy (E_n , MeV)	3.003×10^5	5.748×10^5
Neutron energy deposition efficiency (δ)	1.116%	2.137%
Generated recoil proton energy (E_{r1} , MeV)	2.688×10^5	5.368×10^5
Recoil proton deposition energy (E_{r2} , MeV)	2.323×10^5	4.831×10^5
Recoil proton energy deposition efficiency (σ)	77.35%	84.05%

Table R8 The conversion probability of recoil protons and the grayscale value related data of (BTTP)_{1.8}(HTTP)_{0.2}MnBr₄ and ZnS (Ag): PP.

Parameters	(BTTP) _{1.8} (HTTP) _{0.2} MnBr ₄	ZnS (Ag): PP
Number of incident fast neutrons (N_0)	1×10^7	1×10^7
Number of neutrons that generate recoil protons (N)	1.278×10^5	2.505×10^5
The conversion probability of recoil protons (ϵ)	1.278%	2.505%
Number of neutrons reaching the sample* ¹	10000	10000
Number of neutrons that generate recoil protons* ²	127.8	250.5
Measured grayscale value	208 ± 5	70 ± 3
PLQY	85.54%	31.38%

*¹ Under the detection condition of a neutron flux of 10^7 n cm⁻² s⁻¹, an imaging size of 50 μ m pix⁻¹ and an exposure time of 40 s, the number of neutrons reaching the sample was calculated to be 10000 n pix⁻¹: 10^7 n cm⁻² s⁻¹ \times (50×10^{-4} cm pix⁻¹)² \times 40 s = 10000 n pix⁻¹.

*² Number of neutrons reaching the sample is 10000 n pix⁻¹, The conversion probability of recoil protons (ϵ) of (BTTP)_{1.8}(HTTP)_{0.2}MnBr₄ and ZnS (Ag): PP is 1.278% and 2.505%, respectively. The number of neutrons that generate recoil protons was calculated to be 127.8 and 250.5, respectively: $10000 \times 1.278\% = 127.8$; $10000 \times 2.505\%$

= 250.5.

The fast neutron stopping power is defined by **Equation (19)**:

$$\varphi = 1 - e^{-\mu x} \quad (19)$$

where μ is the elastic scattering cross-section and x is the thickness of the scintillation screen. Calculate the fast neutron stopping power of different samples when x is taken with the same thickness of 1 mm.

The neutron energy deposition efficiency is defined by **Equation (20)**,

$$\delta = \frac{E_n}{E_0} \quad (20)$$

where E_0 is the total energy of fast neutron incidence and E_n is the neutron deposition energy.

The recoil proton energy deposition efficiency is defined by **Equation (21)**,

$$\sigma = \frac{E_{r2}}{E_n} \quad (21)$$

where E_n is the neutron deposition energy and E_{r2} is the recoil proton deposition energy.

The conversion probability of recoil protons is defined by **Equation (22)**,

$$\varepsilon = \frac{N_0}{N} \quad (22)$$

where N_0 is the number of incident fast neutrons and N is the number of neutrons that generate recoil protons (there are cases where one neutron generates multiple recoil protons).

Figure R31 The energy spectra of the fast neutrons in the experiment. The average energy is 2.69 MeV, produced by thermal neutron induced U^{235} fission. A B^{10} filter was used to remove thermal and epithermal neutrons with energy below 0.01 MeV.

Comment 9:

How was the gamma-ray sensitivity of these detectors evaluated? While the authors do use lead shielding in the beam path, it's not specified how efficiently this screens the gamma-ray background – the details of the beam characteristics with and without the shielding should be discussed. If the gamma-ray sensitivity is appreciable, this may contribute to the imaging performance observed here, especially for the spatial resolution testing.

Response 9:

Thank you very much for your professional review. As mentioned by the reviewer, the accompanying γ -ray in the neutron beam can interfere with the experimental results. To address this, a series of tests were conducted to evaluate the γ -ray sensitivity of the scintillation screens. Firstly, the energy response of different scintillation screens to γ -ray was tested using a ^{137}Cs γ -ray source (**Figure R32**) (Adv. Funct. Mater. 2024, 34, 2308263). The results show that, unlike the significant response observed for NaI (Tl), both $(\text{BTPP})_{1.8}(\text{HTPP})_{0.2}\text{MnBr}_4$ and ZnS (Ag): PP exhibit minimal response to γ -ray. This indicates that $(\text{BTPP})_{1.8}(\text{HTPP})_{0.2}\text{MnBr}_4$ and ZnS (Ag): PP have low sensitivity to γ -ray background.

Furthermore, to evaluate the efficiency of lead shielding in blocking γ -rays, the energy response of NaI (Tl) was tested under different thicknesses (**Figure R33**). The results show that as the lead thickness increases, the γ -ray response of NaI (Tl) diminishes significantly (**Figure R33a**). By fitting the peak data, it was found that a lead shielding thickness of 10.42 mm is sufficient to completely block γ -ray (**Figure R33b**). Therefore, for fast neutron resolution imaging experiments, we used a 1 cm thickness of lead shielding and extracted the corresponding grayscale values from the imaging results, achieving a spatial resolution of 0.1 mm (5 lp mm^{-1}) (**Figure R34**). This confirms that γ -rays have negligible impact on the resolution results.

Overall, lead shielding effectively blocks a substantial portion of γ -ray background, while $(\text{BTPP})_{1.8}(\text{HTPP})_{0.2}\text{MnBr}_4$ and ZnS (Ag): PP have low sensitivity to γ -ray. Therefore, the accompanying γ -ray in the neutron beam contribute minimally to the observed imaging performance, including spatial resolution testing.

Figure R32 Pulse-height spectra of NaI (TI), $(\text{BTTP})_{1.8}(\text{HTPP})_{0.2}\text{MnBr}_4$ and ZnS (Ag): PP under γ -ray.

Figure R33 (a) Pulse-height spectra of NaI (TI) with different thicknesses of lead shielding under γ -ray. (b) Peak fitting of pulse-height at different thicknesses.

Figure R34 a, b Fast neutron imaging of resolution test standard samples using $(\text{BTTP})_{1.8}(\text{HTPP})_{0.2}\text{MnBr}_4$ transparent scintillation screen with a 1 cm thickness of lead shielding (a) and the extracted relative gray value (b).

Comment 10:

The transmission data in Fig 2f is only above 80% from 500-600 nm and 750-800 nm, not from 400-800 as stated – the description should clarify this. A more accurate statement could be that the transmission is above 70% from 500-800 nm.

Response 10:

Thanks to the careful reviewer. Upon reviewing the transmission data in **Figure 2f**, we agree that the original statement was inaccurate. The transmission is indeed above 80% only in the ranges of 500-600 nm and 750-800 nm, rather than across the entire 400-800 nm range as initially described. In revised manuscript, we have revised the description to clarify that the transmission is “above 70% from 500-800 nm”, ensuring a more precise and accurate representation of the data. (**Page 1, line 8; Page 3, line 58; Page 9, line 204**).

Comment 11:

The following phrase is unclear to me: “fast neutrons possess a unique advantage in isotope discrimination and radioactive material detection.....explosion-proof” – how are fast neutrons utilized in radioactive material detection? How do neutrons make or test explosion-proof materials? Do the authors mean fast neutron scintillation detectors or the neutrons themselves?

Response 11:

Thank you for your valuable comment. We sincerely apologize for the confusion caused by the phrasing. To clarify, our intended meaning was that “fast neutrons when combined with detectors and other analytical methods, possess unique advantages in isotope differentiation and radioactive material detection”. This is primarily due to the superior penetration capability of fast neutrons compared to thermal neutrons and cold neutrons. Fast neutrons can penetrate dense materials and interact with atomic nuclei through scattering. Different elements exhibit distinct neutron scattering cross-sections, thus providing information about material structure and composition. These characteristics render fast neutrons indispensable in various application fields, including radioactive material detection and the evaluation of explosion-proof materials, etc.

Regarding the use of fast neutrons in radioactive material detection: Fast neutrons contribute to radioactive material detection mainly through neutron activation analysis, elastic and inelastic scattering, detection of special nuclear materials, active detection techniques, and differential die-

away analysis. Leveraging their strong penetrability and sensitivity to light elements, fast neutrons penetrate the tested object and analyze the resulting scattered neutrons or γ -rays from the interaction between fast neutrons and atomic nuclei. This approach enables efficient, non-destructive detection and identification of radioactive materials. For instance, researchers have employed dual-particle (fast neutron and γ -ray) imaging and delayed neutron die-away analysis using a DT neutron generator to detect and characterize Special Nuclear Materials (SNMs), encompassing both metallic and oxide forms of uranium and plutonium (Sci. Rep. 2023, 13, 10432).

Regarding the use of fast neutrons in the detection of explosion-proof materials: Fast neutrons can penetrate thick outer walls and interact with the interior of explosion-proof materials. Due to their varying interactions with different materials, distinct degrees of attenuation are exhibited. By measuring the extent of attenuation, one can infer the material's density, uniformity, and integrity. Fast neutrons offer numerous advantages in the detection of explosion-proof materials, including (1) non-destructive testing, which permits examination of the internal structure and properties without damaging the material; (2) strong penetration capability, enabling them to pass through materials impervious to X-rays; and (3) sensitivity to light elements, facilitating the detection of polymer components, among others. The widespread application of fast neutron technology can effectively enhance the design and detection level of explosion-proof materials, thereby ensuring they meet safety and reliability standards in practical applications.

Finally, while fast neutrons offer substantial benefits, their practical utility requires integration with additional tools such as detectors and activation techniques to fully harness their potential.

In the revised manuscript, we have changed the sentence “fast neutrons possess a unique advantage in isotope discrimination and radioactive material detection...explosion-proof.” (Page 2, lines 25-27) to “fast neutrons binding detectors and related analytical methods offer distinct advantages in isotope differentiation and radioactive material detection...explosion-proof.” to clearly show the advantages of fast neutron detectors.

Comment 12:

This statement is not correct and should be reworded for clarity: “The $[\text{MnBr}_4]^{2-}$ serves as the luminescent center that efficiently transfers the recoil protons and excites them to emit visible light.”

Response 12:

Thanks for your helpful comment. The original statement was indeed inaccurate, and we have reworded it for clarity. The $[\text{MnBr}_4]^{2-}$ anion acts as the luminescent center in the scintillator material, but is not directly involved in transferring recoil protons. It still plays a crucial role in the luminescent mechanism. When recoil protons (produced from neutron interactions) deposit energy onto the $[\text{MnBr}_4]^{2-}$ centers, these centers become excited and subsequently emit visible light. This excitation and emission process is fundamental to the scintillation mechanism and enables the detection of neutron interactions.

In the revised manuscript, the statement has been revised to “The recoil protons deposited energy on the luminescent center $[\text{MnBr}_4]^{2-}$, exciting it to emit visible light.” to express correctly (Page 5, lines 106-107).

Comment 13:

The manuscript should be very carefully edited. A small selection of minor issues are listed for the author’s convenience but there are myriad similar errors in the present manuscript:

- a. Line 88 – presented should be replaced by captured or detected
- b. Line 90 – omit the word seriously
- c. Line 94 – The designing proposal of a should be replaced by “A prospective”

Response 13:

Thanks to the careful reviewer, we have carefully checked the details of the manuscript and corrected some grammar errors and inappropriate expressions.

- a. We have changed the inappropriate expression “presented” (**page 5, line 88**) to “captured” in the revised manuscript.
- b. We have deleted the wrong expression “seriously” (**page 5, line 90**) in the revised manuscript.
- c. We have changed the inappropriate expression “The designing proposal of” (**page 5, line 94**) to “The designing scheme of” in the revised manuscript.

In the revised manuscript, many grammar errors and inappropriate expressions have been highlighted and corrected to improve the quality of the article, specifically listed in the checklist of manuscript (Page 5, line 88; Page 5, line 90; Page 5, line 94).

Revision checklist of manuscript:

1. In the title, the words “*hybrid perovskite*” were changed to “*hybrid metal halide*”.
2. All the words “*perovskite*” were changed to “*hybrid metal halide*”.
3. All the words “*lp/mm*” were changed to “*lp mm⁻¹*”.
4. All the words “*hydrogen content*” were changed to “*hydrogen density*”.
5. All the words “*80% (400-760 nm)*” were changed to “*70% (500-800 nm)*”.
6. All the words “*branch*” were changed to “*branched*”.
7. To limit the word count to within the 150 words, we have revised the abstract. The specific data remains unchanged, except that the transmission rate data was changed from “*The prepared large-area high quality glassy scintillation screen exhibits high visible light transmittance (> 80%)*” to “*The large-area screen boasts >70% visible light transmittance (500-800 nm)*”, and the PLQY data was changed from “*excellent photoluminescence quantum yield (PLQY) (~ 84.32%)*” to “*a high photoluminescence quantum yield (~85.54%)*”. We added a comparison of light output with ZnS (Ag): PP, stating that it has “*threefold higher light output than commercial ZnS (Ag): PP screens.*” Additionally, we added the spatial resolution data of existing scintillators, noting that “*it surpasses existing scintillators*”. The subsequent data has also been adjusted accordingly.
8. Page 2, line 25, the sentence “*fast neutrons possess a unique advantage in isotope discrimination and radioactive material detection*” was changed to “*fast neutrons binding detectors and related analytical methods offer distinct advantages in isotope differentiation and radioactive material detection*”.
9. Page 3, line 48, “*Ref 27*” was added.
10. Page 3, line 49, the words “*hybrid perovskite*” were changed to “*hybrid metal halide with a perovskite-derived structure*”.
11. Page 3, line 53, the word “*to validate the possibility of*” was deleted.
12. Page 3, line 57, the word “*a disordered*” was changed to “*an amorphous*”.
13. Page 3, line 60, the sentence “*which exhibits high resolution of 5 line pair/mm (lp/mm) which is the highest resolution reported in the field of FNR.*” was changed to “*Furthermore, it achieves a remarkable resolution of 5 line pair mm⁻¹ (lp mm⁻¹), representing the highest resolution reported in the field of FNR.*”

14. Page 3, line 62, the word “*other*” was changed to “*high performance*”.
15. Page 5, line 85, the word “*presented*” was changed to “*captured*”.
16. Page 5, line 88, the word “*seriously*” was deleted.
17. Page 5, line 92, the sentences “*The designing proposal of a transparent fast neutron scintillation screen needs to meet three criteria: (1) high transparency to prevent light scattering interference, (2) high hydrogen density to interact with fast neutrons and produce recoil protons, and (3) high quantum yield to ensure bright luminescence. Under this guidance, we designed*” were changed to “*The design of a transparent fast neutron scintillation screen must meet the following four criteria: (1) high hydrogen density to interact with fast neutrons and produce recoil protons, (2) high PLQY to ensure bright luminescence, (3) high transparency to reduce light scattering, thereby enhancing light extracting efficiency and (4) mechanical robustness to support practical usability in FNR. Guided by these criteria, we developed*”.
18. Page 5, line 98, the sentence “*To obtain the target material, we utilized the high-steric-hindrance cations $BTPP^+$ as the main monovalent A^+ cation, which aids in the formation of a transparent medium during quenching.*” was changed to “*To achieve the target material, we utilized $BTPP^+$ as the main monovalent A^+ cation due to its high steric hindrance, which facilitates the formation of a transparent medium during quenching while also providing interaction sites for fast neutrons.*”.
19. Page 5, line 101, “Ref 29” was added.
20. Page 5, line 104, the words “*and its hydrogen density*” were added.
21. Page 5, line 106, the sentence “*The $[MnBr_4]^{2-}$ serves as the luminescent center that efficiently transfers the recoil protons and excites them to emit visible light.*” was changed to “*The recoil protons deposited energy on the luminescent center $[MnBr_4]^{2-}$, exciting it to emit visible light.*”.
22. Page 5, line 110, the paragraph “*To solve the high hydrogen density requirements of fast neutron scintillation screen, we calculated the hydrogen density, elastic scattering cross section, and fast neutron stopping power of the transparent scintillation screens with various doping components, based on their composition and density. ZnS (Ag): PP scintillators and other commonly used scintillators were used as references (Supplementary Table 1). The results show that the hydrogen density, elastic scattering cross section and fast neutron stopping power of the transparent scintillator with different components ranged from 68.92 to 87.09 kg m⁻³, 0.2698*”.

to 0.3052 cm^{-1} , and 2.662% to 3.006%, respectively. These values are comparable to those of stilbene (67.33 kg m^{-3} , 0.2690 cm^{-1} and 2.553 %) and anthracene (71.91 kg m^{-3} , 0.3100 cm^{-1} and 2.982%), but are lower than ZnS (Ag): PP (136.9 kg m^{-3} , 0.4556 cm^{-1} and 4.454%). For a series of transparent scintillation screens, we observe that increasing the length of the doped branched alkyl chain resulted in higher hydrogen density, which subsequently enhanced the scattering cross section and improved the fast neutron stopping ability.” was added.

23. Page 6, line 125, the word “perovskite” was deleted.
24. Page 7, line 129, the sentence “*b, The XRD of quenching scintillation screens show amorphous transparent glassy state, (i): (BTPP)₂MnBr₄, (ii): (BTPP)_{1.8}(HTPP)_{0.2}MnBr₄, (iii): (BTPP)_{1.8}(DTPP)_{0.2}MnBr₄, (iv): (BTPP)_{1.8}(CTPP)_{0.2}MnBr₄ transparent scintillation screen, respectively, while (i)-polycrystal by slow annealing method showing polycrystalline state, basically consistent with the simulated XRD pattern.*” was changed to “*b, The PXRD of the (i)-(iv) quenching transparent scintillation screens show amorphous transparent glassy state, while PXRD of (i)-polycrystal by slow annealing method showing polycrystalline state, basically consistent with the SCXRD simulated pattern.*”
25. Page 7, line 137, the sentence “*((i): (BTPP)₂MnBr₄, (ii): (BTPP)_{1.8}(HTPP)_{0.2}MnBr₄, (iii): (BTPP)_{1.8}(DTPP)_{0.2}MnBr₄, (iv): (BTPP)_{1.8}(CTPP)_{0.2}MnBr₄)*” was added.
26. Page 7, line 144, the sentence “*The selection of the optimal ratio is shown in Supplementary Fig. 1.*” was changed to “*The details regarding the selection of the optimal cation and its ratio are shown in Supplementary Fig. 1-3.*”
27. Page 7, line 150, the words “*Supplementary Information*” were changed to “*Methods*”.
28. Page 8, line 163, the words “*the X-ray diffraction (XRD)*” were changed to “*the powder X-ray diffraction (PXRD)*”. And all the words “*XRD*” were changed to “*PXRD*”.
29. Page 8, line 166, the words “*single crystal X-ray diffraction (SCXRD)*” and “*Supplementary Fig. 6*” were added.
30. Page 8, line 179, the words “*the hydrogen density and*” were added.
31. Page 9, line 189, the word “*perovskites*” was deleted.
32. Page 9, line 191, “*Ref 42*” was added.
33. Page 9, line 191, the sentence “*This is because the large Mn-Mn distance caused by the larger cation size results in reduced luminescence performance (Supplementary Fig. 7).*” was deleted.

34. Page 9, line 196, the sentence “*We also obtained the PLQY of the (BTPP)₂MnBr₄ single crystal and polycrystal, which are 88.54% and 89.24%, respectively (Supplementary Fig. 9).*” was added.
35. Page 12, line 267, the sentence “*as well as the atomic distance between the specific P-P. Compared to the single crystal and polycrystalline state, the P-P distance of glassy state and molten state fluctuates significantly, disrupting the parallelogram with approximately equal opposite sides.*” was deleted.
36. Page 13, line 294, the sentence “*The distance variation between specific P-P can be observed, and compared with the single crystal and polycrystalline state, both the molten and glassy state exhibit substantial disorder. The difference in P-P distance on opposite sides increases taking the P-P distance of single crystal as the benchmark, disrupting the original approximate parallelogram. The disordered organic cations and distorted tetrahedra diminish the rigidity of the system, leading to an augmentation in non-radiative relaxation processes within the glassy state.*” was deleted.
37. Page 16, line 361, the sentence “*The neutron flux in the experiment is about 10^7 n cm⁻² s⁻¹ with an average energy of 2.69 MeV. A boron¹⁰(B¹⁰) filter was utilized to remove thermal and epithermal neutrons with energy below 0.01 MeV (Supplementary Fig. 20).*” was added.
38. Page 16, line 367, the paragraphs “*The superior performance of (BTPP)_{1.8}(HTPP)_{0.2}MnBr₄ is primarily attributed to its amorphous structure induced high transparency. While ZnS (Ag): PP, (an opaque mixture of polymer and powder), exhibits severe light scattering, resulting in a limited optical absorption length of 0.142 mm. As a result, only photons generated within the superficial 0.142 mm layer are effectively captured by the detector. In contrast, high transparency (BTPP)_{1.8}(HTPP)_{0.2}MnBr₄, has an optical absorption length of approximately 15.9 mm, which allows the majority of emitted photons to reach the detector⁵⁶ (Supplementary Fig. 21).*” and “*Furthermore, we utilized Geant4 simulations to model the physical interactions between neutrons and the scintillation screen, providing detailed energy data at each stage of the interaction process. Through analysis of the entire scintillation and energy transfer process, we identified the reasons behind the high light output of (BTPP)_{1.8}(HTPP)_{0.2}MnBr₄. (Supplementary Table 3). The results reveal that the neutron energy deposition efficiency of (BTPP)_{1.8}(HTPP)_{0.2}MnBr₄ (1.116%) is lower than that of ZnS (Ag): PP (2.137%), primarily due*

to difference in hydrogen density. Moreover, a large number of recoil protons and a small amount of secondary γ -rays were observed in the simulated reaction channel (Supplementary Fig. 22-23), but the production of β -rays was not detected. The recoil proton energy deposition efficiency (calculated relative to neutron deposition energy) is 77.35% for $(BTPP)_{1.8}(HTPP)_{0.2}MnBr_4$ and 84.05% for $ZnS (Ag): PP$, respectively. Consequently, in the neutron response process, the primary difference between $(BTPP)_{1.8}(HTPP)_{0.2}MnBr_4$ and $ZnS (Ag): PP$ lies in neutron deposition which has a greater impact on the overall performance, whereas the effect of recoil protons is relatively small. It is noted that, both $(BTPP)_{1.8}(HTPP)_{0.2}MnBr_4$ and $ZnS (Ag): PP$ scintillators were modeled as homogeneous single component structures due to theoretical model limitations, representing idealized values. This modeling does not account for the advantages of a single component over physical mixing. Notably, the PLQY of the $(BTPP)_{1.8}(HTPP)_{0.2}MnBr_4$ is 85.54%, approximately twice that of $ZnS (Ag): PP$ (31.38%) (Supplementary Table 4). From a theoretical perspective, this higher PLQY offsets the impact of neutron deposition on light output. This suggests that the light output of $(BTPP)_{1.8}(HTPP)_{0.2}MnBr_4$ is determined by its superior photon statistics, stemming from its higher transparency and more efficient photon transmission (Supplementary Table 4). Consequently, the light output of $(BTPP)_{1.8}(HTPP)_{0.2}MnBr_4$ surpasses that of $ZnS (Ag): PP$ were added.

39. Page 16, line 374, “Ref 56” was added.
40. Page 17, line 404, the sentence “A standard steel slit plate (Supplementary Fig. 18) was used to test the resolution of the scintillation screen.” was changed to “a standard steel slit plate (Supplementary Fig. 24-26) and modulation transfer function (MTF) (Supplementary Fig. 27-38) to test the resolution of the scintillation screen.”.
41. Page 18, line 422, the sentences “Then the resolution decreases with the increase of the thickness (from 2.8 mm to 3.8 mm). At a thickness of 3.8 mm, the MTF value dropped to 2.1 lp mm^{-1} , corresponding to a spatial resolution of 2.8 lp mm^{-1} , but is still higher than that of $ZnS (Ag): PP$ (Supplementary Fig. 28).” was added.
42. Page 18, line 427, the words “within 2.8 mm range” were added.
43. Page 18, line 431, the sentence “keeps high resolution consistently, regardless of thickness variations that significantly outperforms the commercial $ZnS (Ag): PP$ ” was changed to

“maintains high resolution significantly surpassing that of the commercial ZnS (Ag): PP”.

44. Page 19, line 460, the paragraph *“Given its excellent neutron imaging performance, we further investigated the stability of the transparent scintillation screen. Based on the actual neutron test environment, we monitored the stability of the storage conditions (nitrogen at 25°C) and fast neutron imaging conditions (air at 25°C and 25% humidity) of (BTPP)_{1.8}(HTPP)_{0.2}MnBr₄, respectively (Supplementary Fig. 33-34). The results show that the transparent scintillation screen retains 99.88% of its initial quantum efficiency after 90 days of storage in a nitrogen. However, in actual experiments, the scintillation screen is used for brief periods, allowing it to be stored in nitrogen and exposed only during use. Meanwhile, we also discussed the impact of humidity on transparency. (Supplementary Fig. 35-36). (BTPP)_{1.8}(HTPP)_{0.2}MnBr₄ maintained its transparency for nearly 30 days at 75% humidity. Even became opaque, it can be restored to its original PLQY by reheating and melting, making it practical for long-term use (Supplementary Fig. 37). Furthermore, (BTPP)_{1.8}(HTPP)_{0.2}MnBr₄ also exhibits good fast neutron irradiation stability, the light output intensity remains at around 98.3% of its initial value after 120 minutes of continuous irradiation (Supplementary Fig. 38), which can meet the normal requirements of FNR. Overall, considering the operating and storage conditions, the stability of the transparent (BTPP)_{1.8}(HTPP)_{0.2}MnBr₄ scintillation screen can meet the standard requirements of fast neutron imaging application.”* was added.
45. Page 20, line 488, the words *“with high light output”* were changed to *“and high light output (three times as much as commercial ZnS (Ag): PP).”*.
46. Page 25, line 633, the sentence *“All preparation operations were carried out in a nitrogen-filled glove box.”* was added.
47. Page 25, line 638; Page 25, line 653, the sentence *“Pour it into a prepared silicone mold and quickly cool to RT.”* was changed to *“It is poured into a silicon mold pre-heated to the same temperature as the melt. The silicone mold filled with melt is then removed from the heating stage and placed on the tabletop for natural cooling to RT. This cooling process typically took 1-2 minutes.”*.
48. Page 25, line 648, the sentence *“All preparation operations were carried out in a nitrogen-filled glove box.”* was added.
49. Page 26, line 667, the sentence *“SCXRD was obtained by single crystal X-ray diffractometer*

- (XtaLAB PRO 007HF(Mo)) (Rigaku, Japan).*” was added.
50. Page 27, line 691, the sentence *“The average energy of the fast neutron spectra is 2.69 MeV, produced by thermal neutron induced U^{235} fission.”* was added.
51. Page 27, line 696, the sentences *“In practical applications of fast neutron imaging, the operating temperature is maintained between 23-28°C, while the relative humidity is kept between 20% and 30% to prevent influenced caused by excessive moisture and heat.”* were added.
52. Page 28, line 732, the paragraphs *“Calculation of fast neutron elastic scattering cross-section. The fast neutron reaction cross-sections were calculated and compared by combining the TENDL-2023 database with the energy spectra of the fast neutrons used in the experiment (average energy: 2.69 MeV).”* were added.
53. Page 28, line 737, the paragraphs *“Fast neutron theory calculation Methods. In order to investigate the physical processes of interaction, Geant4 software was used to perform theoretical calculations and simulations of the entire interaction process between fast neutrons and the sample. The physical model used for simulation is FTFP_SERT_HP. During these calculations, the number of neutrons emitted was 10^7 , and the average energy of incident neutrons was 2.69 MeV.”* were added.
54. Page 29, line 772, the sentence *“We thank Dr. Hongchao Yang and Professor Qibiao Wang (Sichuan University of Light Chemical Technology, SUSE) for the helpful support and analysis of simulation calculation of fast neutron physics processes.”* was added.

Revision checklist of Supplementary Information:

1. Page 2-3, Supplementary Table 1, Supplementary Fig. 1-2 and the related texts were added.
2. Page.3, “*Ref 1*” was added.
3. Page 4, Supplementary Fig. 3 has been updated to include error bars.
4. Page 6, the word in Supplementary Table 2 was changed from “*XRD*” to “*Single crystal X-ray diffraction (SCXRD)*”.
5. Page 7, Supplementary Fig. 6 and the related texts were updated.
6. Page 8, Supplementary Fig. 9 has been updated to include error bars.
7. Page 14-18, Supplementary Fig. 20-23, Supplementary Table 3-4 and the related texts were added.
8. Page.14, “*Ref 5*” was added.
9. Page 20-21, Supplementary Fig. 27-28 and the related texts were added.
10. Page.21, “*Ref 6*” was added.
11. Page 25-27, Supplementary Fig. 33-38 and the related texts were added.

Dear editor,

Thank you for your letter and the reviewers' insightful comments concerning our manuscript entitled "A transparent hybrid perovskite glassy scintillation screen for high-resolution fast neutron radiography" (NCOMMS-24-30110-T). We sincerely appreciate the time and effort that Reviewers #1-4 have dedicated to evaluating our work and providing constructive suggestions to further improve the clarity and scientific impact of our study. We have evaluated the comments carefully and made revisions according to these suggestions. Thank you for your feedback and for forwarding the reviewers' additional comments regarding our revised manuscript. Revised portions are highlighted in yellow in the manuscript and Supplementary Information. The main revisions and the responses to the reviewers' comments are as follows:

Reviewer #1 (Remarks to the Author):

I am satisfied with the author's answer and recommend that this article be accepted.

Response: Thank you very much for the positive comment of this work.

Reviewer #4 (Remarks to the Author):

This study demonstrates an innovative application of hybrid halide glass scintillators in fast neutron radiography by suppressing light scattering through amorphous glass-state design. While the suppression of crystal boundary scattering using glassy materials has been well-established in X-ray imaging (as acknowledged in the response letter), the current work needs to better express its scientific breakthrough beyond technology transfer. So I don't think this work meet the criteria of Nature Communication.

Response:

We sincerely appreciate the insightful comments from the reviewer and revise our manuscript accordingly. Below, we address each point systematically:

Firstly, we would like to clarify that this study does not simply involve a direct "technology transfer" of glass-state scintillators from X-ray imaging to fast neutron imaging. Instead, we have achieved original advancements in multiple aspects, including **material design, energy transfer mechanisms, imaging mechanisms, and performance optimization**. These innovations are fundamentally driven by the unique requirements of fast neutron imaging.

Fast neutron imaging and X-ray imaging are fundamentally different in their physical principles, leading to distinct scintillator design criteria. X-ray imaging is based on photon absorption and optical transmission, where resolution is primarily limited by photon scattering and absorption within the scintillator, and the performance of the optical imaging system (*Adv. Mater.* 2024, 36, 2309588). The application of glass-state scintillators in X-ray imaging primarily enhances imaging resolution by reducing grain boundary light scattering, with the highest reported resolution reaching 20-30 lp mm⁻¹ (*Adv. Mater.* 2022, 34, 2110420; *J. Am. Chem. Soc.* 2024, 146, 7373–7385; *ACS Nano* 2024, 18, 16715–16725; *Angew. Chem. Int. Ed.* 2025, e202425661). In contrast, fast neutron imaging depends on recoil protons generated by fast neutrons-hydrogen nuclei elastic scattering. Its imaging signal is not only influenced by optical transmission effects but also by the energy deposition characteristics of neutrons within scintillator. Consequently, the maximum resolution achieved in fast neutron imaging is only 5 lp mm⁻¹ (**Table R1, Fig. 5f, Supplementary Table 5**), which is the highest resolution so far as we know. This pronounced resolution gap highlights that the limiting factors in fast neutron imaging extend beyond light scattering and are primarily governed by the fundamental interaction mechanisms between neutrons and scintillators, which differ significantly from those in X-ray imaging. Recognizing these intrinsic differences, we have developed a tailored design strategy for glass-state scintillators in fast neutron imaging, rather than simply adapting X-ray scintillator technologies. By systematically tuning the A-site cation composition, we optimize the scintillator's fast neutron response characteristics, ultimately enhancing imaging performance and addressing the specific demands of fast neutron detection.

In response to the characteristics of fast neutron imaging, our work has made scientific innovations and advancement in the following aspects:

1. We have proposed and validated the application of transparent glass-state hydrogen-rich scintillators in fast neutron imaging for the first time, which constitutes a significant breakthrough in this field. The light output of our glassy scintillator (BTTP)_{1.8}(HTTP)_{0.2}MnBr₄ is 3 times that of commercial scintillation screen ZnS(Ag):PP (**Fig. 5b**), and it achieves a spatial resolution that of 5 lp mm⁻¹ (**Fig. 5d-e**), far surpassing that of the ZnS(Ag):PP (2 lp mm⁻¹) and the highest-resolution fast neutron scintillation screen reported to date (PVT (2% Ir-complex) at 2.56 lp mm⁻¹ (*Nucl. Instrum. Methods Phys. Res., Sect. A* 2022, 1027, 166331)) (**Table R1, Fig. 5f, Supplementary Table 5**). These experimental results further validate the advantages of the glassy fast neutron

scintillator and demonstrate that it is not simply a technology transfer from X-ray imaging, but represents an innovation in fast neutron imaging.

2. In addition to material design, we also conducted an in-depth investigation into the physical interaction mechanisms between glassy scintillators and fast neutrons during the imaging process. By leveraging the Evaluated Nuclear Data File (ENDF) database and Geant4 simulations, we systematically analyzed multiple critical parameters, including fast neutron scattering cross-section, neutron energy deposition efficiency, recoil proton energy deposition efficiency, and the effects of material absorption length and transparency on light attenuation and light extraction efficiency (**Supplementary Fig. 21-23, Supplementary Table 1, 3-4**). This comprehensive analysis provides new insights into why transparent hydrogen-rich scintillators exhibit superior light extraction efficiency for fast neutron detection. Furthermore, through experimental investigations coupled with *ab initio* molecular dynamics (AIMD) simulations (**Fig. 4, Supplementary Fig. 17-19**), we investigated the formation mechanism of $(\text{BTPP})_{1.8}(\text{HTPP})_{0.2}\text{MnBr}_4$ glassy and polycrystalline states, discussed the essence of glassy structure formation, and the differences in optical transmission performance between glassy and polycrystalline states, demonstrating its superior optical transmission capability. Our findings indicate that the performance of fast neutron scintillators is governed not merely by their intrinsic fluorescence properties but by a complex interplay of neutron energy deposition efficiency, energy transfer efficiency, material physical state, and light extraction efficiency. These results reinforce the originality of our approach and distinguish our work from conventional X-ray scintillator research.

In summary, our research has achieved several key advancements in the field of fast neutron imaging. We have proposed and verified the application of transparent glassy hydrogen-rich scintillators in fast neutron imaging for the first time, significantly enhancing the light extraction efficiency and imaging clarity of fast neutron imaging. By systematically elucidating the physical mechanisms governing the interaction between $(\text{BTPP})_{1.8}(\text{HTPP})_{0.2}\text{MnBr}_4$ and fast neutrons, we provide valuable scientific insights into fast neutron detection. Moreover, the experimental data fully demonstrate the superior performance of our material, laying a solid foundation for its application as the next-generation fast neutron scintillator.

Therefore, we firmly believe that our research represents an original contribution in the areas of fast neutron imaging mechanisms, scintillator material design, and optical performance

optimization, rather than a simple extension of X-ray scintillator technology.

Also, the following issues require clarification:

1. The high imaging resolution stems from the intrinsic advantage of glassy-state materials. So are there other glassy materials used for neutron imaging? A direct comparison with previously reported amorphous scintillators would better position the innovation.

Response 1:

We appreciate the reviewer's comments. While glassy materials are known to offer clear advantages for high-resolution fast neutron imaging, research on glassy scintillator materials in this field are scarcely reported. To better highlight the novelty of this study, we have incorporated a direct comparison with previously reported other types of fast neutron scintillation screens (**Table R1**). As shown in the comparative data, research on fast neutron scintillators has predominantly focused on **powder-polymer composites, organic scintillators, plastic scintillators, and single-crystal scintillators**, with scarcely any reports on glassy materials. This is primarily due to the established technical pathways and the limitations of conventional material systems in fast neutron detection.

Firstly, powder-polymer composites (such as ZnS (Ag): PP) are plagued by severe interface scattering due to the intermingling of internal particles and polymers, which causes multiple scattering and absorption of light within the material, thereby restricting the spatial resolution. The typical resolution is merely $\leq 2 \text{ lp mm}^{-1}$ (*J. Imaging* 2017, 3, 60). Furthermore, the reported nanocrystal suspensions also face challenges regarding low optical homogeneity and surface-state induced losses, achieving only $0.1\text{-}0.185 \text{ lp mm}^{-1}$, which is inadequate for high-resolution imaging (*ACS Nano* 2020, 14, 14686–14697; *ACS Energy Lett.* 2021, 6, 4365–4373).

Secondly, plastic scintillators were fabricated by doping fluorescent molecules into polymer matrices, but suffer from dopant distribution inhomogeneity and refractive index mismatches, leading to severe interface scattering and lowering the resolution. Simultaneously, fluorescent molecules require low dopant concentrations, which limits light output. For example, PVT with 2% Ir-complex achieves a spatial resolution of 2.56 lp mm^{-1} (*Nucl. Instrum. Methods Phys. Res., Sect. A* 2022, 1027, 166331), but it suffers from inefficient exciton transfer. Similarly, the TPE-4Br@PVT system reaches 2.03 lp mm^{-1} (*Adv. Funct. Mater.* 2025, 2503688), but does not overcome the core

resolution bottleneck.

In the aspect of single crystal metal halide scintillators, the reported single-component transparent two-dimensional (2D) single-crystal perovskites (such as $(\text{PEA})_2\text{PbBr}_4$, $\text{Li}-(\text{PEA})_2\text{PbBr}_4$) have exhibited a spatial resolution of 1-2 lp mm^{-1} in the application of fast neutron imaging (*InfoMat.* 2024, e12648; *ACS Appl. Opt. Mater.* 2023, 1, 1856–1861). Nevertheless, these single crystals still present certain limitations: their quantum efficiency is generally low due to the strong exciton binding effect in 2D perovskites (*J. Am. Chem. Soc.* 2021, 143, 21302–21311; *Nat. Mater.* 2018, 17, 550–556). Meanwhile, challenges in large-area fabrication due to the constraints of single-crystal growth make them impractical for large-area fast neutron imaging. Similarly, organic scintillators (like stilbene and anthracene) are also limited by manufacturing conditions, size, light output and sensitivity to storage conditions (*InfoMat.* 2022, 4, e12325).

Unlike these conventional materials, our study introduces and validates the first single-component glass-state scintillator for fast neutron imaging: $(\text{BTPP})_{1.8}(\text{HTPP})_{0.2}\text{MnBr}_4$. The single-component glassy amorphous structure effectively eliminates grain boundary scattering, fundamentally reduces light loss, and combines the advantages of large-area processability, high transparency, and high light output. As a result, it stands as the only material currently capable of achieving a spatial resolution of 5 lp mm^{-1} in fast neutron imaging, significantly outperforming conventional scintillator systems. This research thus addresses the absence of glassy scintillator materials in fast neutron imaging and introduces a novel material platform for high-resolution neutron detection.

This comparison (**Figure R1** and **Table R1**) has been updated in the revised manuscript (**Fig. 5f**) and Supplementary Information (**Supplementary Table 5**).

Table R1 Performance comparison of glassy (BTTP)_{1.8}(HTPP)_{0.2}MnBr₄ and reported fast neutron scintillation screens.

Scintillation screen	Type	Screen component	Hydrogen density (n cm ⁻³)	PLQY (%)	Thickness (mm)	Neutron light output* (photons/neutron)	Spatial resolution (lp mm ⁻¹)
WFC (Waveshift Fiber Converter) ¹	Opaque screen	two-component	-	-	10	-	0.5
PZC (Polyethylene-ZnS Converter) ¹	Opaque screen	two-component	-	-	3	-	> 0.5
B14E MCPs (Microchannel Plates) ²	Opaque screen	two-component	-	-	-	-	2.5
ZnS (Ag): PP (RC TRITEC, Swiss) ³	Opaque screen	two-component	-	-	1.5	-	2
Stilbene ⁴	Organic	single-component	4.05 × 10 ²²	65.00	-	~ 26.51% of the ZnS: Cu(PP)	-
Anthracene ⁴	Organic	single-component	4.32 × 10 ²²	64.00	-	~ 37.88% of the ZnS: Cu(PP)	-
BCF-12 (Crytur) ⁵	Plastic fiber screen	two-component	-	-	50	-	0.5-0.7
PVT (2% X-Flrpic) (Polyvinyl Toluene) ⁶	Plastic	two-component	-	-	3	-	2.28
PVT (2% Ir-complex) ⁷	Plastic	two-component	-	-	3	-	2.56
TPE-4Br@PVT ⁸	@PVT plastic	two-component	5.29 × 10 ²²	53.30	3	-	2.03
CsPbBrCl ₂ : Mn in hexane ⁹	Nanocrystal solution	two-component	-	53.00	10	11.24% of the ZnS: Cu(PP)	0.185
FAPbBr ₃ in toluene ¹⁰	Nanocrystal solution	two-component	-	96.20	10	19.3% of the ZnS: Cu(PP)	0.1
(BA) ₂ PbBr ₄ ¹¹	@PDMS film	two-component	4.82 × 10 ²²	-	2.5	86% of the ZnS(Ag) ⁶ LiF	1
Mn-STA ₂ PbBr ₄ ¹²	Self-standing plate	single-component	9.51 × 10 ²²	58.58	1.135	79.05% of the ZnS(Ag):PP	0.5
(PEA) ₂ PbBr ₄ ¹³	Single crystal	single-component	1.77 × 10 ²¹	-	1.9	-	2
Li-(PEA) ₂ PbBr ₄ ¹⁴	Single crystal	single-component	-	-	1	-	1
LiInSe ₂ ¹⁵	Crystal	single-component	-	-	0.528	-	0.32
(BTTP) _{1.8} (HTPP) _{0.2} MnBr ₄	Transparent glass state	single-component	4.18 × 10 ²²	85.54	1.8	~ 306% of the ZnS(Ag):PP	5
					2.8	~ 531% of the ZnS(Ag):PP	5

*: The fast neutron light output of ZnS (Ag): PP is about 78.90%-84.21% of ZnS (Cu): PP (*J. Imaging* 2017, 3, 60).

Figure R1 Comparison of the thickness and spatial resolution of (BTTPP)_{1.8}(HTPP)_{0.2}MnBr₄ transparent scintillation screen and several reported fast neutron scintillation screens.

2. I don't see much improvement of the doping of HTPP⁺ while using for neutron imaging, as neither light output (Fig.5b) nor transparency (Fig.2f) shows measurable enhancement compared to undoped (BTTPP)₂MnBr₄. The claimed benefits of HTPP⁺ doping require stronger experimental verification including stability tests of pure (BTTPP)₂MnBr₄ under neutron irradiation and humidity.

Response 2:

Thank you for the comments provided by the reviewer. We fully comprehend the reviewers' concerns regarding the functional positioning of HTPP⁺ doping. Below, we systematically clarify its purpose and benefits. The primary objective of incorporating HTPP⁺ doping in this research is not solely to enhance PLQY or transparency but rather to optimize the overall performance balance of (BTTPP)₂MnBr₄ in response to key challenges encountered in practical applications.

Firstly, although pure (BTTPP)₂MnBr₄ exhibits good optical properties, it suffers from high brittleness, making it prone to cracking during imaging experiments. The introduction of long branched alkyl chain cations (such as HTPP⁺, DTPP⁺, CTPP⁺) serves as an effective strategy to enhance mechanical toughness while also increasing hydrogen density, thereby improving fast neutron capture efficiency. Nevertheless, experiments have indicated that the incorporation of long-chain cations might result in a reduction in luminescence performance and transparency (**Fig. 2e**, **Supplementary Fig. 1-3**). Through systematic screening of cation types and doping ratios, we

identified (BTPP)_{1.8}(HTPP)_{0.2}MnBr₄ as an optimized composition, achieving multi-dimensional performance improvements while maintaining a high PLQY (85.54%) and optical transparency (> 70%). Specifically:

Upon doping with HTPP⁺, the hydrogen density of the material rose from 68.92 kg m⁻³ to 69.40 kg m⁻³ (**Supplementary Table 1**), thereby augmenting the fast neutron capture efficiency. Benefiting from this, at the identical thickness (1.65 mm), the light output of the HTPP⁺ doped system was 3 times that of the commercial ZnS (Ag): PP, surpassing that of pure (BTPP)₂MnBr₄ (2.8 times that of the commercial ZnS (Ag): PP). Additionally, the elastic modulus decreased from 1.65 MPa to 0.96 MPa (**Fig. 2c, Supplementary Fig. 7-8**), substantially improving material flexibility and durability.

In response to the stability concerns raised by the reviewers, we further supplemented tests on radiation resistance stability and humidity stability.

Irradiation stability test: After 10 days of continuous irradiation (fast neutron flux: 10⁷ n cm⁻² s⁻¹), the light output of (BTPP)_{1.8}(HTPP)_{0.2}MnBr₄ remained at 84.58% of the initial level, superior to 80.24% of pure (BTPP)₂MnBr₄ (**Figure R2**), indicating that the scintillation screen exhibits excellent irradiation stability, and further enhanced by HTPP⁺ doping.

Humidity stability test: HTPP⁺ incorporation inhibits crystallization by increasing steric hindrance (**Fig. 3b, Supplementary Fig. 14**) and enhances intermolecular interactions. After 15 days of exposure to 25%, 50%, and 75% humidity at 25°C, (BTPP)_{1.8}(HTPP)_{0.2}MnBr₄ remained transparent, with crystallization observed only after 30 days (**Supplementary Fig. 36**). In contrast, pure (BTPP)₂MnBr₄ gradually crystallized and became opaque after just 10 days at 75% humidity (**Figure R3**).

In conclusion, the incorporation of HTPP⁺ not only maintains the original transparency and PLQY, but also evidently enhances hydrogen density, flexibility, light output, and stability, ensuring greater long-term usability and processing reliability in practical fast neutron imaging applications.

The irradiation stability data and its related description (**Figure R2**) have been replaced in the revised Supplementary Information (**page 28, Supplementary Figure 38**) and the revised manuscript (**page 19**). The humidity stability photos (**Figure R3**) have been added to the revised Supplementary Information (**page 27, Supplementary Figure 35**) and the revised manuscript (**page 19**).

Figure R2 The stability of ZnS (Ag): PP, (BTPP)₂MnBr₄ and (BTPP)_{1.8}(HTPP)_{0.2}MnBr₄ scintillation screens under continuous fast neutron irradiation (fast neutron flux: 10^7 n cm⁻² s⁻¹).

Figure R3 Photos of changes in (BTPP)₂MnBr₄ (undoped) and (BTPP)_{1.8}(HTPP)_{0.2}MnBr₄ (doped) under different humidity environments.

Reference

1. Wu, Y. et al. A preliminary study of the performance for high energy neutron radiography converter. *Nucl. Tech.* **43**, 070203 (2020).
2. Wang, W. et al. Experimental study of spatial resolution of MCPs for compact high-resolution neutron radiography system. *Nucl. Instrum. Methods Phys. Res., Sect. A* **1050**, 168179 (2023).
3. Malgorzata, M. et al. Performance of the commercial PP/ZnS:Cu and PP/ZnS:Ag scintillation screens for fast neutron imaging. *J. Imaging* **3**, 60-72 (2017).

4. Xia, M. et al. Organic–inorganic hybrid perovskite scintillators for mixed field radiation detection. *InfoMat* **4**, e12325 (2022).
5. Zboray, R. et al. High-frame rate imaging of two-phase flow in a thin rectangular channel using fast neutrons. *Appl. Radiat. Isot.* **90**, 122-131 (2014).
6. Oksuz, I. et al. *Characterization of polyvinyl toluene (PVT) scintillators for fast neutron imaging* (San Diego, CA, 2018).
7. Oksuz, I. et al. Quantifying spatial resolution in a fast neutron radiography system. *Nucl. Instrum. Methods Phys. Res., Sect. A* **1027**, 166331 (2022).
8. He, S. et al. Hot exciton-based plastic scintillator engineered for efficient fast neutron detection and imaging. *Adv. Funct. Mater.*, 2503688 (2025).
9. Montanarella, F. et al. Highly concentrated, zwitterionic ligand-capped $\text{Mn}^{2+}:\text{CsPb}(\text{Br}_x\text{Cl}_{1-x})_3$ nanocrystals as bright scintillators for fast neutron imaging. *ACS Energy Letters* **6**, 4365-4373 (2021).
10. McCall, K. M. et al. Fast neutron imaging with semiconductor nanocrystal scintillators. *ACS Nano* **14**, 14686-14697 (2020).
11. Shao, W. Y. et al. Synergy of organic and inorganic sites in 2D perovskite for fast neutron and X-ray imaging. *Adv. Funct. Mater.* **33**, 2301767 (2023).
12. Zheng, J. X. et al. Hydrogen-rich 2D halide perovskite scintillators for fast neutron radiography. *J. Am. Chem. Soc.* **143**, 21302-21311 (2021).
13. Yang, B. et al. Inch-sized 2D perovskite single-crystal scintillators for high-resolution neutron and X-ray imaging. *InfoMat*, e12648 (2024).
14. Yan, W. et al. Organic–inorganic hybrid perovskite scintillator for neutron and γ -ray detection. *ACS Appl. Opt. Mater.* **1**, 1856-1861 (2023).
15. Lukosi, E. et al. First evaluation of fast neutron imaging with LiInSe_2 semiconductors. *Nucl. Instrum. Methods Phys. Res., Sect. A* **976**, 164254 (2020).

Revision checklist of manuscript:

1. Page 14, line 325, Fig. 5f was updated.
2. Page 15, line 341, the figure caption “f, Comparison of the thickness and spatial resolution of

(BTTPP)_{1.8}(HTPP)_{0.2}MnBr₄ transparent scintillation screen and several reported fast neutron scintillation screens” was updated.

3. Page 15, line 343; Page 18, line 431, “*Ref 53,56-61*” were added.
4. Page 19, line 466, the sentence “*Meanwhile, we also discussed the impact of humidity on transparency (Supplementary Fig. 35-37).*” was updated.
5. Page 19, line 468, the sentence “*(BTTPP)_{1.8}(HTPP)_{0.2}MnBr₄ maintained its transparency for nearly 30 days at 75% humidity, but (BTTPP)₂MnBr₄ gradually crystallized after 10 days at 75% humidity*” was added.
6. Page 19, line 469, the sentence “*Even became opaque, it can be restored to its original PLQY by reheating and melting, making it practical for long-term use (Supplementary Fig. 37). Furthermore, (BTTPP)_{1.8}(HTPP)_{0.2}MnBr₄ also exhibits good fast neutron irradiation stability, the light output intensity remains at around 98.3% of its initial value after 120 minutes of continuous irradiation (Supplementary Fig. 38), which can meet the normal requirements of FNR.*” was changed to “*Furthermore, (BTTPP)_{1.8}(HTPP)_{0.2}MnBr₄ also exhibits good fast neutron irradiation stability (Supplementary Fig. 38), after 10 days of continuous irradiation (fast neutron flux: 10^7 n cm⁻² s⁻¹), the light output of (BTTPP)_{1.8}(HTPP)_{0.2}MnBr₄ remained at 84.58% of the initial level, superior to 80.24% of pure (BTTPP)₂MnBr₄, indicating that the scintillation screen exhibits excellent irradiation stability, and further enhanced by HTPP⁺ doping. Notably, even after became opaque or its performance declines, it can be restored to its original PLQY by reheating and melting, making it practical for long-term use (Supplementary Fig. 39).*”.
7. Page 27, line 703, the sentence “*The humidity stability experiment was conducted in the environmental testing chamber (HK WEWON TECHNOLOGY LIMITED).*” was added.

Revision checklist of Supplementary Information:

1. Page 23, Supplementary Table 5 was updated.
2. Page.23, “*Ref 8, 10, 13-15, 17, 19-21*” were added.
3. Page 27, Supplementary Fig. 35 and the related texts were added.
4. Page 28, the original Supplementary Fig. 38 and the related texts were replaced.

Thank you for your letter and the insightful comments from the reviewers regarding our manuscript titled “A transparent hybrid metal halide glassy scintillation screen for high-resolution fast neutron radiograph” (NCOMMS-24-30110-T). The constructive feedback provided by the expert reviewers is invaluable and has greatly assisted in enhancing the quality of our paper. We have thoroughly evaluated the comments and implemented revisions accordingly. The revised sections are highlighted in yellow within the manuscript. The main revisions in the paper and the responses to the reviewers’ comments are as follows:

Reviewer #4 (Remarks to the Author):

The authors addressed most of my original comments, however, there are some other problems about the effect of the organic cation chain.

Response:

We appreciate reviewer’s very encouraging comments.

Comment 1:

The output of fast neutron imaging performance of $(\text{BTPP})_{1.8}(\text{HTPP})_{0.2}\text{MnBr}_4$ glass depends on light output and photon transmission effect. With the increase of the length of the doped organic cation chain, the PLQY of DTPP^+ and CTPP^+ decreases very fast, while HTPP^+ does not change, please explain the reason.

Response 1:

Thanks to the thoughtful reviewer. With the increase in the length of the doped organic cation chains, the PLQY of HTPP^+ doped remains unchanged, while the PLQY of long-chain cations DTPP^+ and CTPP^+ doped decreases rapidly, which is attributed to the balance among **self-absorption effect, steric hindrance, and the density of luminescent centers**. The following is a detailed explanation combined with multiple characterization methods such as ultraviolet-visible (UV-vis) absorption spectroscopy, photoluminescence (PL) spectroscopy, Raman spectroscopy and literature:

Firstly, through UV-vis absorption spectroscopy and PL spectroscopy tests (**Figure R1**), we observed that as the length of the doped cation branched alkyl chains increases, the absorption intensity of the glass at approximately 365 nm and 450 nm significantly weakens, indicating a decline in the absorption capacity of excitation light, thereby leading to a reduction in excitation

efficiency and a subsequent attenuation of emission intensity. On the other hand, in the pure cationic metal halide system (**Figure R1b**), we discovered that as the length of the cation branched alkyl chains increases, the absorption intensity in the 500-600 nm band gradually enhances, intensifying the self-absorption effect on the green light emission (525 nm) of $[\text{MnBr}_4]^{2-}$, further suppressing the output of effective photons. Consequently, the luminescence intensities of $(\text{DTPP})_2\text{MnBr}_4$ and $(\text{CTPP})_2\text{MnBr}_4$ significantly decrease, which is consistent with the trend of PLQY variations in the manuscript (**Figure 2e, Supplementary Fig. 1**). The photos (**Figure R2**) also exhibit the same pattern: as the length of the doped branched alkyl chains increases, the sample color gradually deepens (**Figure R2, Supplementary Fig. 2**, changing from green to yellow-green under visible light), further indicating the enhancement of the self-absorption effect.

In addition, to further investigate the changes in the luminescent centers, we carried out quantitative Raman spectroscopy tests (**Figures R3-4**), where the peak intensity is positively correlated with the concentration of the corresponding structure. Within the range of 500-3200 cm^{-1} , we detected the vibration modes of organic cations, and in the low wavenumber range of 40-300 cm^{-1} , the Mn-Br characteristic vibration of the $[\text{MnBr}_4]^{2-}$ units was observed (*Chem. Sci.*, 2024, 15, 16338-16346). In **Figures R3b** and **Figures R4**, as the length of the doped branched alkyl chains increases, the vibration mode of the Mn-Br characteristic vibration peak in the low wavenumber region (40-300 cm^{-1}) becomes non-Gaussian gradually, and weakens in intensity. This evolution clearly demonstrates that long-chain cations cause dilution and excessive spatial separation of $[\text{MnBr}_4]^{2-}$ luminescent units, leading to a significant suppression of luminescence efficiency (*Angew. Chem. Int. Ed.*, 2023, 62, e202216504; *Adv. Optical Mater.*, 2022, 2102793; *J. Mater. Chem. C*, 2021, 9, 9952-9961).

Based on this, we systematically analyzed the impact of cation chain length on the material structure and optical properties. For the HTPP⁺ doped samples, the moderate chain length introduced appropriate steric hindrance, which effectively suppressed crystallization (**Figure 3b, Supplementary Fig. 18**) without causing excessive separation of $[\text{MnBr}_4]^{2-}$ units. This facilitated a high-density distribution of luminescent centers and promoted efficient radiative recombination, resulting in a high PLQY of approximately 85.54%. In contrast, the longer alkyl chains in DTPP⁺ and CTPP⁺ induced excessive steric effects and enhanced self-absorption, leading to spatial separation of $[\text{MnBr}_4]^{2-}$ units, reduction of photon transport efficiency, and a significant drop in

PLQY (approximately 68.64% and 56.43%, respectively).

In summary, the PLQY variation arises from the synergistic interplay of steric hindrance, self-absorption, and the spatial distribution of luminescent centers, all governed by the chain length of the organic cations. Among them, HTPP⁺ doped represents an optimal balance, maintaining high luminescence efficiency, while DTPP⁺ and CTPP⁺ doped compromise performance due to their overly extended chains.

In response to these opinions, we have included the revised Supplementary Information (**Supplementary Fig. 9-12, Page 8-11**) and relevant descriptions in the revised manuscript (**Page 9**).

Figure R1 The UV-vis spectra and PL spectra of the (a), (BTTP)₂MnBr₄, (BTTP)_{1.8}(HTPP)_{0.2}MnBr₄, (BTTP)_{1.8}(DTPP)_{0.2}MnBr₄, (BTTP)_{1.8}(CTPP)_{0.2}MnBr₄; (b), (BTTP)₂MnBr₄, (HTPP)₂MnBr₄, (DTPP)₂MnBr₄, (CTPP)₂MnBr₄.

Figure R2 Photos of (BTTP)₂MnBr₄, (HTPP)₂MnBr₄, (DTPP)₂MnBr₄, (CTPP)₂MnBr₄,

(BTTP)_{1.8}(HTPP)_{0.2}MnBr₄, (BTTP)_{1.8}(DTPP)_{0.2}MnBr₄, (BTTP)_{1.8}(CTPP)_{0.2}MnBr₄ (up: visible light; bottom: 365 nm UV light).

Figure R3 (a), Raman spectra (40-3200 cm⁻¹) and (b), Raman spectra of the vibrational characteristics of the Mn-Br (40-300 cm⁻¹) of glassy (BTTP)₂MnBr₄, (BTTP)_{1.8}(HTPP)_{0.2}MnBr₄, (BTTP)_{1.8}(DTPP)_{0.2}MnBr₄, (BTTP)_{1.8}(CTPP)_{0.2}MnBr₄.

Figure R4 Raman spectra of the vibrational characteristics of the Mn-Br (40-300 cm⁻¹) of glassy (BTTP)₂MnBr₄, (HTPP)₂MnBr₄, (DTPP)₂MnBr₄, (CTPP)₂MnBr₄.

Comment 2:

In Figure 2b, further demonstrate that the XRD of the slowly annealed polycrystalline sample i[(BTTP)₂MnBr₄] is consistent with the simulated SCXRD pattern.

Response 2:

We appreciate the reviewer's comments. The structural consistency between the

polycrystalline sample $i[(\text{BTPP})_2\text{MnBr}_4]$ and the single crystal simulated SCXRD was confirmed by comparing their PXRD patterns and unit cell parameters. As shown in the newly added **Figure R5b**, we have enlarged and detailed the comparison between the PXRD of the slowly annealed polycrystalline sample $i[(\text{BTPP})_2\text{MnBr}_4]$ (presented in **Figure 2b** of the manuscript) and the simulated SCXRD pattern. The diffraction peaks of the polycrystalline sample basically correspond to the simulated single-crystal diffraction peaks, with only exhibited certain disparities in the intensity of diffraction peaks. Specifically, some diffraction peaks (such as $2\theta = 8.57^\circ, 8.95^\circ, 10.8^\circ, 17.2^\circ, 21.5^\circ, 21.6^\circ$) are relatively stronger, indicating the preferential crystal growth orientations along the $(00\bar{2})$, $(\bar{1}11)$, (102) , $(00\bar{4})$, $(00\bar{5})$, (204) planes during the crystallization process. Such intensity variations are attributed to the preferred orientation of grains in the powder sample rather than any intrinsic structural difference and are a common phenomenon in powder X-ray diffraction analysis (*ACS Appl. Mater. Interfaces*, 2023, 15, 932-941; *Chem. Eng. J*, 2024, 483, 149239; *J. Mater. Chem. A*, 2022, 10, 15990-15998).

Therefore, the XRD results of the slowly annealed polycrystalline sample $i[(\text{BTPP})_2\text{MnBr}_4]$ are basically agreement with the simulated SCXRD pattern, indicating good structural retention.

In response to these opinions, we have included **Figure R5b** as **Supplementary Fig. 6b**, along with its corresponding description in the revised Supplementary Information (**Page 7**), and added relevant descriptions in the revised manuscript (**Page 11**).

Figure R5 (a), the PXRD of the (i)-(iv) quenching transparent scintillation screens show amorphous transparent glassy state, while PXRD of $(\text{BTPP})_2\text{MnBr}_4$ -polycrystal by slow annealing method showing polycrystalline state, basically corresponds to the SCXRD simulated pattern ((i): $(\text{BTPP})_2\text{MnBr}_4$, (ii): $(\text{BTPP})_{1.8}(\text{HTPP})_{0.2}\text{MnBr}_4$, (iii): $(\text{BTPP})_{1.8}(\text{DTPP})_{0.2}\text{MnBr}_4$, (iv):

(BTTP)_{1.8}(CTPP)_{0.2}MnBr₄). (b), Enlarged and detailed the comparison between the PXRD of the (BTTP)₂MnBr₄-polycrystal and the SCXRD simulation of (BTTP)₂MnBr₄ single crystal.

Comment 3:

Please explain in detail the reasons for the excellent stability of (BTTP)_{1.8}(HTPP)_{0.2}MnBr₄ glass in the high humidity environment. The author explains that it is attributed to the large steric hindrance of organic cations, so does the doping of DTPP⁺ and CTPP⁺ have the same or better effect?

Response 3:

Thank you for the reviewer's careful suggestions. The excellent humidity stability of (BTTP)_{1.8}(HTPP)_{0.2}MnBr₄ glass is primarily attributed to the optimized steric hindrance introduced by HTPP⁺ doping.

Specifically, the moderately long heptyl chain of HTPP⁺, combined with the bulky triphenylphosphine groups, promotes dense molecular packing. This enhanced steric hindrance not only stabilizes the amorphous glass state and inhibits crystallization (**Figure 3b, Supplementary Fig. 18**), but also avoids excessive separation of the luminescent [MnBr₄]²⁻ units, thereby preserving high transparency under humid conditions. As a result, the (BTTP)_{1.8}(HTPP)_{0.2}MnBr₄ glass remains transparent for nearly 30 days at 75% humidity (**Supplementary Fig. 41**). Theoretically, the longer branched alkyl chains of DTPP⁺ and CTPP⁺ could introduce even greater steric hindrance (*Nat. Commun.*, 2020, 11, 3395; *Angew. Chem. Int. Ed.*, 2025, e202504658). To verify this, we conducted comparative humidity stability tests by exposing different doped glass samples at 25°C and 75% humidity for 15 days (**Figure R6**). The results showed that all glass samples doped with long-chain cations, including HTPP⁺, DTPP⁺ and CTPP⁺, remained transparent over the test period, whereas the undoped (BTTP)₂MnBr₄ sample exhibited visible crystallization after 10 days. These findings confirm that doped with long-chain cations effectively enhances humidity resistance, and that DTPP⁺/CTPP⁺ doped glasses achieve a similar stability improvement to that of (BTTP)_{1.8}(HTPP)_{0.2}MnBr₄.

However, in the material selection process for practical applications, stability alone was not sufficient. The priority was placed on simultaneously achieving high PLQY, high light output, and high transparency (i.e., the HTPP⁺ doped system (BTTP)_{1.8}(HTPP)_{0.2}MnBr₄). Therefore, the

DTPP⁺/CTPP⁺ doped samples were not included in the subsequent systematic fast neutron performance tests due to insufficient optical performance.

In summary, the humidity stability of (BTTP)_{1.8}(HTPP)_{0.2}MnBr₄ originates from the optimized steric hindrance provided by the HTPP⁺ cations. While DTPP⁺ and CTPP⁺ doped samples also exhibit similarly enhanced stability, but their insufficient optical properties precluded their use in practical applications.

The humidity stability photos and its related description (**Figure R6**) have been added to the revised Supplementary Information (**Page 29, Supplementary Fig.40**).

Figure R6 Photos of changes in (i)-(iv) under 75% humidity environments for 15 days ((i): (BTTP)₂MnBr₄, (ii): (BTTP)_{1.8}(HTPP)_{0.2}MnBr₄, (iii): (BTTP)_{1.8}(DTPP)_{0.2}MnBr₄, (iv): (BTTP)_{1.8}(CTPP)_{0.2}MnBr₄).

Revision checklist of manuscript:

1. Page 7, line 13, the word “consistent” was changed to “corresponds”.
2. Page 7, line 133; Page 8, line 180 the word “normalized” was added.
3. Page 8, line 166, Supplementary Fig. 6 was updated.
4. Page 8, line 183 the words “the normalized” were added.
5. Page 9, line 190, the sentence “Moreover, the PLQY was basically not impacted by small amount of doping with HTPP⁺, but it decreased after doping DTPP⁺ and CTPP⁺ due to their overly long chain lengths leading to excessive spatial separation and self-absorption (Fig. 2e,

Supplementary Fig. 9-12)” was updated.

6. Page 9, line 192, Supplementary Fig. 9-12 were added.
7. Page 9, line 192, “*Ref 43*” was added.
8. Page 11, line 246, the sentence “*As the temperature increased above 70°C, diffraction peaks gradually appeared and sharpen, accompanied preferred orientation along specific crystallographic planes.*” was updated.
9. Page 19, line 470, Supplementary Fig. 40 was added.
10. Page 27, line 694, the sentence “*UV-vis transmittance and absorption spectra of transparent and polycrystalline screens were obtained on a Cary 7000 spectrophotometer (Agilent, USA) equipped with an integrating sphere accessory.*” was updated.
11. Page 27, line 696, the sentence “*Raman spectra were obtained by the inVia-Qontor Raman microscope (Renishaw, England) with a laser ($\lambda = 785$ nm).*” was added.

Revision checklist of Supplementary Information:

1. Page 7, Supplementary Fig. 6 and the related texts were updated.
2. Page 8-11, Supplementary Fig. 9-12 and the related texts were added.
3. Page.10, “*Ref 4, 6, 7, 8*” were added.
4. Page 29, Supplementary Fig. 40 and the related texts were added.